# The random free field scalar theory

**Alessandro Piazza[a,b], Marco Serone[a,b], and Emilio Trevisani[c]**

[a]*SISSA, Via Bonomea 265, I-34136 Trieste, Italy*

[b]*INFN, Sezione di Trieste, Via Valerio 2, I-34127 Trieste, Italy*

[c]*Laboratoire de Physique Théorique et Hautes Énergies, CNRS & Sorbonne Université, 4 Place Jussieu, 75252 Paris, France*

`a.piazza@sissa.it, serone@sissa.it, trevisani@lpthe.jussieu.fr`

## Abstract

Quantum field theories with quenched disorder are so hard to study that even exactly solvable free theories present puzzling aspects. We consider a free scalar field $\phi$ in $d$ dimensions coupled to a random source $h$ with quenched disorder. Despite the presence of a mass scale governing the disorder distribution, we derive a new description of the theory that allows us to show that the theory is gapless and invariant under conformal symmetry, which acts in a non-trivial way on $\phi$ and $h$. This manifest CFT description reveals the presence of exotic continuous symmetries, such as nilpotent bosonic ones, in the quenched theory. We also reconsider Cardy's CFT description defined through the replica trick. In this description, the nilpotent symmetries reveal a striking resemblance with Parisi-Sourlas supersymmetries. We provide explicit maps of correlation functions between such CFTs and the original quenched theory. The maps are non-trivial and show that conformal behaviour is manifest only when considering suitable linear combinations of averages of products of correlators. We also briefly discuss how familiar notions like normal ordering of composite operators and OPE can be generalized in the presence of the more complicated local observables in the quenched theory.

# 1 Introduction

Theoretical modelling of real-world phenomena is necessarily built on approximation. Any experimental setup contains impurities which are virtually impossible to describe exactly at the microscopic level. Sometimes these impurities can be neglected, but in other cases, they cannot and can lead to a drastic change in the physics of the problem. When impurities are in thermal equilibrium with the system they define the so-called *annealed* type of disorder. When they are out of equilibrium in a fixed configuration we have the so-called *quenched* type of disorder. We will be interested in the case of *quenched* disorder. If we assume that impurities are random, a notable class of observables are taken by averaging over the impurities with a chosen distribution.

Probably the most notable model of quenched disorder is the Ising model on a $d$-dimensional lattice with random couplings. The case where the spin-spin interaction coupling is random is usually called "random bond" while a random magnetic field gives rise to the so-called "random field". The case of random field (RF) is particularly interesting because this type of disorder leads to drastic effects in the theory. The RF Ising model has a very long history and rich phenomenology, see e.g. [1–15].

In the continuum limit, lattice models of quenched disorder can be described by an ensemble of Euclidean QFTs where the coupling constants are space-dependent. More in general, one can study QFTs with quenched disorder independently of a UV lattice realization. QFTs with quenched disorder are notoriously very hard to study analytically. For example, basic notions of ordinary QFTs such as how to properly define an RG flow or how to write general selection rules have started to be analyzed for quenched disordered QFTs only very recently, see e.g. [16, 17]. In fact, such theories present theoretical challenges which make even free theories puzzling at first sight!

The aim of this paper is to elucidate a basic aspect of quenched QFTs which somehow has not been investigated much so far, namely how to identify a good set of local observables in the theory. In order to understand the problem, consider QFTs of the form

$$S[h] = S_0 + \int \mathrm{d}^d x \, h(x) \mathcal{O}_0(x) \,, \tag{1.1}$$

where $S_0$ is the action of a pure system, $h(x)$ is the random source, and $\mathcal{O}_0$ is a local operator of $S_0$. In an ordinary pure QFT, a good set of local observables are given by correlation functions of local operators. For instance, if we start from a UV CFT and the theory undergoes an RG flow, we can determine whether the theory flows to a conformal one (or scale-invariant only) or to a gapped theory by looking at the IR behaviour of such correlation functions.

In disordered theories, the "quenched" analogue of correlation functions of local operators are given by correlation functions at fixed random field $h$ configuration

$$\langle \mathcal{O}_1(x_1) \cdots \mathcal{O}_k(x_k) \rangle_h = \frac{\int [\mathrm{d}\mu] \, e^{-S[h]} \mathcal{O}_1(x_1) \cdots \mathcal{O}_k(x_k)}{\int [\mathrm{d}\mu] \, e^{-S[h]}} \,, \tag{1.2}$$

where $[\mathrm{d}\mu]$ denote the path integral measure and $\mathcal{O}_i$ are local operators of the theory $S_0$, which are then averaged against a statistical distribution $P[h]$ for the random field

$$\overline{\langle \mathcal{O}_1(x_1) \cdots \mathcal{O}_k(x_k) \rangle} = \int [\mathrm{d}h] P[h] \, \langle \mathcal{O}_1(x_1) \cdots \mathcal{O}_k(x_k) \rangle_h \,. \tag{1.3}$$

In the presence of disorder, however, we can also consider averages of products of correlators (not necessarily all independent of each other) of the form

$$\overline{\left\langle \prod_{j_1=1}^{k_1} \mathcal{O}_{j_1} \right\rangle \cdots \left\langle \prod_{j_N=1}^{k_N} \mathcal{O}_{j_N} \right\rangle}. \tag{1.4}$$

A couple of questions immediately arise: which correlation functions should we look at to determine the IR fate of the theory? How can we accommodate the existence of the additional correlators (1.4) within a pure CFT description? It is difficult to answer to these questions in some generality but, as mentioned, the above questions turn out to be non-trivial also for free theories, in the absence of an actual RG flow. In other words, in contrast to pure theories, the identification of the correct UV theory and its symmetries before adding deformations is not totally trivial.

We will then study in detail a free scalar field theory of the form (1.1) with

$$S_0 = \int \mathrm{d}^d x \, \frac{1}{2} (\partial \phi)^2 \,, \qquad \mathcal{O}_0(x) = -\phi(x) \,, \tag{1.5}$$

and take a Gaussian distribution for $P[h]$ defined in (1.3) as

$$P[h] \propto \exp\left( -\frac{1}{2v^2} \int \mathrm{d}^d x \, h(x)^2 \right). \tag{1.6}$$

We will denote such theory as the random free field theory (RFFT).[1] Despite the presence of the dimensionful parameter $v$ governing the width of the Gaussian distribution $P[h]$, we will show that the RFFT can be seen as a CFT, to all values of $v$, at any scale. The presence of the correlators (1.4) makes the mapping between the quenched average correlators and CFT correlators very non-trivial.

We will provide two different CFT descriptions of the RFFT. The first, denoted by $(\alpha, \beta)$ theory, is just based on a trivial, but rather peculiar, change of variables in the disordered theory. The second description, which already appeared in [13,14,18], is based on the replica trick defined in a suitable basis introduced by Cardy [4], and will be denoted as the Cardy theory.[2]

---

[1] Another relevant class of random theories is obtained by taking $\mathcal{O}_0 = \phi^2$, which describes the continuum limit of random bond lattice models. The resulting QFT in this case is not exactly solvable, so we will discuss in this paper only the random field case with $\mathcal{O}_0 = -\phi$.

[2] The CFT nature of the RFFT in the Cardy description was already recognized and worked out in some detail in [13, 14, 18]. However, the main emphasis of these works was on the RG of the random field Ising model seen as a deformation of a UV CFT. The latter captures the RFFT but the precise map to the RFFT and various aspects of the symmetries of the theory have not been considered.

The two descriptions are complementary in various respects. The $(\alpha, \beta)$ theory is the simplest and it is the most useful to reveal some of the exotic (including higher spin) symmetries of the RFFT, as well as to generalize properties like normal ordering and OPE in a quenched theory. This description is however hardly extended to interacting theories, since local deformations in the RFFT are generally mapped into non-local ones in the $(\alpha, \beta)$ theory. The Cardy description is the actual proper CFT description of the RFFT, mapping local deformations in the RFFT to local ones in the CFT. On the other hand, the analysis here is somewhat complicated by the categorical nature of the theory, which is based on a free vector model with $O(-2)$ symmetry. Yet we can identify the manifestations of the exotic symmetries found in the $(\alpha, \beta)$ theory above in the Cardy description, as well as other symmetries, providing in this way an alternative perspective possibly useful when studying deformations of the RFFT.

We start in section 2 by introducing the RFFT. We show why assigning a scaling dimension to $\phi$ does not make sense for $v \neq 0$ and how scaling invariance is realized in a peculiar way. We also discuss how the usual notion of normal ordering and OPE should be extended for averages of products of correlators like (1.4).

The $(\alpha, \beta)$ theory is discussed in section 3. In section 3.1 we show that the RFFT is equivalent to the direct sum of two free CFTs based on a higher derivative ($\Box^2$) and an ordinary ($\Box$) free scalar, denoted respectively as $\alpha$ and $\beta$. They define the $(\alpha, \beta)$ theory, whose action is (3.7). The $(\alpha, \beta)$ theory is an ordinary CFT, but the map with the most general correlator (1.4) of the RFFT requires to consider products of correlators. The map of local correlators between the two theories is reported in (3.14). The $(\alpha, \beta)$ theory is useful in elucidating how normal ordering of operators and OPE work in the RFFT, topics discussed in section 3.2, where various examples of correlator maps are also given. The global symmetries of the $(\alpha, \beta)$ theory are considered in section 3.3. Interestingly enough, even simple symmetries such as a $\mathbb{Z}_2$ symmetry $\beta \to -\beta$, when mapped back to the RFFT, reveals a non-manifest symmetry of the theory. These are exact disordered symmetries which emerge on average, in the terminology of [17]. The $(\alpha, \beta)$ theory reveals the existence of further interesting (continuous) nilpotent Heisenberg-like symmetries of the RFFT. These symmetries are intrinsic of the disordered theory and go beyond those discussed in [17]. Together with other (higher spin) symmetries, they give rise to a large set of selection rules among the correlators (1.4).

We discuss the Cardy description in section 4. We review the construction of this theory in section 4.1, which is based on the replica trick and on a way to take the $n \to 0$ limit at the action level by-passing possible subtleties occurring at $n = 0$ by a suitable change of field basis. Also the Cardy theory can be recast as the direct sum of two free CFTs, a higher derivative ($\Box^2$) and a $O(-2)$ free scalar theory (see (4.18) with $n \to 0$). The latter can formally be defined in terms of Deligne categories, as recently discussed [19], although such a description would not be needed for our purposes. The map of local correlators between the Cardy theory and RFFT is given in (4.19). We discuss various examples of correlator maps in section 4.2 and show how the set of Cardy correlators describes all possible observables of the RFFT, providing

an important consistency check of this description. Like the $(\alpha, \beta)$ theory, the Cardy theory enjoys a large set of symmetries, some of which are analyzed in section 4.3. We exemplify how symmetries translate into selection rules for the RFFT observables. While some symmetries, like $O(-2)$ and additional $\mathbb{Z}_2$ symmetries, are quite obvious and were already considered in [13–15], we show that new less trivial symmetries are present. Interestingly, while all new symmetries are bosonic, some of them take the form of a Heisinberg-like symmetry which is nilpotent (of degree three). We show how these share striking similarities with the fermionic Parisi-Sourlas superconformal symmetries discussed e.g. in [3, 18, 20]. Moreover, we explain that the replica permutation symmetry (which should be also preserved by interactions) can be written in terms of a combination of a $O(-2)$ rotation combined with a nilpotent transformation.

In section 5 we extend the discussion to the case in which $\phi$ is a generalized free field (GFF) of dimension $\Delta$. The derivation of the $(\alpha, \beta)$ theory and the Cardy theory can be easily applied to this case. The resulting theory is again a CFT, given by a direct sum of a GFF of dimension $2\Delta - \frac{d}{2}$ with a GFF $\Delta$ in the $(\alpha, \beta)$ theory, the latter being replaced by an $O(-2)$ GFF of dimension $\Delta$ in the Cardy case. The correlator mapping with the random field theory works in the same way as for the RFFT. This generalization is not totally academic, since we can induce an actual (though exactly tractable) RG flow by adding a $\phi^2$ deformation. When the perturbation is relevant, the IR theory is the direct sum of two identical GFFs of dimension $d - \Delta$ in the $(\alpha, \beta)$ theory, one of the two being replaced again by an $O(-2)$ GFF of the same dimension in the Cardy description.

We summarize our results and discuss possible future developments in the concluding section 6. Two appendices complement the paper. In appendix A we discuss the spectrum of primary operators in the $(\alpha, \beta)$ theory,[3] while in appendix B we provide further details on correlator mapping between the RFFT and the Cardy description.

## 2 Random field description

The action of the RFFT is defined by (1.1), (1.5), and (1.6). Ordinary dimensional analysis fixes $[v] = 1$, where $v$ is the width of the Gaussian distribution (1.6). In a free theory, we have the luxury to do computations directly in the disordered theory, by-passing the replica trick. Correlators involving only the elementary field $\phi$ can be obtained from the following generating functional:

$$Z[J, h] = \int [\mathrm{d}\phi] e^{-S + \int J\phi} = \exp\left(\frac{1}{2} \int \mathrm{d}^d x \, \mathrm{d}^d y \, (h(x) + J(x)) G(x - y)(h(y) + J(y))\right), \quad (2.1)$$

---

[3]In non-unitary CFTs, such as the $\Box^2$ $\alpha$-theory, it can happen that a primary operator at the unitary bound does not lead to a short multiplet, and that null states do not completely decouple. This leads to a peculiar case of multiplet recombination of operators. The phenomenon has been studied in [21] and explained in terms of "extended Verma modules", with operators which are neither primary nor descendants. We observe (by inspection in $d = 5$ and $d = 6$) that the characters of the "extended Verma modules" can be recast as the sum of two characters of standard Verma modules (corresponding to the primary and the first null primary-descendant).

where

$$G(x) = \frac{\kappa}{|x|^{d-2}}, \qquad \kappa = \frac{\Gamma\left(\frac{d}{2}-1\right)}{4\pi^{\frac{d}{2}}}, \tag{2.2}$$

is the massless propagator of a free scalar in $d$ dimensions. Upon taking functional derivatives of $Z[J, h]$ with respect to $J$ and averaging over $h$, we get

$$\overline{\langle \phi(x)\phi(y)\rangle} = G(x-y) + v^2 \int \mathrm{d}^d z\, G(x-z)G(y-z) = \frac{\kappa}{|x-y|^{d-2}} + \frac{\tilde{\kappa}}{|x-y|^{d-4}}, \tag{2.3}$$

where we used $\overline{h(x)h(y)} = v^2 \delta^{(d)}(x-y)$ and we defined

$$\tilde{\kappa} = \frac{v^2}{2(d-4)}\kappa. \tag{2.4}$$

We also have

$$\langle \phi(x)\rangle_h = \int \mathrm{d}^d y\, G(x-y)h(y) \neq 0, \qquad \overline{\langle \phi(x)\rangle} = 0. \tag{2.5}$$

The second term in (2.3) corresponds to the disconnected average:

$$\overline{\langle \phi(x)\rangle \langle \phi(y)\rangle} = v^2 \int \mathrm{d}^d z\, G(x-z)G(y-z) = \frac{\tilde{\kappa}}{|x-y|^{d-4}}, \tag{2.6}$$

so that the averaged connected component of the two-point function reads

$$\overline{\langle \phi(x)\phi(y)\rangle_{\mathrm{c}}} = \frac{\kappa}{|x-y|^{d-2}}. \tag{2.7}$$

The correlator (2.6) does not decay with distance when $d \leq 4$. In general, correlators in the RFFT satisfy cluster decomposition only for $d > 4$, so we will focus on this case in what follows. We can also compute correlators of composite operators. For example, the two-point function of $\phi^2$ reads

$$\overline{\langle \phi^2(x)\phi^2(y)\rangle} = \frac{2\kappa^2}{|x-y|^{2d-4}} + \frac{2\kappa\tilde{\kappa}}{|x-y|^{2d-6}} + \frac{2\tilde{\kappa}^2}{|x-y|^{2d-8}}, \tag{2.8}$$

where, as we will explain later, we have normal ordered $\phi^2$ such that $\overline{\langle \phi^2(x)\rangle} = 0$. Similarly we can compute average correlators of higher $n$-point functions $\overline{\langle \mathcal{O}_1(x_1)\dots\mathcal{O}_n(x_n)\rangle}$ of local operators $\mathcal{O}_i$. In the presence of disorder, we can also consider more general averages of products of correlators of the form (1.4). The pattern for a generic correlator is similar to (2.3) and (2.8). Correlators are given by a sum of terms which decay as power-like. The mass scale $v$ enters polynomially ($\tilde{\kappa} \propto v^2$) and is responsible for the presence of different powers of $|x-y|$ in the correlators.

There are peculiar features of this theory which are confusing at first sight. The theory is manifestly gapless since all correlators are power-like suppressed. A gapless theory generally is expected to be conformal invariant, or at least scale-invariant, but at the same time, it has a mass scale, the Gaussian width $v$. The presence of this mass scale allows us in principle to define an IR of the theory, as $v|x| \gg 1$. From this point of view, we might naively interpret the exact two-point function (2.3) as an RG flow from a UV where the scaling dimension $\Delta_\phi^{\mathrm{UV}} = (d-2)/2$

to an IR where $\Delta_\phi^{\mathrm{IR}} = (d-4)/2$. However, a UV scaling $(d-2)/2$ is not compatible with (2.6), which would suggest a scaling $(d-4)/2$ to all scales. A consistent IR scaling is also problematic, as has been shown in [22] by looking at the behaviour of the connected two and three-point functions of $\phi^2$. This puzzling behaviour led [22] to conclude that the RFFT cannot be scale invariant. As we will see and develop in the rest of the paper, in contrast, the theory is actually conformal invariant, but the identification of which operators should have a well-defined scaling dimension (i.e. are primary operators of the putative CFT) is non-trivial. The Gaussian width $v$ is dimensionless with this modified scaling assignment, and no RG flow takes place, the RFFT being the same CFT for all finite values of $v \neq 0$.[4]

In order to simplify the notation, it is useful to often omit to write integrals and the space dependencies of the fields $\phi$ and the random coupling $h$. We then have

$$S_0(\phi) = \frac{1}{2}\phi K \phi \,, \qquad Z[h] = Z[J = 0, h] = e^{\frac{1}{2}hGh} \,, \qquad (2.9)$$

where

$$K(x,y) = -\Box_x \delta^{(d)}(x-y) \,, \qquad (2.10)$$

and $G = K^{-1}$ is given in (2.2).

For the same reason, we denote simply by $A(\phi) = \mathcal{O}_1(x_1)\cdots\mathcal{O}_k(x_k)$ a product of local operators at different points, and write

$$\langle A(\phi)\rangle_h = \frac{1}{Z[h]}\int [\mathrm{d}\phi] e^{-S_0 + h\phi} A(\phi) \,, \qquad \overline{\langle A(\phi)\rangle_h} = \int [\mathrm{d}h] e^{-\frac{h^2}{2v^2}} \langle A(\phi)\rangle_h \,. \qquad (2.11)$$

**Conformal symmetry**   The RFFT is invariant under a scaling symmetry $x \to \lambda x$ where the fields $\phi$ and $h$ transform as follows:

$$\phi \to \lambda^{-\frac{d-2}{2}}(\phi - Gh) + \lambda^{-\frac{d-4}{2}}Gh \,,$$
$$h \to \lambda^{-\frac{d}{2}}h \,. \qquad (2.12)$$

The origin of (2.12) will be clear when we will discuss the $(\alpha, \beta)$ theory in section 3. The scaling of $h$ is precisely what is needed to have a scale-invariant Gaussian measure (1.6). The transformation of the action is easily established:

$$S = \frac{1}{2}\phi K \phi - h\phi \quad \to \quad \frac{\lambda^d}{2}\left(\lambda^{-\frac{d-2}{2}}(\phi - Gh) + \lambda^{-\frac{d-4}{2}}Gh\right)\lambda^{-2}K\left(\lambda^{-\frac{d-2}{2}}(\phi - Gh) + \lambda^{-\frac{d-4}{2}}Gh\right)$$
$$- \lambda^d \lambda^{-\frac{d}{2}}h\left(\lambda^{-\frac{d-2}{2}}(\phi - Gh) + \lambda^{-\frac{d-4}{2}}Gh\right) \qquad (2.13)$$
$$= S + F[h] \,, \qquad F[h] = \frac{1}{2}(1 - \lambda^2)hGh \,.$$

Note that the action is *not* invariant, but the non-invariance is given by a functional of $h$ only. The same functional is obtained in the denominator of (1.2) when computing normalized correlators,

---

[4]Of course, at $v = 0$ all correlators collapse to the ones of an ordinary free scalar field (and of products thereof).

so that such factors cancel. Before average, we have

$$\left\langle A\left(\phi'[\phi,h]\right)\right\rangle_h = \langle A(\phi)\rangle_{h'[h]}, \tag{2.14}$$

where $h'(x) = \lambda^{d/2} h(\lambda x)$. For averages of arbitrary products of correlators, we can always absorb the change in $h$ by a change of variables in the $h$ path-integral. Therefore, on average we have "scaling invariance" in the following sense:

$$\overline{\prod_{i=1}^{k}\left\langle A_i\left(\lambda^{\frac{d-2}{2}}\phi(\lambda x) + \left(\lambda^{\frac{d-4}{2}} - \lambda^{\frac{d-2}{2}}\right)\langle\phi(\lambda x)\rangle_h\right)\right\rangle_h} = \overline{\prod_{i=1}^{k}\langle A_i(\phi(x))\rangle_h}. \tag{2.15}$$

Consider for example the two-point function $\overline{\langle\phi(x_1)\phi(x_2)\rangle}$. This is not scale invariant in the usual sense (it is a sum of two homogeneous pieces with different scaling), but under the above transformation we have

$$\overline{\langle\phi'(x_1)\phi'(x_2)\rangle} = \lambda^{d-2}\overline{\langle\phi(\lambda x_1)\phi(\lambda x_2)\rangle} + \left(\lambda^{d-4} - \lambda^{d-2}\right)\overline{\langle\phi(\lambda x_1)\rangle\langle\phi(\lambda x_2)\rangle} = \overline{\langle\phi(x_1)\phi(x_2)\rangle}. \tag{2.16}$$

One can similarly verify invariance under inversion symmetry which is however significantly more involved and will not be reported. The invariance under rotations is automatic, while invariance under translations is explicitly broken by the random field coupling (1.1) but is restored after averaging. Altogether, this shows that the RFFT enjoys conformal symmetry. Finally, the scaling transformation (2.12) makes clear that the Gaussian width $v$ is dimensionless. Note that insisting in assigning $[v] = 1$ and hence to take a "RG picture", is inconsistent. As discussed, a UV scaling $(d-2)/2$ (for $v|x| \ll 1$) and an IR scaling $(d-4)/2$ (for $v|x| \gg 1$) cannot be assigned to $\phi$ consistently at operator level across observables of the type (1.4).

A more complicated example is given by four-point correlators of the elementary field. In this case, the set of average correlators in a RFFT is significantly larger than correlators in an ordinary theory, due to the possibility of taking averages of products of correlators. They take the form

$$V_{\text{RF}} = \left\{\overline{\langle\phi(x_1)\rangle\langle\phi(x_2)\rangle\langle\phi(x_3)\rangle\langle\phi(x_4)\rangle}, \overline{\langle\phi(x_1)\rangle\langle\phi(x_3)\rangle\langle\phi(x_2)\phi(x_4)\rangle}, \overline{\langle\phi(x_1)\rangle\langle\phi(x_2)\phi(x_3)\rangle\langle\phi(x_4)\rangle}, \right.$$

$$\overline{\langle\phi(x_1)\rangle\langle\phi(x_2)\rangle\langle\phi(x_3)\phi(x_4)\rangle}, \overline{\langle\phi(x_2)\rangle\langle\phi(x_1)\phi(x_3)\rangle\langle\phi(x_4)\rangle}, \overline{\langle\phi(x_2)\rangle\langle\phi(x_3)\rangle\langle\phi(x_1)\phi(x_4)\rangle},$$

$$\overline{\langle\phi(x_1)\phi(x_2)\rangle\langle\phi(x_3)\rangle\langle\phi(x_4)\rangle}, \overline{\langle\phi(x_1)\phi(x_3)\rangle\langle\phi(x_2)\phi(x_4)\rangle}, \overline{\langle\phi(x_2)\phi(x_3)\rangle\langle\phi(x_1)\phi(x_4)\rangle},$$

$$\overline{\langle\phi(x_1)\phi(x_2)\rangle\langle\phi(x_3)\phi(x_4)\rangle}, \overline{\langle\phi(x_1)\rangle\langle\phi(x_2)\phi(x_3)\phi(x_4)\rangle}, \overline{\langle\phi(x_2)\rangle\langle\phi(x_1)\phi(x_3)\phi(x_4)\rangle},$$

$$\left.\overline{\langle\phi(x_3)\rangle\langle\phi(x_1)\phi(x_2)\phi(x_4)\rangle}, \overline{\langle\phi(x_4)\rangle\langle\phi(x_1)\phi(x_2)\phi(x_3)\rangle}, \overline{\langle\phi(x_1)\phi(x_2)\phi(x_3)\phi(x_4)\rangle}\right\}. \tag{2.17}$$

While each term in $V_{\text{RF}}$ does not scale like a CFT four-point function, we will show that in the correct variables, these appear in linear combinations which define standard CFT correlators.

Summarizing, the conformal symmetry of the original free theory action is exactly restored after quenching on average correlators. While conformal symmetry was present also in the pure theory, the action on the fields of the disordered conformal symmetry is different from the pure

one. For example, while $\phi$ is a good primary operator under the pure conformal symmetry, it fails to be so under the average conformal symmetry because of a non-trivial mixing between the field $\phi$ and the random coupling $h$, as we have seen. This kind of symmetry restoration is beyond those recently discussed in [17].

**Normal ordering**  A feature of disordered theories that we will discuss in the next sections in the context of the RFFT is the appearance of a new form of coincident point singularities. As well-known, even in ordinary free quantum field theories, naive composite operators are affected by coincident point singularities. The usual way to regulate these divergences in free theories is to use normal ordering, namely, we define e.g. the operator $:\phi^n:$ as the regulated form of the operator $\phi^n$, where all self-contractions among the $n$ fields have been subtracted. When inserted in a correlation function, this operation amounts to neglect all Wick contractions which involve the elementary fields within $\phi^n$. In a RFFT we can similarly have coincident point singularities associated with a composite operator, e.g.

$$\overline{\langle \phi(x_1)^2\, \phi(x_2)\phi(x_3)\rangle}\,, \tag{2.18}$$

but also new forms of coincident point singularities when two fields in *different* correlators approach the same space-time point, e.g.

$$\overline{\langle\phi(x_1)\rangle\,\langle\phi(x_1)\phi(x_2)\phi(x_3)\rangle}\,. \tag{2.19}$$

The normal ordering in principle could be performed, for each given correlator, by subtracting the singularity through an explicit computation. For example by computing the correlator (2.18)

$$
\begin{aligned}
\overline{\langle\phi(x_1)^2\phi(x_2)\phi(x_3)\rangle} =\ & 2\left(\frac{\kappa}{|x_{12}|^{d-2}}+\frac{\tilde{\kappa}}{|x_{12}|^{d-4}}\right)\left(\frac{\kappa}{|x_{13}|^{d-2}}+\frac{\tilde{\kappa}}{|x_{13}|^{d-4}}\right) \\
& + \lim_{x_1'\to x_1}\left(\frac{\kappa}{|x_1-x_1'|^{d-2}}+\frac{\tilde{\kappa}}{|x_1-x_1'|^{d-4}}\right)\left(\frac{\kappa}{|x_{23}|^{d-2}}+\frac{\tilde{\kappa}}{|x_{23}|^{d-4}}\right),
\end{aligned}
\tag{2.20}
$$

we can interpret the divergent part as the disconnected correlator $\overline{\langle\phi(x_1)^2\rangle}\ \overline{\langle\phi(x_2)\phi(x_3)\rangle}$, which indeed equals the second line of (2.20). We can thus define the normal ordered correlator in the RFFT as

$$\overline{\langle :\phi^2(x_1):\,\phi(x_2)\phi(x_3)\rangle} = \overline{\langle\phi(x_1)^2\phi(x_2)\phi(x_3)\rangle} - \overline{\langle\phi(x_1)^2\rangle}\ \overline{\langle\phi(x_2)\phi(x_3)\rangle}\,. \tag{2.21}$$

Similarly for the more exotic singularity appearing in (2.19) an explicit computation shows that the divergent piece equals $\overline{\langle\phi(x_1)\rangle^2}\ \overline{\langle\phi(x_2)\phi(x_3)\rangle}$. We can thus define a regularized correlator

$$\overline{:\langle\phi(x_1)\rangle\,\langle\phi(x_1):\,\phi(x_2)\phi(x_3)\rangle} = \overline{\langle\phi(x_1)\rangle\,\langle\phi(x_1)\phi(x_2)\phi(x_3)\rangle} - \overline{\langle\phi(x_1)\rangle^2}\ \overline{\langle\phi(x_2)\phi(x_3)\rangle}\,. \tag{2.22}$$

Regularizing specific correlators is however not enough. In order to define a normal ordering at the *operatorial* level, we have to regularize at once all possible correlators where $\phi^2$ can be inserted. In section 3.2 we will show how to do that and we will see that the needed subtractions

are automatically implemented by the usual normal ordering in the two CFT descriptions. In particular, we will see that operatorially

$$:\phi^2(x): = \phi^2(x) - \overline{\langle \phi^2(x) \rangle}\,, \tag{2.23}$$

valid when inserted in any average of product of correlators. Similarly in order to regulate the new forms of coincident point singularities we have

$$\overline{:\langle \phi(x)A_1(\phi) \rangle \, \langle \phi(x): A_2(\phi) \rangle \, \mathcal{B}} = \overline{\langle \phi(x)A_1(\phi) \rangle \, \langle \phi(x)A_2(\phi) \rangle \, \mathcal{B}} - \overline{\langle \phi(x) \rangle^2} \, \overline{\langle A_1(\phi) \rangle \, \langle A_2(\phi) \rangle \, \mathcal{B}}\,, \tag{2.24}$$

where $\mathcal{B}$ schematically denotes products of correlation functions with local operators supported away from $x$.

**OPE**   A similar remark works for the leading OPE singularity, which takes the following form

$$\lim_{x \to 0} \quad \overline{\langle \phi(x)\phi(0)A(\phi) \rangle \, \mathcal{B}} \sim \overline{\langle \phi(x)\phi(0) \rangle} \times \overline{\langle A(\phi) \rangle \, \mathcal{B}} \; + \; \text{reg.}\,, \tag{2.25}$$

$$\lim_{x \to 0} \quad \overline{\langle \phi(x)A_1(\phi) \rangle \, \langle \phi(0)A_2(\phi) \rangle \, \mathcal{B}} \sim \overline{\langle \phi(x) \rangle \, \langle \phi(0) \rangle} \times \overline{\langle A_1(\phi) \rangle \, \langle A_2(\phi) \rangle \, \mathcal{B}} \; + \; \text{reg.}\,, \tag{2.26}$$

where reg. denotes regular terms. These leading OPE will also serve as a tool to show the operatorial definitions for the normal ordering. For example from (2.25) we obtain

$$\overline{\langle :\phi^2(x): A(\phi) \rangle \mathcal{B}} = \overline{\langle \phi^2(x)A(\phi) \rangle \mathcal{B}} - \overline{\langle \phi^2(x) \rangle} \, \overline{\langle A(\phi) \rangle \, \mathcal{B}}\,, \tag{2.27}$$

which directly proves (2.23). Similarly we see that the relation (2.24) naturally follows from (2.26). In section 3.2 we will show how (2.25) and (2.26) can be proved using the $(\alpha, \beta)$ theory.

# 3   CFT description I: $(\alpha, \beta)$ theory

We start in section 3.1 by defining the $(\alpha, \beta)$ theory and by showing how arbitrary average correlators in the RFFT are mapped to this theory. Various examples of this map and its inverse, as well as the normal ordering of composite operators and OPE in the RFFT, are analyzed in section 3.2. We discuss in section 3.3 the symmetries of the theory. The spectrum of the theory is discussed in appendix A.

## 3.1   Derivation of the $(\alpha, \beta)$ theory

The first CFT description of the RFFT is based on simple manipulations in which the external field $h$ is effectively treated as a quantum field. Within quenched disorder, this is possible because we can exactly compute $Z[h]$ appearing in the denominator of (1.2), reported in (2.9). Consider first the average of a single general $n$-point correlation function of composite operators:

$$\overline{\langle A(\phi) \rangle_h} = \int [\mathrm{d}h] e^{-\frac{h^2}{2v^2}} e^{-\frac{1}{2}hGh} \int [\mathrm{d}\phi] e^{-S_0 + h\phi} A(\phi)\,. \tag{3.1}$$

We can put together the two path integral measures and interpret the $hGh$ term as part of a new non-local action $\widetilde{S}$ involving both $h$ and $\phi$ as quantum fields:

$$\overline{\langle A(\phi)\rangle_h} = \int [\mathrm{d}h][\mathrm{d}\phi] e^{-\widetilde{S}(h,\phi)} A(\phi) \,, \tag{3.2}$$

where

$$\widetilde{S}(h,\phi) = \frac{h^2}{2v^2} + \frac{1}{2}hGh + \frac{1}{2}\phi K\phi - h\phi \,. \tag{3.3}$$

The action $\widetilde{S}$ is diagonalized by performing the following change of variables

$$\phi = \alpha + \beta \,, \qquad h = K\alpha \,. \tag{3.4}$$

Since the transformation is linear in the fields, the Jacobian carries no field dependence. Thanks to the form of the transformation of $h$ in (3.4), the action is now *local* in $\alpha$ and $\beta$:

$$S_{(\alpha,\beta)} = \frac{1}{2v^2}(K\alpha)^2 + \frac{1}{2}\beta K\beta \,. \tag{3.5}$$

The action (3.5) defines the $(\alpha,\beta)$ theory, namely a CFT which is the sum of two free CFTs: a non-unitary free scalar with a higher-derivative $\square^2$ kinetic term ($\alpha$) and an ordinary free scalar ($\beta$). The scaling dimensions of the two fields are

$$\Delta_\alpha = \frac{d-4}{2} \,, \qquad\qquad \Delta_\beta = \frac{d-2}{2} \,. \tag{3.6}$$

In a less compact notation, the action (3.5) reads

$$S_{(\alpha,\beta)} = \int \mathrm{d}^d x \left( \frac{1}{2v^2}(\square\alpha)^2 + \frac{1}{2}(\partial\beta)^2 \right) \,. \tag{3.7}$$

Plugging the action (3.5) in (3.2) gives

$$\overline{\langle A(\phi)\rangle_h} = \int [\mathrm{d}\alpha][\mathrm{d}\beta] e^{-S_{(\alpha,\beta)}} A(\phi = \alpha + \beta) \,. \tag{3.8}$$

Averages of products of multiple correlation functions of composite operators

$$\overline{\prod_{i=1}^{k}\langle A_i(\phi)\rangle_h} = \int [\mathrm{d}h] e^{-\frac{h^2}{2v^2}} \prod_{i=1}^{k}\langle A_i(\phi)\rangle_h \,, \tag{3.9}$$

can also be mapped in the $(\alpha,\beta)$ theory, but they require a bit more work. First, we define $h = K\alpha$ in (3.9), namely we change variables (3.4) only for the external coupling $h$. In this way we get

$$\overline{\prod_{i=1}^{k}\langle A_i(\phi)\rangle_h} = \int [\mathrm{d}\alpha] e^{-\frac{(K\alpha)^2}{2v^2}} \prod_{i=1}^{k}\langle A_i(\phi)\rangle_{K\alpha} \,. \tag{3.10}$$

The individual blocks $\langle A_i(\phi)\rangle_{K\alpha}$ are easily evaluated:

$$\langle A_i(\phi)\rangle_{K\alpha} = e^{-\frac{1}{2}\alpha K\alpha} \int [\mathrm{d}\phi] e^{-\frac{1}{2}\phi K\phi + \phi K\alpha} A_i(\phi) = \int [\mathrm{d}\beta] e^{-\frac{1}{2}\beta K\beta} A_i(\alpha + \beta) \,, \tag{3.11}$$

where in the last step we changed variables from $\phi$ to $\beta$ via a shift: $\phi = \beta + \alpha$. We can also Taylor expand the factors $A_i$, taken to be polynomials in the fields:

$$\langle A_i(\alpha + \beta)\rangle_{K\alpha} = \sum_{l \geq 0} \frac{1}{l!} \int \mathrm{d}^d x_1 \cdots \mathrm{d}^d x_l \, \langle A_i^{(l)}[\beta]\rangle \, \alpha(x_1) \cdots \alpha(x_l)\,, \qquad (3.12)$$

where we have defined

$$A_i^{(l)}[\beta] = A_i^{(l)}[\beta](x_1, \ldots, x_l) \equiv \frac{\delta}{\delta f(x_1)} \cdots \frac{\delta}{\delta f(x_l)} A(f)\bigg|_{f=\beta}\,. \qquad (3.13)$$

The $n$-th derivatives are now a function of $\beta$ only. Proceeding in this way in all the terms in (3.9), the correlator turns into a product of independent correlators determined by the free theory ($\beta$), while the products of fields $\alpha$ appearing in all the $A_i$ combine into a unique correlator in the $\alpha$ theory. Putting everything together, we arrive at the final formula

$$\overline{\prod_{i=1}^{k} \langle A_i(\phi)\rangle_h} = \sum_{l_i \geq 0} \frac{1}{l_1! \cdots l_n!} \left(\int \prod_{i=1}^{k} \prod_{m=1}^{l_i} \mathrm{d}^d y_m^{(i)}\right) \left(\prod_{i=1}^{k} \Big\langle A_i^{(l_i)}[\beta]\Big\rangle\right) \Big\langle \prod_{i=1}^{k} \prod_{m=1}^{l_i} \alpha(y_m^{(i)})\Big\rangle\,. \qquad (3.14)$$

For $k = 1$ (3.14) boils down to the Taylor expansion of (3.8), as it should be. The CFT nature of the RFFT is made manifest in the $(\alpha, \beta)$ description. Any average correlator in the RFFT is mapped to a linear combination of products of correlators in a pure (non-unitary) CFT.

We also have a map from a single $n$-point correlator in the $(\alpha, \beta)$ theory to the RFFT. The inverse of (3.4) is

$$\beta = \phi - \langle\phi\rangle_h\,, \qquad \alpha = \langle\phi\rangle_h\,, \quad \text{with} \quad \langle\phi\rangle_h = Gh\,, \qquad (3.15)$$

and hence

$$\Big\langle \prod_{i=1}^{n} \mathcal{O}_i(\alpha(x_i), \beta(x_i))\Big\rangle = \overline{\Big\langle \prod_{i=1}^{n} \mathcal{O}_i\Big(\alpha = \langle\phi(x_i)\rangle_h\,, \beta = \phi(x_i) - \langle\phi(x_i)\rangle_h\Big)\Big\rangle_h}\,. \qquad (3.16)$$

We will discuss these maps and other features of the $(\alpha, \beta)$ theory in more detail in the rest of this section.

Finally, let us comment on the role of the Gaussian width $v$ in (3.7). As long as $v \neq 0$, this parameter could be removed from the CFT description with a field redefinition $\alpha \to v\alpha$. This would change however the map (3.14) where $v$ factors would enter explicitly. For the sake of having lighter formulas, we have decided to keep the convention where $v^2$ is included in the $\alpha$ kinetic term and not in the map with the RFFT.

## 3.2 Correlator mapping, OPE, and normal ordering

We elucidate here the non-trivial map (3.14) by discussing a few relevant examples, starting from correlators involving elementary fields only. The simplest case is the average of a single 2-point

correlator $A(\phi) = \phi(x_1)\phi(x_2)$. Using (3.14) or the simpler (3.8), we have

$$
\begin{aligned}
\overline{\langle\phi(x_1)\phi(x_2)\rangle} &= \langle\alpha(x_1)\alpha(x_2)\rangle + \langle\alpha(x_1)\rangle\langle\beta(x_2)\rangle + \langle\beta(x_1)\rangle\langle\alpha(x_2)\rangle + \langle\beta(x_1)\beta(x_2)\rangle \,, \\
&= \langle\alpha(x_1)\alpha(x_2)\rangle + \langle\beta(x_1)\beta(x_2)\rangle = G_\beta(x_{12}) + G_\alpha(x_{12}) \,,
\end{aligned}
\tag{3.17}
$$

where terms with an odd number of fields vanish by $\mathbb{Z}_2$ symmetry and $x_{ij} = x_i - x_j$. The two propagators $G_\alpha$ and $G_\beta$ are defined as[5]

$$
\begin{aligned}
G_\alpha(x_{12}) &= v^2 \int \frac{\mathrm{d}^d p}{(2\pi)^d} \frac{1}{(p^2)^2} e^{ip\cdot x_{12}} = \frac{\tilde{\kappa}}{|x_{12}|^{d-4}} \,, \\
G_\beta(x_{12}) &= \int \frac{\mathrm{d}^d p}{(2\pi)^d} \frac{1}{p^2} e^{ip\cdot x_{12}} = \frac{\kappa}{|x_{12}|^{d-2}} \,,
\end{aligned}
\tag{3.19}
$$

where $\kappa$ and $\tilde{\kappa}$ are respectively given in (2.2) and (2.4). Of course (3.17) reproduces the result (2.3) discussed previously, but it makes manifest that the correlator $\overline{\langle\phi(x_1)\phi(x_2)\rangle}$ does not qualify as a two-point function of a primary field in a CFT, but is rather the sum of 2 two-point functions of two distinct CFT primaries. In contrast, the combinations of correlation functions in the RFFT

$$
\begin{aligned}
\overline{\langle\phi(x_1)\phi(x_2)\rangle} - \overline{\langle\phi(x_1)\rangle\langle\phi(x_2)\rangle} &= \langle\beta(x_1)\beta(x_1)\rangle = G_\beta(x_{12}) \,, \\
\overline{\langle\phi(x_1)\rangle\langle\phi(x_2)\rangle} &= \langle\alpha(x_1)\alpha(x_2)\rangle = G_\alpha(x_{12}) \,,
\end{aligned}
\tag{3.20}
$$

map to proper 2-point correlators in a CFT. Higher point correlators can be treated similarly. For example, we have

$$
\begin{aligned}
\overline{\langle\phi(x_1)\phi(x_2)\phi(x_3)\phi(x_4)\rangle} &= \langle\beta(x_1)\beta(x_2)\beta(x_3)\beta(x_4)\rangle \\
&+ (\langle\beta(x_1)\beta(x_2)\rangle\langle\alpha(x_3)\alpha(x_4)\rangle + 5\,\mathrm{perms.}) \\
&+ \langle\alpha(x_1)\alpha(x_2)\alpha(x_3)\alpha(x_4)\rangle \,.
\end{aligned}
\tag{3.21}
$$

We can also consider averages of products of more than one correlator. For example, using (3.14), we have

$$
\begin{aligned}
\overline{\langle\phi(x_1)\phi(x_2)\rangle\langle\phi(x_3)\phi(x_4)\rangle} &= \langle\beta(x_1)\beta(x_2)\rangle\langle\beta(x_3)\beta(x_4)\rangle \\
&+ \langle\beta(x_1)\beta(x_2)\rangle\langle\alpha(x_3)\alpha(x_4)\rangle + \langle\alpha(x_1)\alpha(x_2)\rangle\langle\beta(x_3)\beta(x_4)\rangle \\
&+ \langle\alpha(x_1)\alpha(x_2)\alpha(x_3)\alpha(x_4)\rangle \,.
\end{aligned}
\tag{3.22}
$$

We see again that an average of products of correlators does not define a proper 4-point function in a CFT, but a linear combination of correlators. Moreover, they also involve products of correlation functions in the $(\alpha, \beta)$ description, as follows from (3.14).

The map in the $(\alpha, \beta)$ theory of averages of products of correlators becomes quite involved as the number of fields increases. For illustration, we report the general map involving four

---

[5]Recall that

$$
\int \frac{\mathrm{d}^d p}{(2\pi)^d} \frac{1}{(p^2)^\alpha} e^{ip\cdot(x-y)} = \frac{\Gamma(d/2-\alpha)}{4^\alpha \pi^{d/2}\Gamma(\alpha)} \frac{1}{|x-y|^{d-2\alpha}} \,.
\tag{3.18}
$$

elementary fields. On the random field side, the possible 4-point functions are given by $V_{\mathrm{RF}}$ defined in (2.17), while in the $(\alpha, \beta)$ theory the possible 4-point structures, taking products of $\beta$-correlators into account, can be grouped into the following vector:

$$
\begin{aligned}
V_{(\alpha,\beta)} = \Big\{ & \langle \alpha(x_1)\alpha(x_2)\alpha(x_3)\alpha(x_4) \rangle , \langle \alpha(x_1)\alpha(x_3) \rangle \langle \beta(x_2)\beta(x_4) \rangle , \langle \alpha(x_1)\alpha(x_4) \rangle \langle \beta(x_2)\beta(x_3) \rangle , \\
& \langle \alpha(x_1)\alpha(x_2) \rangle \langle \beta(x_3)\beta(x_4) \rangle , \langle \alpha(x_2)\alpha(x_4) \rangle \langle \beta(x_1)\beta(x_3) \rangle , \langle \alpha(x_2)\alpha(x_3) \rangle \langle \beta(x_1)\beta(x_4) \rangle , \\
& \langle \alpha(x_3)\alpha(x_4) \rangle \langle \beta(x_1)\beta(x_2) \rangle , \langle \beta(x_1)\beta(x_3) \rangle \langle \beta(x_2)\beta(x_4) \rangle , \langle \beta(x_2)\beta(x_3) \rangle \langle \beta(x_1)\beta(x_4) \rangle , \\
& \langle \beta(x_1)\beta(x_2) \rangle \langle \beta(x_3)\beta(x_4) \rangle , \langle \alpha(x_1)\beta(x_2)\beta(x_3)\beta(x_4) \rangle , \langle \beta(x_1)\alpha(x_2)\beta(x_3)\beta(x_4) \rangle \\
& \langle \beta(x_1)\beta(x_2)\alpha(x_3)\beta(x_4) \rangle , \langle \beta(x_1)\beta(x_2)\beta(x_3)\alpha(x_4) \rangle , \langle \beta(x_1)\beta(x_2)\beta(x_3)\beta(x_4) \rangle \Big\} .
\end{aligned}
\tag{3.23}
$$

There is a linear invertible map relating the two vectors of observables

$$
V_{\mathrm{RF}} = M_{(\alpha\beta)} \, V_{(\alpha,\beta)} , \qquad
M_{(\alpha\beta)} =
\begin{pmatrix}
1 & & & & & & & & & & & & & & \\
1 & 1 & & & & & & & & & & & & & \\
1 & & 1 & & & & & & & & & & & & \\
1 & & & 1 & & & & & & & & & & & \\
1 & & & & 1 & & & & & & & & & & \\
1 & & & & & 1 & & & & & & & & & \\
1 & & & & & & 1 & & & & & & & & \\
1 & 1 & & & 1 & & & 1 & & & & & & & \\
1 & & 1 & & & 1 & & & 1 & & & & & & \\
1 & & & 1 & & & 1 & & & 1 & & & & & \\
1 & 1 & 1 & 1 & & & & & & & 1 & & & & \\
1 & & & 1 & 1 & 1 & & & & & & 1 & & & \\
1 & 1 & & & & 1 & 1 & & & & & & 1 & & \\
1 & & 1 & & 1 & & 1 & & & & & & & 1 & \\
1 & 1 & 1 & 1 & 1 & 1 & 1 & & & & 1 & 1 & 1 & 1 & 1
\end{pmatrix} ,
\tag{3.24}
$$

where the empty entries are zero. Note that the basis of 4-point functions defining $V_{(\alpha,\beta)}$ includes manifestly vanishing correlators whenever we have an odd number of $\alpha$ or $\beta$ fields. When mapped to RFFT observables, we get non-trivial selection rules. We postpone a discussion of symmetries and selection rules to section 3.3.

We can use (3.16) to identify the linear combination of RFFT correlators which give rise to a bona fide CFT 4-point correlator. For example, we have

$$
\begin{aligned}
\langle \beta(x_1)\beta(x_2)\beta(x_3)\beta(x_4) \rangle = -3 & \overline{\langle \phi(x_1) \rangle \langle \phi(x_2) \rangle \langle \phi(x_3) \rangle \langle \phi(x_4) \rangle} + \overline{\langle \phi(x_1)\phi(x_2) \rangle \langle \phi(x_3) \rangle \langle \phi(x_4) \rangle} \\
+ & \overline{\langle \phi(x_2) \rangle \langle \phi(x_1)\phi(x_3) \rangle \langle \phi(x_4) \rangle} + \overline{\langle \phi(x_1) \rangle \langle \phi(x_2)\phi(x_3) \rangle \langle \phi(x_4) \rangle} - \overline{\langle \phi(x_1)\phi(x_2)\phi(x_3) \rangle \langle \phi(x_4) \rangle} \\
+ & \overline{\langle \phi(x_2) \rangle \langle \phi(x_3) \rangle \langle \phi(x_1)\phi(x_4) \rangle} + \overline{\langle \phi(x_1) \rangle \langle \phi(x_3) \rangle \langle \phi(x_2)\phi(x_4) \rangle} - \overline{\langle \phi(x_3) \rangle \langle \phi(x_1)\phi(x_2)\phi(x_4) \rangle} \\
+ & \overline{\langle \phi(x_1) \rangle \langle \phi(x_2) \rangle \langle \phi(x_3)\phi(x_4) \rangle} - \overline{\langle \phi(x_2) \rangle \langle \phi(x_1)\phi(x_3)\phi(x_4) \rangle} - \overline{\langle \phi(x_1) \rangle \langle \phi(x_2)\phi(x_3)\phi(x_4) \rangle} \\
+ & \overline{\langle \phi(x_1)\phi(x_2)\phi(x_3)\phi(x_4) \rangle} .
\end{aligned}
\tag{3.25}
$$

Specific classes of averaged correlators map to a single product of correlators in the $(\alpha, \beta)$ theory. For example,

$$
\begin{aligned}
& \overline{\langle \phi(x_1)\phi(x_2) \rangle_{\mathrm{c}} \cdots \langle \phi(x_{2n-1})\phi(x_{2n}) \rangle_{\mathrm{c}} \langle \phi(x_{2n+1}) \rangle \cdots \langle \phi(x_{2n+k}) \rangle} \\
& = \langle \beta(x_1)\beta(x_2) \rangle \cdots \langle \beta(x_{2n-1})\beta(x_{2n}) \rangle \langle \alpha(x_{2n+1}) \cdots \alpha(x_{2n+k}) \rangle .
\end{aligned}
\tag{3.26}
$$

**OPE**  The $(\alpha, \beta)$ theory can also be used to provide an intuition on how the OPE works in a disordered theory. In particular, it allows us to easily prove (2.25) and (2.26). In the presence of averages of products of correlators, the OPE can be taken either between two operators in the same correlator, as usual, or between different correlators. For concreteness, we focus on an example for each of the two cases, starting with OPE between operators in the same correlator. We want to determine the most singular part of the correlator

$$\overline{\langle \phi(x)\phi(0)A(\phi)\rangle}\,, \tag{3.27}$$

where $A(\phi)$ schematically denotes products of local operators with support outside a neighbourhood of $x = 0$. In the $(\alpha, \beta)$ description this is

$$\overline{\langle \phi(x)\phi(0)A(\phi)\rangle} = \langle (\alpha(x) + \beta(x))(\alpha(0) + \beta(0))A(\alpha + \beta)\rangle\,. \tag{3.28}$$

The singular contributions as $|x| \to 0$ arise from the identity exchange in the $\alpha \times \alpha$ and $\beta \times \beta$ OPEs:

$$\overline{\langle \phi(x)\phi(0)A(\phi)\rangle} = \frac{\kappa}{|x|^{d-2}} \langle A(\beta + \alpha)\rangle + \frac{\tilde{\kappa}}{|x|^{d-4}} \langle A(\beta + \alpha)\rangle + \text{ reg.}. \tag{3.29}$$

Using (3.20) and going back to the random field notation we simply have

$$
\begin{aligned}
\overline{\langle \phi(x)\phi(0)A(\phi)\rangle} &= \overline{\langle (\phi(x) - \langle\phi(x)\rangle)(\phi(0) - \langle\phi(0)\rangle)\rangle}\,\overline{\langle A(\phi)\rangle} + \overline{\langle\phi(x)\rangle\,\langle\phi(0)\rangle}\,\overline{\langle A(\phi)\rangle} + \text{ reg.}\\
&= \overline{\langle\phi(x)\phi(0)\rangle}\,\overline{\langle A(\phi)\rangle} + \text{ reg.} \qquad \text{as} \qquad |x| \to 0\,.
\end{aligned} \tag{3.30}
$$

Consider now the case where the two fields lie in distinct correlators:

$$\overline{\langle \phi(x)A_1(\phi)\rangle\,\langle\phi(0)A_2(\phi)\rangle}\,, \tag{3.31}$$

where again $A_1(\phi)$ and $A_2(\phi)$ denote products of local operators with support outside a neighbourhood of $x = 0$. In the $(\alpha, \beta)$ theory we make use of (3.14) to write

$$
\begin{aligned}
&\overline{\langle \phi(x)A_1(\phi)\rangle\,\langle\phi(0)A_2(\phi)\rangle}\\
&= \sum_{m,n\geq 0} \frac{1}{m!n!} \int \langle \alpha(x)\alpha(0)\alpha(y_1)\cdots\alpha(y_m)\alpha(z_1)\cdots\alpha(z_n)\rangle \Big\langle A_1^{(m)}[\beta](y_i)\Big\rangle\Big\langle A_2^{(n)}[\beta](z_i)\Big\rangle\\
&+ \sum_{m,n\geq 0} \frac{1}{m!n!} \int \langle \alpha(0)\alpha(y_1)\cdots\alpha(y_m)\alpha(z_1)\cdots\alpha(z_n)\rangle \Big\langle \beta(x)A_1^{(m)}[\beta](y_i)\Big\rangle\Big\langle A_2^{(n)}[\beta](z_i)\Big\rangle\\
&+ \sum_{m,n\geq 0} \frac{1}{m!n!} \int \langle \alpha(x)\alpha(y_1)\cdots\alpha(y_m)\alpha(z_1)\cdots\alpha(z_n)\rangle \Big\langle A_1^{(m)}[\beta](y_i)\Big\rangle\Big\langle \beta(0)A_2^{(n)}[\beta](z_i)\Big\rangle\\
&+ \sum_{m,n\geq 0} \frac{1}{m!n!} \int \langle \alpha(y_1)\cdots\alpha(y_m)\alpha(z_1)\cdots\alpha(z_n)\rangle \Big\langle \beta(x)A_1^{(m)}[\beta](y_i)\Big\rangle\Big\langle \beta(0)A_2^{(n)}[\beta](z_i)\Big\rangle\,.
\end{aligned} \tag{3.32}
$$

A singular contribution in the $|x| \to 0$ limit can only arise from the first line in (3.32) due to the identity exchange in the $\alpha \times \alpha$ OPE. Note that this time there is no contribution for the $\beta \times \beta$

OPE because they lie in different correlators. Hence

$$\overline{\langle\phi(x)A_1(\phi)\rangle\,\langle\phi(0)A_2(\phi)\rangle}$$
$$= \frac{v^2}{|x|^{d-4}}\sum_{m,n\geq 0}\frac{1}{m!n!}\int\langle\alpha(y_1)\cdots\alpha(y_m)\alpha(z_1)\cdots\alpha(z_n)\rangle\Big\langle A_1^{(m)}[\beta](y_i)\Big\rangle\Big\langle A_2^{(n)}[\beta](z_i)\Big\rangle \ + \ \text{reg.}$$
$$= \overline{\langle\phi(x)\rangle\,\langle\phi(0)\rangle}\,\overline{\langle A_1(\phi)\rangle\,\langle A_2(\phi)\rangle} \ + \ \text{reg.} \qquad \text{as} \qquad |x|\to 0\,. \tag{3.33}$$

This analysis can be generalized to more general averages of multiple correlators, resulting eventually in (2.25) and (2.26). More general OPEs can also be analyzed. The upshot of the analysis is that we can use the ordinary OPE in the $(\alpha,\beta)$ theory to argue how the OPE works in an arbitrary average product of correlators in the RFFT.

**Normal ordering**  We now turn to composite operators. First, we show how to define more general classes of normal ordered operators in RFFT, then we show that these normal ordered RFFT correlators are mapped to normal ordered correlators in the $(\alpha,\beta)$ theory.

Before considering the RFFT case let us review the pure case. For a pure free scalar field $\phi$ with a two-point function $G$, a proper formal operatorial definition of $:\phi^n:$ can be given using smooth test functions $f$ in terms of $\phi(f)=\int\mathrm{d}^d y\,f(y)\phi(y)$. One has (see e.g. eq. (6.3.9) of [23])

$$:\phi^n(f){:}= c^{n/2}P_n(c^{-1/2}\phi(f))\,, \tag{3.34}$$

where

$$c = G(f,f) = \int\mathrm{d}^d x\,\mathrm{d}^d y\,G(x,y)f(x)f(y)\,, \tag{3.35}$$

and

$$P_n(x) = (-1)^n e^{x^2/2}\frac{\mathrm{d}^n}{\mathrm{d}x^n}e^{-x^2/2} \tag{3.36}$$

are Hermite polynomials. We will use in what follows the more heuristic notation $:\phi^n(x):$ (used in most QFT textbooks), which corresponds to (3.34) if we formally take $f = \delta^{(d)}(x-y)$. For simplicity, we focus on scalar composite operators but a similar analysis can be repeated for spinning composite operators.

We now turn to the RFFT case and generalize (2.23). First we notice that (2.23) coincides with (3.34) for $n=2$, provided one replaces the formally divergent constant $c$ with its average analogue

$$c_1 = \overline{\langle\phi(x)^2\rangle}\,. \tag{3.37}$$

Then one can check that this generalizes for any $n$ as follows

$$:\phi^n(x){:}= c_1^{n/2}P_n\Big(c_1^{-1/2}\phi(x)\Big)\,. \tag{3.38}$$

In order to generalize the peculiar normal ordering (2.24) it is convenient to introduce the notation $\phi(x)_i\phi(x)_j$ which defines a composite operator where each field is in a different correlator: one $\phi$ appears inside $A_i$, while the other in $A_j$ in the notation of (3.9). In this way (2.24)

can be compactly rewritten as $:\phi(x)_1\phi(x)_2:= \phi(x)_1\phi(x)_2 - \overline{\langle\phi(x)\rangle^2}$. Higher dimensional composite operators can have more exotic forms as we can insert pieces of the operators in different correlators. For example one can consider all operators of the form

$$\prod_{i=1}^{n} \phi^{q_i}(x)_i\,, \tag{3.39}$$

namely all possible combinations of composite operators $\phi_i^{q_i}$ entering the correlator $i$ in the quenched average. One can check that the normal ordering of the class of operators (3.39) with $q_i = 1$, when $A_i(\phi) = 1$, can be written in closed form and is similar to the pure case (3.34) and to (3.38):

$$:\langle\phi(x)\rangle^n:= c_2^{n/2}P_n\left(c_2^{-1/2}\langle\phi(x)\rangle\right)\,, \qquad c_2 = \overline{\langle\phi(x)\rangle^2}\,. \tag{3.40}$$

We did not attempt to find a general form for the normal ordered version of all operators (3.39). We report for illustration a couple of slightly more complicated examples

$$\overline{:\langle\phi(x)A(\phi)\rangle\langle\phi(x)\rangle^2:} = \overline{\langle\phi(x)A(\phi)\rangle\langle\phi(x)\rangle^2} - 2c_2\overline{\langle A(\phi)\rangle\langle\phi(x)\rangle} - c_2\overline{\langle\phi(x)A(\phi)\rangle\langle\phi(x)\rangle}$$

$$\overline{:\langle\phi^2(x)A(\phi)\rangle\langle\phi^2(x)\rangle:} = \overline{\langle\phi^2(x)A(\phi)\rangle\langle\phi^2(x)\rangle} - c_1\overline{\langle A(\phi)\rangle\langle\phi^2(x)\rangle} - c_1\overline{\langle\phi^2(x)A(\phi)\rangle} \tag{3.41}$$

$$- 4c_2\overline{\langle\phi(x)A(\phi)\rangle\langle\phi(x)\rangle} + c_1^2\overline{\langle A(\phi)\rangle} + 2c_2^2\overline{\langle A(\phi)\rangle}\,.$$

Having exemplified some composite operators in the RFFT, we now want to show that the normal ordering of composite operators in the $(\alpha, \beta)$ theory as given by (3.34) is enough to properly regularize coincident point singularities occurring in the RFFT. As an example we report the expression of the correlators (2.21) and (2.22) when mapped to the $(\alpha, \beta)$ theory

$$\overline{\langle:\phi^2(x_1):\phi(x_2)\phi(x_3)\rangle} = 2\langle\beta(x_1)\beta(x_2)\rangle\langle\alpha(x_1)\alpha(x_3)\rangle + 2\langle\alpha(x_1)\alpha(x_2)\rangle\langle\beta(x_1)\beta(x_3)\rangle$$

$$+ \langle:\beta^2(x_1):\beta(x_2)\beta(x_3)\rangle + \langle:\alpha^2(x_1):\alpha(x_2)\alpha(x_3)\rangle\,, \tag{3.42}$$

$$\overline{:\langle\phi(x_1)\rangle\langle\phi(x_1):\phi(x_2)\phi(x_3)\rangle} = \langle:\alpha^2(x_1):\alpha(x_2)\alpha(x_3)\rangle + \langle\alpha(x_1)\alpha(x_2)\rangle\langle\beta(x_1)\beta(x_3)\rangle$$

$$+ \langle\alpha(x_1)\alpha(x_3)\rangle\langle\beta(x_1)\beta(x_2)\rangle\,. \tag{3.43}$$

We see that the right-hand sides only contain composite operators which are normal ordered in the usual sense of (3.34). From (3.16) one can find another class of average correlators in the RFFT that maps uniquely to the $\alpha$-theory:

$$\overline{\prod_{k=1}^{m}:\langle\phi(x_k)\rangle^{n_k}:} = \left\langle\prod_{k=1}^{m}:\alpha^{n_k}(x_k):\right\rangle\,, \tag{3.44}$$

namely, all correlators which are averages of products of one-point functions $\langle\phi\rangle$ only. Note that we have in (3.44) $:\langle\phi(x_k)\rangle^{n_k}:$ and not $\langle:\phi(x_k)^{n_k}:\rangle$, which would lead to a correlator involving both $\alpha$ and $\beta$ fields. A class of RFFT correlators which maps uniquely to the $\beta$-theory are instead of the peculiar form

$$\overline{\left\langle\prod_{k=1}^{m}:\left(\phi(x_k) - \langle\phi(x_k)\rangle\right)^{n_k}:\right\rangle} = \left\langle\prod_{k=1}^{m}:\beta^{n_k}(x_k):\right\rangle\,. \tag{3.45}$$

Like for elementary operators, generic averages of correlators involving composite operators (or products thereof) do not define proper CFT correlators, but linear combinations of them. This in particular applies to the $\overline{\langle \phi^2 \phi^2 \rangle}$ two-point function:

$$\overline{\langle {:}\phi^2(x_1){:}\ {:}\phi^2(x_2){:}\rangle} = \langle {:}\alpha^2(x_1){:}\ {:}\alpha^2(x_2){:}\rangle + \langle {:}\beta^2(x_1){:}\ {:}\beta^2(x_2){:}\rangle + 4\langle \alpha(x_1)\alpha(x_2)\rangle \langle \beta(x_1)\beta(x_2)\rangle \,, \tag{3.46}$$

which reproduces the correlator (2.8) discussed previously. The upshot of our analysis, based on various examples considered, is that the ordinary normal ordering in the $(\alpha, \beta)$ theory regularizes the coincident point singularities of the RFFT, both the ones coming from ordinary composite operators and the ones coming from the more exotic operators (3.39). We conjecture that this applies generally to all composite operators of the RFFT.

### 3.3  Symmetries

The $(\alpha, \beta)$ theory is manifestly conformal invariant. It contains a conserved traceless symmetric stress tensor which takes the form (see e.g. appendix A of [21])

$$\begin{aligned} T_{\mu\nu} = {}&\frac{1}{2(d-1)}\Big( d\,\partial_{\{\mu}\beta\partial_{\nu\}}\beta - (d-2)\beta\partial_{\{\mu}\partial_{\nu\}}\beta \Big) \\ &- \frac{1}{2v^2(d-1)}\Big( 2(d+2)\partial_{\{\mu}\alpha\partial_{\nu\}}\Box\alpha - (d-4)\alpha\partial_{\{\mu}\partial_{\nu\}}\Box\alpha - 4\partial_\rho\alpha\partial^\rho\partial_{\{\mu}\partial_{\nu\}}\alpha \\ &\qquad\qquad + \frac{4d}{d-2}\partial_\rho\partial_{\{\mu}\alpha\partial^\rho\partial_{\nu\}}\alpha - \frac{d(d+2)}{d-2}\Box\alpha\partial_{\{\mu}\partial_{\nu\}}\alpha \Big)\,, \end{aligned} \tag{3.47}$$

where $\{\}$ denotes the traceless-symmetric part. From the scaling assignments (3.6) it is trivial to prove the transformations (2.12) introduced in section 2. Under a scaling $x \to \lambda x$, we have

$$\begin{aligned} \phi = \alpha + \beta \quad &\to \lambda^{-\frac{d-4}{2}}\alpha + \lambda^{-\frac{d-2}{2}}\beta = \lambda^{-\frac{d-4}{2}}(\phi - Gh) + \lambda^{-\frac{d-2}{2}}Gh\,, \\ h = K\alpha \quad &\to \lambda^{-\frac{d-4}{2}}\lambda^{-2}K\alpha = \lambda^{-\frac{d}{2}}h\,. \end{aligned} \tag{3.48}$$

Let us now turn to global symmetries. The RFFT has a manifest $\mathbb{Z}_2$ symmetry under which $\phi \to -\phi$, $h \to -h$. Since the $h$ distribution (1.6) is $\mathbb{Z}_2$ invariant, we also have that

$$\overline{\prod_{i=1}^{k} \langle A_i(-\phi)\rangle_h} = \overline{\prod_{i=1}^{k} \langle A_i(\phi)\rangle_h}\,, \tag{3.49}$$

which is interpreted as a restoration of the $\mathbb{Z}_2$ symmetry $\phi \to -\phi$ of the free theory before adding the random coupling. The $(\alpha, \beta)$ theory action (3.7), on the other hand, is manifestly invariant under *two* $\mathbb{Z}_2$ global symmetries[6]

$$\mathbb{Z}_2 \quad : \quad \begin{cases} \alpha \to -\alpha \\ \beta \to -\beta \end{cases} \,, \qquad \mathbb{Z}_2' \quad : \quad \begin{cases} \alpha \to \alpha \\ \beta \to -\beta \end{cases} \tag{3.50}$$

---

[6]In generalizations of random free theories with $N$ scalars $\phi_i$ with $O(N)$ global symmetry coupled to random couplings $\phi_i h_i$ with an $O(N)$-invariant Gaussian distribution, the non-manifest $\mathbb{Z}_2$ symmetry upgrades to a whole non-manifest $O(N)$ continuous symmetry as the $(\alpha, \beta)$ theory enjoys a $O(N)_\alpha \times O(N)_\beta$ global symmetry.

The first $\mathbb{Z}_2$ corresponds to the $\phi \to -\phi$, $h \to -h$ symmetry just described, while the second one is less manifest in the RFFT. In terms of $h$ and $\phi$ it reads

$$\mathbb{Z}_2' \quad : \quad \begin{cases} \phi \to 2\langle\phi\rangle_h - \phi \\ h \to h \end{cases} . \tag{3.51}$$

This is indeed a symmetry of the action (1.1). Recalling that $\langle\phi\rangle_h = Gh$ and $G = K^{-1}$, we have

$$S' = \frac{1}{2}(2Gh - \phi)K(2Gh - \phi) - h(2Gh - \phi) = \frac{1}{2}\phi K\phi - 2h\phi - 2hGh + 2hGh + h\phi = S . \tag{3.52}$$

Note that this is a symmetry at fixed $h$, and hence for each theory in the ensemble. Upon average, we get the identities

$$\overline{\prod_{i=1}^{k} \left\langle A_i \left( 2\langle\phi\rangle_h - \phi \right) \right\rangle_h} = \overline{\prod_{i=1}^{k} \langle A_i(\phi)\rangle_h} , \tag{3.53}$$

valid for arbitrary averages of products of correlators. At face value, the symmetry (3.51) is not manifest in the RFFT. For example, for the $\phi^2$ two-point function (3.53) implies that

$$\begin{aligned} &- 16\overline{\langle\phi(x_1)\rangle^2 \langle\phi(x_2)\rangle^2} + 4\overline{\langle\phi(x_1)^2\rangle \langle\phi(x_2)\rangle^2} + 16\overline{\langle\phi(x_1)\rangle \langle\phi(x_2)\rangle \langle\phi(x_1)\phi(x_2)\rangle} \\ &- 4\overline{\langle\phi(x_2)\rangle \langle\phi(x_1)^2\phi(x_2)\rangle} + 4\overline{\langle\phi(x_1)\rangle^2 \langle\phi(x_2)^2\rangle} - 4\overline{\langle\phi(x_1)\rangle \langle\phi(x_1)\phi(x_2)^2\rangle} = 0 , \end{aligned} \tag{3.54}$$

which is not immediately clear. When mapped to the $(\alpha, \beta)$ theory, the correlators in (3.54) become linear combinations of products of correlators of $\alpha$ and $\beta$ all containing an odd number of fields inside each correlator, manifestly vanishing one by one.

Similarly, one can start from the $(\alpha, \beta)$ theory to obtain non-trivial selection rules in the RFFT. The symmetries in (3.50) imply that correlators with an odd number of $\alpha$ or $\beta$ fields vanish. For example, one has

$$\begin{aligned} 0 = \langle\beta(x_1)\beta(x_2)\beta(x_3)\alpha(x_4)\rangle = {}& 2\overline{\langle\phi(x_1)\rangle \langle\phi(x_2)\rangle \langle\phi(x_3)\rangle \langle\phi(x_4)\rangle} - \overline{\langle\phi(x_1)\phi(x_2)\rangle \langle\phi(x_3)\rangle \langle\phi(x_4)\rangle} \\ &- \overline{\langle\phi(x_2)\rangle \langle\phi(x_1)\phi(x_3)\rangle \langle\phi(x_4)\rangle} - \overline{\langle\phi(x_1)\rangle \langle\phi(x_2)\phi(x_3)\rangle \langle\phi(x_4)\rangle} + \overline{\langle\phi(x_1)\phi(x_2)\phi(x_3)\rangle \langle\phi(x_4)\rangle} . \end{aligned}$$

The right-hand side is non manifestly vanishing. The appearance of relations among the correlators in the RFFT is not surprising, since averages of products of correlators represent a vastly over-complete set of correlators.

**Heisenberg-like symmetry** Interestingly enough, the $(\alpha, \beta)$ theory has an additional continuous exchange symmetry which does not commute with scaling symmetry. To exhibit its form, it is convenient to integrate-in an auxiliary field $\gamma$ and rewrite the action (3.7) as

$$S(\alpha, \beta, \gamma) = \int \mathrm{d}^d x \left( (\partial\alpha)(\partial\gamma) - \frac{v^2}{2}\gamma^2 + \frac{1}{2}(\partial\beta)^2 \right) . \tag{3.55}$$

The action (3.55) is invariant under the following one-parameter group of transformations

$$\begin{pmatrix} \alpha \\ \beta \\ \gamma \end{pmatrix} \to \mathbf{T}(a) \begin{pmatrix} \alpha \\ \beta \\ \gamma \end{pmatrix}, \qquad \mathbf{T}(a) = \begin{pmatrix} 1 & a & -\frac{1}{2}a^2 \\ 0 & 1 & -a \\ 0 & 0 & 1 \end{pmatrix} \quad (a \in \mathbb{R}). \tag{3.56}$$

The matrices $\mathbf{T}(a)$ form an abelian group

$$\mathbf{T}(a+b) = \mathbf{T}(a)\mathbf{T}(b), \qquad \mathbf{T}(-a) = \mathbf{T}(a)^{-1}, \tag{3.57}$$

which is a subgroup of the Heisenberg group. Its generator is given by

$$\mathbf{S} = \begin{pmatrix} 0 & 1 & 0 \\ 0 & 0 & -1 \\ 0 & 0 & 0 \end{pmatrix}, \tag{3.58}$$

it is nilpotent of degree three $\mathbf{S}^3 = 0$, and therefore

$$e^{a\mathbf{S}} = \mathbb{1} + a\mathbf{S} + \frac{a^2}{2}\mathbf{S}^2 = \mathbf{T}(a). \tag{3.59}$$

The infinitesimal form of the transformation with parameter $\epsilon$, with and without the auxiliary field $\gamma$, are respectively given by

$$\begin{cases} \delta\alpha = \epsilon\beta \\ \delta\beta = -\epsilon\gamma \\ \delta\gamma = 0 \end{cases} , \qquad \begin{cases} \delta\alpha = \epsilon\beta \\ \delta\beta = \dfrac{\epsilon}{v^2}\Box\alpha \end{cases} . \tag{3.60}$$

The Noether current associated with this symmetry is

$$J_\mu = \partial_\mu\beta\Box\alpha - \beta\partial_\mu\Box\alpha. \tag{3.61}$$

The Ward identities involving $J_\mu$ relate correlators in the $\alpha$ and $\beta$-theory. For example, the Ward identity related to $\langle \partial^\mu J_\mu \, \alpha \, \beta \rangle$ gives rise to the identity $\langle \beta\beta \rangle = \frac{1}{v^2}\langle \alpha\Box\alpha \rangle$. The resulting identities in higher correlators are clearly less trivial than in this example. In the RFFT the transformations (3.60) act on $\phi$ and $h$ as

$$\delta\phi = \epsilon\left(\phi - Gh - \frac{h}{v^2}\right), \qquad \delta h = \epsilon(K\phi - h), \tag{3.62}$$

and are a symmetry of the action (3.3). The current (3.61) is mapped to the operator

$$\widetilde{J}_\mu = J_\mu - \langle J_\mu \rangle_h , \qquad J_\mu = h\partial_\mu\phi - \phi\partial_\mu h. \tag{3.63}$$

The conserved current (3.63) has the same form as the ones of disordered symmetries discussed in [17], i.e. symmetries of the pure action $S_0$ which are broken by $h$ but reappear after the quenched average. However, the current (3.63) is not a conserved current of the free theory (1.5),

as can be seen by its explicit dependence on $h$, but a truly "emergent" symmetry induced by the disorder. The current (3.63) implements a set of Ward identities relating different average correlators, including also insertions of the disorder $h$. We will not discuss further the selection rules coming from such identities.

Note that the current (3.61) is not a primary operator, but a level-one descendant of the spin two primary

$$\mathcal{K}_{\mu\nu} = \partial_{\{\mu}\alpha\partial_{\nu\}}\beta - \frac{d-4}{2d}\alpha\partial_{\{\mu}\partial_{\nu\}}\beta - \frac{1}{2}\beta\partial_{\{\mu}\partial_{\nu\}}\alpha\,. \qquad (3.64)$$

Indeed one can check that $\partial^\mu\mathcal{K}_{\mu\nu} \propto J_\nu$. The operator $\mathcal{K}_{\mu\nu}$ is a spin-two primary with dimension $d-1$, which therefore satisfies a higher order "conservation equation" $\partial^\mu\partial^\nu\mathcal{K}_{\mu\nu} = 0$.[7] Starting from $\mathcal{K}_{\mu\nu}$, we can construct other $d+1$ Noether currents by combining the primary $\mathcal{K}_{\mu\nu}$ with appropriate Killing scalars satisfying $\partial_\mu\partial_\nu\epsilon(x) = 0$ (see e.g. [21, 25] for generalizations). The currents are obtained as

$$J_\mu^t = \epsilon^t\partial_\nu\mathcal{K}_{\mu\nu} - \mathcal{K}_{\mu\nu}\partial_\nu\epsilon^t\,, \qquad (3.65)$$

where $\epsilon^t(x)$ are the three types $t = 1, 2, 3$ of Killing scalars given by $\epsilon^1(x) = \epsilon$, $\epsilon^2(x) = \epsilon^\mu x_\mu$, $\epsilon^3(x) = \tilde{\epsilon}x^2$, with constant $\epsilon, \epsilon_\mu, \tilde{\epsilon}$ (for a total of $d+2$ currents). The current $J_\mu^1$ coincides with (3.61). The other $d+1$ currents define $d+1$ continuous symmetries which give rise to the following infinitesimal transformations on the fields:

$$\begin{cases} \delta\alpha = -\frac{v^2}{2}\epsilon_\mu x^\mu\beta \\ \delta\beta = \frac{v^2}{2}\epsilon_\mu x^\mu\gamma + \epsilon_\mu\partial^\mu\alpha \\ \delta\gamma = \epsilon_\mu\partial^\mu\beta \end{cases}, \qquad \begin{cases} \delta\alpha = -\tilde{\epsilon}\frac{v^2}{2}x^2\beta \\ \delta\beta = 2\tilde{\epsilon}\big(x\cdot\partial + \frac{d}{2} - 2\big)\alpha + \frac{v^2}{2}\tilde{\epsilon}x^2\gamma \\ \delta\gamma = 2\tilde{\epsilon}\big(x\cdot\partial + \frac{d}{2}\big)\beta \end{cases}. \qquad (3.66)$$

It is not difficult to show that also the generator of the infinitesimal scalar transformations shown in the right of (3.66) is nilpotent of degree three.

**Higher-spin symmetries**  The $\beta$-theory is an ordinary unitary free scalar theory, it satisfies the equation of motions $\Box\beta = 0$, and it has the usual higher-spin conserved currents

$$\mathcal{I}_{\mu_1\cdots\mu_\ell} = \beta\partial_{\{\mu_1}\cdots\partial_{\mu_\ell\}}\beta + \cdots\,, \qquad \partial^{\mu_1}\mathcal{I}_{\mu_1\cdots\mu_\ell} = 0\,, \quad \Delta_\mathcal{I} = d + \ell - 2\,, \quad (\ell \geq 2 \text{ even})\,, \quad (3.67)$$

where $\{\}$ denotes the traceless symmetric part and "$\cdots$" denotes additional terms necessary to have a primary operator. The $\alpha$-theory is a more peculiar non-unitary $\Box^2$-theory where the field $\alpha$ satisfies the equation of motion $\Box^2\alpha = 0$. As discussed in [21], there are two families of spinning operators that are conserved when contracted with one or three derivatives:

$$\mathcal{J}^{(0)}_{\mu_1\cdots\mu_\ell} = \alpha\partial_{\{\mu_1}\cdots\partial_{\mu_\ell\}}\alpha + \cdots\,, \qquad \partial^{\mu_1}\partial^{\mu_2}\partial^{\mu_3}\mathcal{J}^{(0)}_{\mu_1\mu_2\mu_3\cdots\mu_\ell} = 0\,, \quad \Delta_{\mathcal{J}^{(0)}} = d + \ell - 4\,, \quad (\ell \geq 4 \text{ even})\,,$$
$$\mathcal{J}^{(1)}_{\mu_1\cdots\mu_\ell} = \alpha\partial_{\{\mu_1}\cdots\partial_{\mu_\ell\}}\Box\alpha + \cdots\,, \qquad \partial^{\mu_1}\mathcal{J}^{(1)}_{\mu_1\cdots\mu_\ell} = 0\,, \quad \Delta_{\mathcal{J}^{(1)}} = d + \ell - 2\,, \quad (\ell \geq 2 \text{ even})\,.$$
$$\qquad (3.68)$$

---

[7] According to the classification of [24], $\mathcal{K}_{\mu\nu}$ is a type II operator with $n = 2$. It is the first of the class of higher-spin conserved currents (3.69) discussed below. This has a degenerate multiplet because its level-two descendant $\partial^\mu\partial^\nu\mathcal{K}_{\mu\nu}$ is a primary.

We also have conserved currents which are obtained by combining $\alpha$ and $\beta$:

$$\mathcal{K}_{\mu_1\cdots\mu_\ell} = \beta\partial_{\{\mu_1}\cdots\partial_{\mu_\ell\}}\alpha + \cdots, \quad \partial^{\mu_1}\partial^{\mu_2}\mathcal{K}_{\mu_1\cdots\mu_\ell} = 0, \quad \Delta_\mathcal{K} = d + \ell - 3, \quad (\ell \geq 2). \quad (3.69)$$

The current $\mathcal{K}_{\mu\nu}$ in (3.64) is the explicit form of the operator written in (3.69) for $\ell = 2$, while the stress tensor is given by the sum of $\mathcal{I}$ and $\mathcal{J}^{(1)}$ with $\ell = 2$ as in (3.47). From each of the higher-spin conserved currents, one can build several Noether currents (associated with conserved charges), as we exemplified for $\mathcal{K}_{\mu\nu}$ in (3.65) and as it notably occurs for the energy-momentum tensor. We will not discuss the nature of the algebra defined by this extended set of currents and the form of selection rules they give rise to.

In appendix A we study in detail the spectrum of the $(\alpha, \beta)$ theory by explicitly decomposing the partition function in conformal characters in $d = 5$ and $d = 6$. The structure of higher-spin conserved currents is confirmed by tables 1 and 2, where we find the first operators of all the towers of currents listed in (3.67), (3.68), and (3.69) as primary operators (blue rows).

## 4  CFT description II: Cardy theory

The second description of the RFFT is more standard and is based on the replica trick and then on a change of basis due to Cardy [4], explaining its name. This approach has been recently discussed in [13, 14, 18], but no detailed analysis of the RFFT has been provided, the main focus being instead on the RF Ising model and the fate of the so-called Parisi-Sourlas supersymmetry [3].

We start in section 4.1 by briefly recalling the replica trick and the Cardy basis in the context of the free field theory. We show in section 4.2 how average correlators in the RFFT are mapped to this theory, as well as various examples of the map. We discuss in section 4.3 the symmetries of the theory. Further details are found in appendix B.

### 4.1  Derivation of the Cardy theory

The standard way to discuss quenched average correlation functions in a disorder QFT is given by the replica trick. In the schematic notation introduced in section 2, let $\overline{\langle A(\phi)\rangle_h}$ be the average of a generic correlation function of local operators. We can insert an arbitrary number of $\langle \mathbb{1}\rangle_h = Z[h]/Z[h] = 1$ in the average and write

$$\overline{\langle A(\phi)\rangle_h} = \overline{\langle A(\phi)\rangle_h \,(\langle\mathbb{1}\rangle_h)^{n-1}} = \int[\mathrm{d}h]\frac{P[h]}{Z[h]}\int[\mathrm{d}\phi]e^{-S_0(\phi)+h\phi}A(\phi)\left(\frac{1}{Z[h]}\int[\mathrm{d}\phi]e^{-S_0(\phi)+h\phi}\right)^{n-1}. \tag{4.1}$$

We combine the integrals by assigning a label, obtaining

$$\overline{\langle A(\phi)\rangle_h} = \int[\mathrm{d}h]P[h]\frac{1}{Z[h]^n}\int[\mathrm{d}\phi_1]\cdots[\mathrm{d}\phi_n]e^{-\sum_{a=1}^n S_0(\phi_a)+h\sum_{a=1}^n \phi_a}A(\phi_1). \tag{4.2}$$

Since the LHS is independent of $n$ we can take the $n \to 0$ limit. By assuming that we can exchange the limit and the integral in $[\mathrm{d}h]$ we have $Z[h]^n \to 1$.[8] If the random coupling distribution $P[h]$ is Gaussian as in (1.6), we can perform the $h$ integral:

$$\overline{\langle A(\phi) \rangle_h} = \lim_{n \to 0} \int [\mathrm{d}\phi_1] \cdots [\mathrm{d}\phi_n] e^{-S_{\mathrm{rep}}} A(\phi_1) \,, \tag{4.3}$$

where

$$S_{\mathrm{rep}} = \sum_{a=1}^{n} S_0(\phi_a) - \frac{v^2}{2} \int \mathrm{d}^d x \left( \sum_{a=1}^{n} \phi_a(x) \right)^2 . \tag{4.4}$$

A similar trick can be performed for more generic observables

$$\overline{\langle A_1(\phi) \rangle_h \cdots \langle A_k(\phi) \rangle_h} = \lim_{n \to 0} \int \prod_{a=1}^{n} [\mathrm{d}\phi_a] \, e^{-S_{\mathrm{rep}}} A_1(\phi_1) \cdots A_k(\phi_k) \,. \tag{4.5}$$

This construction allows us to recast average correlators (and average products thereof) in terms of a pure theory with replica action $S_{\mathrm{rep}}$ for generic $n$. Assuming analyticity in $n$, which in a Gaussian theory is satisfied, the original correlator is obtained by taking the $n \to 0$ limit. The replica action in general enjoys a $\mathcal{S}_n$ permutation symmetry that permutes the various copies of the replica theory. In the specific case where the action $S_0$ is the free scalar theory, the first term in (4.4) is invariant under a continuous $O(n)$ symmetry, broken to $O(n-1)$ by the $(\sum_a \phi_a)^2$ term. Hence the replica symmetry $\mathcal{S}_n$ is enhanced to $O(n-1)$.

Due to the replica symmetry, the propagator decomposes as

$$\langle \phi_a(x) \phi_b(0) \rangle_{\mathrm{rep}} = \delta_{ab} G_1 + G_2 \,, \tag{4.6}$$

where

$$G_1 = \int \frac{\mathrm{d}^d p}{(2\pi)^d} \frac{1}{p^2} e^{ip \cdot x} \,, \qquad G_2 = \int \frac{\mathrm{d}^d p}{(2\pi)^d} \frac{v^2}{p^2(p^2 - nv^2)} e^{ip \cdot x} \,. \tag{4.7}$$

We have

$$\langle \phi_1(x) \phi_1(0) \rangle_{\mathrm{rep}} = G_1 + G_2 \,, \qquad \langle \phi_1(x) \phi_2(0) \rangle_{\mathrm{rep}} = G_2 \,. \tag{4.8}$$

Using the map (4.5) between the replica and the disordered theory, we see that (3.20) is reproduced since $G_1 = G_\beta$ and $G_2 \to G_\alpha$ as $n \to 0$. Note that the theory has very different features when $n \neq 0$ since the propagator $G_2$ has a massive pole proportional to $n$. This mass makes it puzzling to identify the conformal symmetry of the replica theory, which reflects the peculiar features of the RFFT discussed in section 2. It turns out that we can make more manifest many properties of the $n \to 0$ limit by a change of basis in the replica fields, at the expense, however,

---

[8]One typically assumes that the $n \to 0$ limit can be applied first to $Z[h]^n$ and then to the rest. In the free case, this assumption is not needed, since we can compute $Z[h]$ exactly, see (2.9). The result amounts effectively to change the disorder distribution $\exp(-\frac{h^2}{2v^2})$ to $\exp(-\frac{1}{2}h(\frac{1}{v^2} + nG)h)$. This changes the form of the replica action (4.11) below by introducing some $n$-dependent non-locality but does not alter the Cardy action (4.12). We drop these contributions in what follows to keep the rest of the section more readable.

of losing manifest replica symmetry. The new basis consists in $n+1$ fields $\varphi, \omega, \chi_i$, $i = 2, \ldots, n$, subject to a constraint. The change of basis reads [4]:

$$\phi_1 = \varphi + \frac{1}{2}\omega \,, \qquad \phi_i = \varphi - \frac{1}{2}\omega + \chi_i \,, \qquad \sum_{i=2}^{n} \chi_i = 0 \,. \tag{4.9}$$

Its inverse is

$$\varphi = \frac{1}{2}\phi_1 + \frac{1}{2(n-1)}\rho \,, \qquad \omega = \phi_1 - \frac{1}{n-1}\rho \,, \qquad \chi_i = \phi_i - \frac{1}{n-1}\rho \,, \qquad \rho = \sum_{i=2}^{n} \phi_i \,. \tag{4.10}$$

The replica action in this basis reads

$$S_{\mathrm{rep}} = S_{\mathrm{Cardy}} + \mathcal{O}(n) \,, \tag{4.11}$$

where

$$S_{\mathrm{Cardy}}(\varphi, \chi, \omega) = \int \mathrm{d}^d x \left( (\partial\varphi)(\partial\omega) - \frac{v^2}{2}\omega^2 + \frac{1}{2}\sum_{i=2}^{n}(\partial\chi_i)^2 \right) \,. \tag{4.12}$$

The manifest symmetry is reduced to $O(n-2)$, the symmetry rotating the fields $\chi_i$. We will later discuss the fate of the missing generators of the original $O(n-1)$ symmetry. The advantage of this basis appears if we naively take the limit $n \to 0$ directly at the action level. In that case the fields $\varphi$, $\chi_i$ and $\omega$ acquire well-defined scaling dimensions, namely

$$\Delta_\varphi = \frac{d-4}{2} \,, \qquad \Delta_\chi = \frac{d-2}{2} \,, \qquad \Delta_\omega = \frac{d}{2} \,. \tag{4.13}$$

This naive procedure of taking the limit $n = 0$ partially, i.e. neglecting the $\mathcal{O}(n)$ terms in (4.12) but keeping all the replica fields $\varphi, \chi_i, \omega$, compute the correlators, and then taking once again the limit $n \to 0$, simply works.

The equation of motion of $\omega$, derived from $S_{\mathrm{Cardy}}$, indicates that $\omega$ is a level-two descendant of $\varphi$. Integrating it out fixes

$$\omega = -\frac{1}{v^2}\Box\varphi \,, \tag{4.14}$$

and leads to

$$S_{\mathrm{Cardy}}(\varphi, \chi) = \int \mathrm{d}^d x \left( \frac{1}{2v^2}(\Box\varphi)^2 + \frac{1}{2}\sum_{i=2}^{n}(\partial\chi_i)^2 \right) \,, \qquad \sum_{i=2}^{n} \chi_i = 0 \,. \tag{4.15}$$

The non-vanishing two-point functions are given by

$$\langle\varphi(x_1)\varphi(x_2)\rangle = \frac{\tilde{\kappa}}{x_{12}^{d-4}} \,, \qquad \langle\chi_i(x_1)\chi_j(x_2)\rangle = G_{ij}(n)\frac{\kappa}{x_{12}^{d-2}} \,, \qquad G_{ij}(n) = \delta_{ij} - \frac{1}{n-1} \,, \tag{4.16}$$

with $\tilde{\kappa}$ and $\kappa$ as in (2.2) and (2.4). The constraint on the $\chi_i$'s is satisfied since $\sum_{i=2}^{n} G_{ij}(n) = 0$.[9] We can also choose a basis for $\chi_i$ that diagonalizes the matrix $G(n)$. This matrix has two

---

[9]The matrix $G(n)$ also satisfies the following properties $G(n)^T = G(n)$, $G(n)^2 = G(n)$ and $\mathrm{tr}[G(n)] = n - 2$, which are useful in computations.

eigenspaces: one of dimension 1 with eigenvalue 0, the second of dimension $n-2$ with eigenvalue 1. The eigenspace of dimension 1 is associated with the singlet $\sum_{i=2}^{n} \chi_i$ which is set to zero by the constraint. The second eigenspace is spanned by the eigenvectors

$$\zeta_k = \frac{1}{\sqrt{k(k-1)}} \left( -\chi_2 - \sum_{j=1}^{k-2} \chi_{n-j+1} + (k-1)\chi_{n-k+2} \right), \tag{4.17}$$

for $k = 2, \ldots, n-1$. In these variables, we get yet another action for the Cardy theory in terms of unconstrained fields:

$$S_{\mathrm{Cardy}}(\varphi, \zeta) = \int \mathrm{d}^d x \left( \frac{1}{2v^2}(\Box\varphi)^2 + \frac{1}{2}\sum_{i=2}^{n-1}(\partial\zeta_i)^2 \right). \tag{4.18}$$

The action (4.18) is the direct sum of a $\Box^2$ free scalar CFT, the same CFT entering in the $(\alpha, \beta)$ description, and the CFT given by $n-2$ free scalars. While the limit of $n \to 0$ may seem unconventional, it is clear that the $n-2$ free fields $\zeta_k$ for any integer $n > 2$ give rise to a well-defined free CFT. The formulation (4.18) makes clear that the limit $n \to 0$ of the CFT data is consistent and can be given a proper mathematical meaning by the same categorical description used in [19] to define $O(N)$ models with $N \in \mathbb{R}$. We will however not use a categorical description, but rather the descriptions (4.15) or (4.18) in terms of "index-ful" fields $\chi_i$ and $\zeta_i$, which we find more convenient in computations. Putting all the steps together, we get the final form of the map between RFFT correlators and the Cardy description:

$$\overline{\prod_{p=1}^{k} \langle A_p(\phi)\rangle_h} = \lim_{n \to 0} \int [\mathrm{d}\varphi] \prod_{i=2}^{n} [\mathrm{d}\chi_i]\, \delta\left(\sum_{i=2}^{n}\chi_i\right) e^{-S_{\mathrm{Cardy}}(\varphi, \chi)} \prod_{p=1}^{k} A_p\Big(\phi_p(\varphi, \chi)\Big), \tag{4.19}$$

where $S_{\mathrm{Cardy}}$ is the action (4.15) and the map between the fields $\phi_a$ and $(\varphi, \chi)$ is given by (4.9) with $\omega = -\Box\varphi/v^2$. It is important to emphasize that the map (4.19) is *not* unique as a consequence of the loss of the manifest $\mathcal{S}_n$ replica symmetry in the Cardy basis. If we reshuffle the assignments of replica fields to the various factors $\langle A_p(\phi)\rangle_h$, the Cardy correlator will correspondingly change. For this reason, and to get rid from the beginning of the field $\omega$, it is practically more convenient to start from Cardy correlators in terms of the fields $\varphi$ and $\chi_i$ and use (4.19) in its inverse form to read the resulting RFFT correlators. We will discuss various examples and properties of correlator maps in the next subsection and in appendix B.

We will denote in what follows by "Cardy theory" the theory obtained using the action (4.15) and by "replica theory" the theory defined by (4.11), including the $\mathcal{O}(n)$ terms. Care should be paid in distinguishing the two descriptions when working with $n \neq 0$.

## 4.2 Correlator mapping and normal ordering

The simplest example is given by the two-point functions of fundamental Cardy fields. For the two-point function of $\varphi$, the result is

$$
\begin{aligned}
\langle \varphi(x_1)\varphi(x_2)\rangle &= \lim_{n\to 0} \frac{1}{4}\Big(\langle \phi_1(x_1)\phi_1(x_2)\rangle - \langle \phi_1(x_1)\rho(x_2)\rangle - \langle \rho(x_1)\phi_1(x_2)\rangle + \langle \rho(x_1)\rho(x_2)\rangle\Big)\\
&= \overline{\langle \phi(x_1)\rangle\,\langle \phi(x_2)\rangle}\,,
\end{aligned}
\tag{4.20}
$$

where in the first equality we used (4.10) and we recall that $\rho = \sum_{i=2}^{n}\phi_i$. For the second equality we used the fact that $\langle \phi_i\phi_j\rangle = \overline{\langle \phi\rangle\,\langle \phi\rangle}$ for any $i\neq j$ and $\langle \phi_i\phi_i\rangle = \overline{\langle \phi\phi\rangle}$ for any $i$. So, we reproduce the fact that the disconnected correlator $\overline{\langle \phi\rangle\,\langle \phi\rangle}$ qualifies as a good, independent, two-point correlator for a primary operator with scaling dimension $\Delta = (d-4)/2$.

We now consider $\langle \chi_i\chi_j\rangle$,

$$
\begin{aligned}
\langle \chi_i(x_1)\chi_j(x_2)\rangle &= \lim_{n\to 0}\Big(\langle \phi_i(x_1)\phi_j(x_2)\rangle + \langle \phi_i(x_1)\rho(x_2)\rangle + \langle \rho(x_1)\phi_j(x_2)\rangle + \langle \rho(x_1)\rho(x_2)\rangle\Big)\\
&= G_{ij}\Big(\overline{\langle \phi(x_1)\phi(x_2)\rangle} - \overline{\langle \phi(x_1)\rangle\,\langle \phi(x_2)\rangle}\Big)\,,
\end{aligned}
\tag{4.21}
$$

where

$$
G_{ij} \equiv G_{ij}(n=0) = \delta_{ij} + 1\,,
\tag{4.22}
$$

is the tensor structure of the two-point function in (4.16) evaluated at $n = 0$. As expected, the connected correlator qualifies as a good, independent, two-point correlator for a primary operator with scaling dimension $\Delta = (d-2)/2$.

Let us show how to apply this map to four-point functions. If we consider four fundamental operators, the possible 15 observables in the RF side are the ones defined in (2.17). We want to find one set of 15 Cardy correlators that generates all RF correlators. A set that works is the following

$$
\begin{aligned}
V_{\mathrm{Ca}} = \Big\{ &\langle \varphi\varphi\varphi\varphi\rangle\,, \langle \varphi\varphi\chi_2\chi_3\rangle\,, \langle \varphi\chi_2\varphi\chi_3\rangle\,, \langle \chi_2\varphi\varphi\chi_3\rangle\,, \langle \varphi\chi_2\chi_3\varphi\rangle\,, \langle \chi_2\varphi\chi_3\varphi\rangle\,, \langle \chi_2\chi_3\varphi\varphi\rangle\,,\\
&\langle \varphi\chi_2\chi_2\chi_3\rangle\,, \langle \chi_2\varphi\chi_2\chi_3\rangle\,, \langle \chi_2\chi_2\varphi\chi_3\rangle\,, \langle \chi_2\chi_2\chi_3\varphi\rangle\,,\\
&\langle \chi_2\chi_2\chi_3\chi_4\rangle\,, \langle \chi_2\chi_2\chi_3\chi_3\rangle\,, \langle \chi_2\chi_3\chi_2\chi_3\rangle\,, \langle \chi_2\chi_3\chi_4\chi_5\rangle\Big\}\,,
\end{aligned}
\tag{4.23}
$$

where we used the convention that the $i$-th operator in the correlators is inserted at $x_i$ (we will use this same notation below). By mapping $V_{\mathrm{Ca}}$ to RF variables we find that

$$
V_{\mathrm{Ca}} = \left(M_{\mathrm{Ca}}^{(4)}\right)^{-1} V_{\mathrm{RF}}\,,
\tag{4.24}
$$

where $M_{\mathrm{Ca}}^{(4)}$ is reported in (B.10). The matrix (B.10) is invertible so that we can reconstruct all possible RF correlators $V_{\mathrm{RF}}$ from the ones defined in Cardy theory. Notice that the set in $V_{\mathrm{Ca}}$ is not unique but all other choices are related by symmetries.

Before concluding this section let us show a couple of examples of inverse maps as we did in (3.20) and (3.21) for the $(\alpha, \beta)$ theory. As explained in appendix B.3, these can be compactly written as

$$\overline{\langle \phi\phi\phi\phi \rangle} = \langle \varphi\varphi\varphi\varphi \rangle + (\langle \varphi\varphi\chi_2\chi_3 \rangle + 5 \text{ perms}) + \langle \chi_2\chi_3\chi_4\chi_5 \rangle \,,$$

$$\overline{\langle \phi\phi \rangle \langle \phi\phi \rangle} = \langle \varphi\varphi\varphi\varphi \rangle + \langle \varphi\chi_2\chi_3\varphi \rangle + \langle \chi_2\varphi\varphi\chi_3 \rangle \quad (4.25)$$

$$- \langle \chi_2\chi_2\chi_3\chi_4 \rangle - \frac{1}{3} \langle \chi_2\chi_3\chi_2\chi_3 \rangle + \frac{7}{3} \langle \chi_2\chi_3\chi_4\chi_5 \rangle \,.$$

We stress again that a single average (of products) of RF correlators generically corresponds to a linear combination of Cardy correlators. In the Cardy formulation, in contrast to the $(\alpha, \beta)$ description, no product of correlators is present on the right-hand side of the equations.

**Normal ordering**   Normal-ordered correlators can be also considered in the Cardy theory. The situation here is very similar to what was already said for the $(\alpha, \beta)$ description, so we will be brief (more details are also presented in appendix B.1 and B.2). The upshot is that the set of normal ordered correlators in the Cardy theory describes the set of correlators of composite operators of the RFFT when regularized to subtract the coincident-point singularities precisely in the way discussed in section 2 and 3.2. As an example, let us consider the normal ordered correlators (2.21) and (2.22) mapped to Cardy variables. We have

$$\overline{\langle {:}\phi(x_1)^2{:}\ \phi(x_2)\phi(x_3) \rangle} = \langle {:}\varphi^2{:}\ \varphi\,\varphi \rangle + 2 \langle {:}\varphi\chi_2{:}\ \varphi\,\chi_3 \rangle + 2 \langle {:}\varphi\chi_2{:}\ \chi_3\,\varphi \rangle + \langle {:}\chi_2\chi_3{:}\ \chi_4\,\chi_5 \rangle \,, \quad (4.26)$$

$$\overline{{:}\langle \phi(x_1) \rangle\ \langle \phi(x_1){:}\ \phi(x_2)\phi(x_3) \rangle} = \langle {:}\varphi^2{:}\ \varphi\,\varphi \rangle + \langle {:}\varphi\chi_2{:}\ \varphi\,\chi_3 \rangle + \langle {:}\varphi\chi_2{:}\ \chi_3\,\varphi \rangle \,, \quad (4.27)$$

as shown in appendix B.2. Both (4.26) and (4.27) are written in terms of three-point functions of normal-ordered Cardy fields, which ensures that the expressions are finite.

The class of correlators in the LHS of (3.44) admits a simple mapping also in the Cardy theory:

$$\overline{\prod_{k=1}^m {:}\langle \phi(x_k) \rangle^{n_k}{:}} = \left\langle \prod_{k=1}^m {:}\varphi(x_k)^{n_k}{:} \right\rangle . \quad (4.28)$$

Similarly, the correlators (3.45) can be mapped to the Cardy theory as follows

$$\overline{\left\langle \prod_{k=1}^m {:}\Big(\phi(x_k) - \langle \phi(x_k) \rangle \Big)^{n_k}{:} \right\rangle} = \left\langle \prod_{k=1}^m {:}\chi_k^{n_k}(x_k){:} \right\rangle . \quad (4.29)$$

Note that if we simply map the RHS of (4.28) and (4.29) to RFFT variables, we typically find complicated linear combinations of RF observables. However, because of selection rules, most of the terms vanish leaving the results above.

## 4.3   Symmetries

### 4.3.1   Symmetries of the Cardy theory

The symmetries of the Cardy theory are more easily discussed in the formulation with the variables $\zeta_i$. This has two main advantages. First, in terms of $\zeta_i$, the theory looks more similar

to the $(\alpha, \beta)$ theory which we have already discussed. As we will see, several conserved currents will have a similar structure provided one replaces $(\alpha, \beta) \to (\varphi, \zeta_i)$ in the appropriate manner. Second, the formulation in terms of $\zeta_i$ can more easily be rewritten using categories. However, we do not use the categorical language for clarity of the exposition. We will then loosely talk about representations of $O(-2)$ as if it was $O(N)$ for $N$ integer (analytically continued to $N \to -2$).

Let us start by writing the complete set of infinite conserved currents as we did for the $(\alpha, \beta)$ theory. We will then discuss some interesting examples.

The currents of the $\varphi$ sector are identical to those in (3.68) with $\alpha \to \varphi$:

$$\mathcal{J}^{(0)}_{\mu_1 \cdots \mu_\ell} = \varphi \partial_{\{\mu_1} \cdots \partial_{\mu_\ell\}} \varphi + \cdots , \qquad \partial^{\mu_1} \partial^{\mu_2} \partial^{\mu_3} \mathcal{J}^{(0)}_{\mu_1 \mu_2 \mu_3 \cdots \mu_\ell} = 0 , \quad \Delta_{\mathcal{J}^{(0)}} = d + \ell - 4 , \quad (\ell \geq 4 \text{ even}) ,$$
$$\mathcal{J}^{(1)}_{\mu_1 \cdots \mu_\ell} = \varphi \partial_{\{\mu_1} \cdots \partial_{\mu_\ell\}} \Box \varphi + \cdots , \qquad \partial^{\mu_1} \mathcal{J}^{(1)}_{\mu_1 \cdots \mu_\ell} = 0 , \quad \Delta_{\mathcal{J}^{(1)}} = d + \ell - 2 , \quad (\ell \geq 2 \text{ even}) .$$
$$(4.30)$$

In the $\zeta_i$ sector, we have the well-known currents of a free $O(N)$ model (see e.g. the review of [26]) in the limit of $N \to -2$. In particular (3.67) are replaced by currents valued in the singlet $\mathbf{S}$, antisymmetric $\mathbf{A}$ and traceless-symmetric $\mathbf{T}$ representations of $O(-2)$:[10]

$$\mathcal{I}^{(\mathbf{r})}_{kl,\mu_1 \cdots \mu_\ell} = \zeta_i \partial_{\{\mu_1} \cdots \partial_{\mu_\ell\}} \zeta_j \, [P_{\mathbf{r}}]^{ij}_{kl} + \cdots , \quad \partial^{\mu_1} \mathcal{I}^{(\mathbf{r})}_{kl,\mu_1 \cdots \mu_\ell} = 0 , \quad \Delta_{\mathcal{I}} = d + \ell - 2 , \quad (\ell \geq 1) . \quad (4.31)$$

where $[P_{\mathbf{r}}]^{ij}_{kl}$ are projector decomposing the tensor product of two vectors into a representation $\mathbf{r} = \mathbf{S}, \mathbf{A}, \mathbf{T}$ (with $\ell$ even for $\mathbf{S}, \mathbf{T}$ and $\ell$ odd for $\mathbf{A}$). Finally the analogues of (3.69) read

$$\mathcal{K}_{i,\mu_1 \cdots \mu_\ell} = \zeta_i \partial_{\{\mu_1} \cdots \partial_{\mu_\ell\}} \varphi + \cdots , \qquad \partial^{\mu_1} \partial^{\mu_2} \mathcal{K}_{i,\mu_1 \cdots \mu_\ell} = 0 , \quad \Delta_{\mathcal{K}} = d + \ell - 3 , \quad (\ell \geq 2) , \quad (4.32)$$

and transform in the vector representation of $O(-2)$.

Let us now exemplify the symmetries associated with some of these operators.

**Conformal symmetry** The stress tensor is built as a combination of the stress tensor of the $O(-2)$ theory $\mathcal{I}^{(\mathbf{S})}_{\mu\nu}$ with the one of the $\varphi$ theory $\mathcal{J}^{(1)}_{\mu\nu}$. As usual, this operator defines the Noether currents and the associated charges that generate the conformal group.

$O(-2)$ **symmetry** Another important operator is the spin-one $O(-2)$ current. This operator and its associated transformation take the canonical form

$$\mathcal{I}^{(\mathbf{A}),\mu}_{ij} = \zeta_{[i} \partial^\mu \zeta_{j]} , \qquad \qquad \delta \zeta_i = \epsilon_{ij} \zeta_j , \qquad \qquad (4.33)$$

where the parameter $\epsilon_{ij}$ is in the antisymmetric representation of $O(-2)$.

---

[10]In categorical language, these can be written following [19] as

$$\mathcal{I}^{(\mathbf{r})}_{\mu_1 \cdots \mu_\ell} = \zeta \partial_{\{\mu_1} \cdots \partial_{\mu_\ell\}} \zeta \circ P^{\mathbf{n,n}}_{\mathbf{r}} + \cdots$$

where now $\zeta$ is a field in the vector representation $\mathbf{n}$ of $O(-2)$ seen as an object in a Deligne category, and $P^{\mathbf{n,n}}_{\mathbf{r}}$ is the morphism embedding the $\mathbf{r} = \mathbf{S}, \mathbf{A}, \mathbf{T}$ representation into the tensor product $\mathbf{n} \otimes \mathbf{n}$.

**Heisenberg-like symmetry**   As discussed in the $(\alpha, \beta)$ theory, see (3.65), we can construct three types $t = 1, 2, 3$ of Noether currents $J_{i\mu}^t = \epsilon_i^t \partial_\nu \mathcal{K}_i^{\mu\nu} - \mathcal{K}_i^{\mu\nu} \partial_\nu \epsilon_i^t$, starting from the tensor $\mathcal{K}_i^{\mu\nu}$ in (4.32), where $\epsilon_i^1(x) = \epsilon_i, \epsilon_i^2(x) = \epsilon_i^\mu x_\mu, \epsilon_i^3(x) = \tilde{\epsilon}_i x^2$. They give rise to the following field transformations:

$$
\begin{cases}
\delta\varphi = \epsilon_i \zeta_i \\
\delta\zeta_i = -\epsilon_i \, \omega \\
\delta\omega = 0
\end{cases}
,
\begin{cases}
\delta\varphi = \epsilon_i^\mu \left( -\frac{v^2}{2} x_\mu \zeta_i \right) \\
\delta\zeta_i = \epsilon_i^\mu \left( \frac{v^2}{2} x_\mu \omega + \partial_\mu \varphi \right) \\
\delta\omega = \epsilon_i^\mu \partial_\mu \zeta_i
\end{cases}
,
\begin{cases}
\delta\varphi = \tilde{\epsilon}_i \left( -\frac{v^2}{2} x^2 \zeta_i \right) \\
\delta\zeta_i = 2\tilde{\epsilon}_i \left[ \left( x \cdot \partial + \frac{d}{2} - 2 \right) \varphi + \frac{v^2}{2} x^2 \omega \right] \\
\delta\omega = 2\tilde{\epsilon}_i \left( x \cdot \partial + \frac{d}{2} \right) \zeta_i
\end{cases}
. \quad (4.34)
$$

Here we reintroduced the Lagrange multiplier $\omega$ of (4.12) that plays the same role of $\gamma$ in the $(\alpha, \beta)$ theory and can be eliminated by the equations of motions (4.14). There are a total of $d + 2$ parameters, each transforming in the vector representation of $O(-2)$. We shall refer to the first set of symmetries parameterized by $\epsilon_i$ as the "Heisenberg-like symmetries" as we did for the $(\alpha, \beta)$ theory. It is easy to see that the transformations parameterized by $\epsilon_i$ and $\tilde{\epsilon}_i$ are nilpotent of degree three.

Since the Heisenberg-like symmetry is abelian and is valued in the vector representation of $O(-2)$, the full algebra formed by the transformation in (4.33) and the first transformation in (4.34) results in a $ISO(-2)$ algebra (as we will also see in section 4.3.4). Moreover, it is important to stress that while the $O(-2)$ part commutes with dilations, the Heisenberg-like part does not, and indeed the first transformation in (4.34) increases the conformal dimension of the fundamental fields by one unit.

In principle, one can build several Noether currents from each of the other primary operators. We will not discuss the nature of the algebra defined by such a set of currents.

**Spectrum**   The spectrum of the Cardy theory is the direct product of the one of a $\square^2$ theory and of a $O(-2)$ vector model, as follows from (4.18). Its explicit form could be derived with character techniques, though the computation is technically more challenging than the one in the $(\alpha, \beta)$ theory because of the $O(-2)$ sector, and we have not attempted to do it. A possible way to proceed is by reconstructing the $O(N)$ characters for generic positive integer $N$ from those of the permutation group $\mathcal{S}_N$, as discussed e.g. in appendix B.2 of [27], and then continue the result to $N = -2$. This could also be used to check the above structure of higher-spin conserved currents as we did for the $(\alpha, \beta)$ theory. We expect the presence of a complicated structure of multiplet recombination among the operators due to non-decoupling null states, as discussed in the context of the spectrum of the $(\alpha, \beta)$ theory. It would be interesting to address this point in the future.

### 4.3.2   Consequences in the RFFT

In this subsection, we want to exemplify how some of the previously defined symmetries can be used to obtain selection rules in the RFFT. To do so we will need to use the map (4.10) which replaces correlators of $\varphi, \chi_i, \omega$ with RFFT observables.

In principle, we could take all the results of the previous subsection and map them in terms of $\chi_i$, since they are related via the linear map (4.17). This becomes a bit cumbersome since the map is complicated. It is simpler to give directly a prescription to build irreducible representations in $\chi_i$ and use it to directly build the spectrum of the theory. Let us explain how this works. Besides the $n \to 0$ limit (which can be tackled using [19]), the main complication is that the $O(n-2)$ symmetry of the action (4.15) does not act conventionally because there are $n-1$ fields subject to the constraint $\sum_i \chi_i = 0$. However, this problem can be easily taken into account using the fact that $G_{ij}$ defined in (4.22) is the invariant tensor of $O(-2)$ when parametrized in terms of $\chi_i$.[11] Representations follow the usual classification for $O(N)$ models, just replacing the invariant tensor $\delta_{ij}$ by $G_{ij}$. As usual, we define the representations by applying projectors. For a rank-two tensor, the projectors are

$$[\Pi_{\mathbf{S}}]^{ij}_{kl} = -\frac{G_{ij}G_{kl}}{2}, \quad [\Pi_{\mathbf{A}}]^{ij}_{kl} = \frac{G_{ik}G_{jl} - G_{il}G_{jk}}{2}, \quad [\Pi_{\mathbf{T}}]^{ij}_{kl} = \frac{G_{ik}G_{jl} + G_{il}G_{jk}}{2} + \frac{G_{ij}G_{kl}}{2}.$$
(4.35)

All operators can then be constructed as usual. For example, the scalar quadratic operators made out of $\chi_i$ are constructed by contraction with the projectors above and result in

$$\mathcal{O} = {:}\chi_k\chi_k{:}, \qquad \mathcal{O}_{ij} = {:}\chi_i\chi_j + \frac{G_{ij}}{2}\chi_k\chi_k{:},$$
(4.36)

where we used that $G_{ij}\chi_j = \chi_i$ to simplify the expressions. Similarly, we can classify currents which look exactly as in section 4.3.1 with the only difference that $\zeta_i \to \chi_i$ and the projectors are built using $\Pi$ instead of $P$. As an example, the $O(-2)$ current is

$$\mathcal{I}^{(\mathbf{A}),\mu}_{ij} = \chi_{[i}\partial^\mu\chi_{j]}.$$
(4.37)

This takes the canonical form since $G_{ij}\chi_j = \chi_i$. The variation will also look similar,

$$\delta\chi_i = \hat{\epsilon}_{ij}\chi_j$$
(4.38)

but now crucially the parameter $\hat{\epsilon}_{ij}$ should be antisymmetrized using $\Pi_{\mathbf{A}}$, namely $\hat{\epsilon}_{ij} = [\Pi_{\mathbf{A}}]^{kl}_{ij}\epsilon_{kl}$. As a result of this construction, the parameter satisfies

$$\hat{\epsilon}_{ij} = -\hat{\epsilon}_{ij}, \qquad \sum_i \hat{\epsilon}_{ij} = 0.$$
(4.39)

The constraints on $\hat{\epsilon}_{ij}$ reduce the number of independent generators to the correct one. A similar story holds for all other symmetries, but let us give only the extra example of Heisenberg-like

---

[11] The $O(n-2)$ symmetry is realized by $(n-1)\times(n-1)$ linear transformations that preserve the vector $(1, 1, \ldots, 1)$ (up to a sign). Rotations act as $\chi_i \to R_{ij}\chi_j$ where $R$ satisfies both $R^T R = G(n)$ and $R^T G(n)R = G(n)$ with $G_{ij}(n)$ as in (4.16). The fact that $R^T G(n)R = G(n)$ implies that $G(n)$ is an invariant tensor. The action of rotations can be derived by using the map (4.17) to the standard $O(n-2)$ basis, which we write as $\zeta_i = A_{ij}\chi_j$. $A$ is a rectangular $(n-2, n-1)$ matrix satisfying $AA^T = \mathbb{1}$ and $A^T A = G(n)$ and $\sum_j A_{ij} = 0$. It is then easy to see that a rotation $\zeta_i \to Q_{ij}\zeta_j$ with $Q^T Q = \mathbb{1}$ is mapped to $\chi_i \to R_{ij}\chi_j$ where $R = A^T QA$. The constraints on $R$ are then a consequence of the properties of $A$ and the constraint $Q^T Q = \mathbb{1}$.

symmetries, which take the form

$$\begin{cases} \delta\varphi = \hat{\epsilon}_i \chi_i \,, \\ \delta\chi_i = -\hat{\epsilon}_i \, \omega \,, \\ \delta\omega = 0 \,, \end{cases} \tag{4.40}$$

where $\hat{\epsilon}_i = G_{ij}\epsilon_j$ projects to the correct vector representation and automatically satisfies the constraint $\sum_i \hat{\epsilon}_i = 0$. The transformation (4.40) is nilpotent of degree three, the finite transformation is obtained by exponentiation and is given by

$$\varphi \to \varphi + \hat{b}_i \chi_i - \frac{1}{2}\hat{b}_i^2 \omega \,, \qquad \chi_i \to \chi_i - \hat{b}_i \omega \,, \qquad \omega \to \omega \,, \tag{4.41}$$

where $\hat{b}_i = G_{ij}b_j$ and $b_j$ is a finite vector. Now that we have explained how to translate the representation theory and the symmetries to $\chi_i$ variables, let us give some examples of the selection rules that they imply in RFFT.

Let us exemplify the case of $O(-2)$ symmetry. We consider the generic two-point function of quadratic operators in $\chi_i$ which can be decomposed by $O(-2)$ symmetry as a sum of its singlet and traceless symmetric part,

$$\langle :\chi_i \chi_j: :\chi_k \chi_l: \rangle = \langle \mathcal{O}_{ij} \mathcal{O}_{kl} \rangle + \frac{1}{4}G_{ij}G_{kl}\langle \mathcal{O}\mathcal{O} \rangle \,. \tag{4.42}$$

By this symmetry, we find that varying $i, j, k, l$, all possible correlators should be expressed in terms of only two different quantities. In particular (4.42) implies

$$\langle :\chi_2 \chi_2: :\chi_3 \chi_3: \rangle - 3\langle :\chi_2 \chi_2: :\chi_3 \chi_4: \rangle + 2\langle :\chi_2 \chi_3: :\chi_4 \chi_5: \rangle = 0 \,. \tag{4.43}$$

As shown in appendix B.1, the condition (4.43) gives a non-trivial selection rule on the RF correlators which takes the form[12]

$$\begin{aligned} 6\overline{\langle \phi(x_1) \rangle^2 \langle \phi(x_2) \rangle^2} &- 2\overline{\langle \phi^2(x_1) \rangle \langle \phi(x_2) \rangle^2} - 8\overline{\langle \phi(x_1) \rangle \langle \phi(x_2) \rangle \langle \phi(x_1)\phi(x_2) \rangle} \\ &+ 2\overline{\langle \phi(x_1)\phi(x_2) \rangle^2} + 2\overline{\langle \phi(x_2) \rangle \langle \phi^2(x_1)\phi(x_2) \rangle} - 2\overline{\langle \phi(x_1) \rangle^2 \langle \phi^2(x_2) \rangle} \\ &+ \overline{\langle \phi^2(x_1) \rangle \langle \phi^2(x_2) \rangle} + 2\overline{\langle \phi(x_1) \rangle \langle \phi(x_1)\phi^2(x_2) \rangle} - \overline{\langle \phi^2(x_1)\phi^2(x_2) \rangle} = 0 \,. \end{aligned} \tag{4.44}$$

Similarly, the four-point function of $\chi_i$ can be decomposed according to $O(-2)$ symmetry in 3 tensor structures (associated to the exchange of operators in the **S**, **A** and **T** representations in any channel). Explicitly we have

$$\langle \chi_i \chi_j \chi_k \chi_l \rangle = \kappa^2 \left( \frac{G_{ij}G_{kl}}{x_{12}^{d-2}x_{34}^{d-2}} + \frac{G_{ik}G_{jl}}{x_{13}^{d-2}x_{24}^{d-2}} + \frac{G_{il}G_{jk}}{x_{14}^{d-2}x_{23}^{d-2}} \right) \,. \tag{4.45}$$

As we discuss in appendix B.3, it is necessary to include 4 of the $\chi_i$-correlators to reconstruct the whole $V_{\text{RF}}$ in (2.17). There must be then linear relations among the RFFT correlators in $V_{\text{RF}}$ because of the form (4.45), see appendix B.3 for further details.[13]

---

[12]This relation maps to $\langle :\beta^2(x_1): :\beta^2(x_2): \rangle - 2\langle \beta(x_1)\beta(x_2) \rangle^2 = 0$ in the $(\alpha, \beta)$ theory. It is interesting to notice that (4.44) is implied by Wick contraction in the $(\alpha, \beta)$ formulation, while in the Cardy formulation by rotational symmetry of the fields $\chi_i$.

[13]Since the theory is free, the three structures are actually all related to each other, leading to further constraints.

### 4.3.3 Parisi-Sourlas supersymmetry

It is known that the correlators of the Cardy theory (4.12) which enjoy an extra $O(-2)$ symmetry in the fields $\chi$, e.g. $(\sum_i \chi_i^2)^k$, can be described by a model containing 2 Grassmann-valued scalar fields $\psi$ and $\bar\psi$ whose action reads [4]:[14]

$$S_{\text{susy}} = \int d^d x \left( (\partial\varphi)(\partial\omega) - \frac{v^2}{2}\omega^2 + (\partial\psi)(\partial\bar\psi) \right).$$
(4.46)

Each of the two theories contains operators which are not present in the other theory. However, $O(-2)$ invariant operators $\chi$ turn into $Sp(2)$ invariant operators in terms of $\psi, \bar\psi$, e.g.

$$\sum_i \chi_i^2 \leftrightarrow 2\psi\bar\psi.$$
(4.47)

The operators which are different are the ones that are not $O(-2)$ invariant in terms of the fields $\chi_i$ and that are not $Sp(2)$ invariant with respect to $\psi, \bar\psi$. For example the field $\chi$ itself or $\sum_i \chi_i^k$ for $k > 2$ cannot be mapped to $\psi, \bar\psi$. Similarly, the field $\psi$ cannot be mapped to the variables $\chi$.

While this formulation does not capture the full RFFT, it has some nice features. First, it does not involve a negative number of fields, thus it is less subtle than the Cardy formulation. Secondly, the action (4.46) enjoys Parisi-Sourlas superconformal symmetry $OSp(d+1,1|2)$ [3,20] (in particular since it is a free theory it has a higher spin enhancement of this symmetry, which we will disregard in the following). Using the notation of [18] the generators of $OSp(d+1,1|2)$ are supertranslations $P^A$, superrotations $M^{AB}$, superdilation $D$ and special superconformal transformations $K^A$, where the indices take values $A, B = 1, \ldots, d, \theta, \bar\theta$. When $A, B = 1, \ldots, d$ the generators define the conformal algebra, the $Sp(2)$ $R$-symmetry is generated by $M^{\theta\theta}, M^{\theta\bar\theta}, M^{\bar\theta\bar\theta}$, while the other generators with a single $\theta$ or $\bar\theta$ are fermionic. The fermionic transformation $P^a$, $M^{a\mu}$, $K^a$ with $a = \theta, \bar\theta$ are respectively given by

$$\begin{cases} \delta\varphi = \psi_a \epsilon^a \\ \delta\psi^a = -\omega\epsilon^a \\ \delta\omega = 0 \end{cases}, \quad \begin{cases} \delta\varphi = \left(-\frac{v^2}{2}x^\mu\psi_a\right)\epsilon_\mu^a \\ \delta\psi^a = \left(\frac{v^2}{2}x^\mu\omega + \partial^\mu\varphi\right)\epsilon_\mu^a \\ \delta\omega = \partial^\mu\psi_a\epsilon_\mu^a \end{cases}, \quad \begin{cases} \delta\varphi = \left(-\frac{v^2}{2}x^2\psi_a\right)\tilde\epsilon^a \\ \delta\psi^a = 2\left[\left(x\cdot\partial + \frac{d}{2} - 2\right)\varphi + \frac{v^2}{2}x^2\omega\right]\tilde\epsilon^a \\ \delta\omega = 2\left(x\cdot\partial + \frac{d}{2}\right)\psi_a\tilde\epsilon^a \end{cases}, \quad (4.48)$$

where all the parameters $\epsilon^a = \{\epsilon^\theta, \epsilon^{\bar\theta}\}$, $\epsilon_\mu^a = \{\epsilon_\mu^\theta, \epsilon_\mu^{\bar\theta}\}$, $\tilde\epsilon^a = \{\tilde\epsilon^\theta, \tilde\epsilon^{\bar\theta}\}$ are femionic, $\psi^a = \{\psi, \bar\psi\}$ and $\psi_a = \{-\bar\psi, \psi\}$. These transformations bear striking resemblances to (4.34) if we think of $\psi^a \leftrightarrow \zeta_i$ (or similarly $\psi^a \leftrightarrow \chi_i$ in (4.40)). The number of parameters also formally matches if we count the $-1$ bosonic parameter as 1 fermionic parameter. However, due to the different statistics and number of fields involved, it is not trivial to find a precise map between the two transformations. Let us also mention that the $O(-2)$ symmetry is related to the $R$-symmetry $Sp(2)$. Here there is also a formal match of the parameters since the dimension of the antisymmetric representation of $-2$ components is equal to 3. So roughly speaking we can map all generators of $OSp(d+1,1|2)$

---

[14]Intuitively, this can be seen from the fact that $-2$ scalars have formally the same partition function as $+2$ Grassmann scalars. The $O(-2)$ symmetry of the fields $\chi$ is converted to a $Sp(2)$ symmetry of $\psi, \bar\psi$.

to generators of the Cardy theory, in terms of the Cardy stress-tensor, the operator $\mathcal{K}^{\mu\nu}$ and the $O(-2)$ current.

It is instructive to show explicitly how correlators that involve the fermionic fields are mapped to the original RF variables. At first sight, this might look puzzling since the RFFT only has bosonic fields. However, we should remember that only $Sp(2)$ singlets have good representatives in the RFFT and these contain combinations of the type $\psi\bar{\psi}$ which behave like bosons. The simplest example is the two-point function of $\langle :\psi\bar{\psi}(x)::\psi\bar{\psi}(y):\rangle$ which can be rewritten in terms of $\chi_i$ using (4.47). The correlator in terms of $\chi_i$ can be easily expressed in RF variables using (B.3) and (4.42). The result is

$$
\begin{aligned}
\langle :\psi\bar{\psi}(x)::\psi\bar{\psi}(y):\rangle = {} & 6\overline{\langle\phi(x)\rangle^2\langle\phi(y)\rangle^2} - 3\overline{\langle\phi(x)^2\rangle\langle\phi(y)\rangle^2} - 6\overline{\langle\phi(x)\rangle\langle\phi(y)\rangle\langle\phi(x)\phi(y)\rangle} \\
& + \overline{\langle\phi(x)\phi(y)\rangle^2} - \overline{\langle\phi(x)^2\phi(y)^2\rangle} + 2\overline{\langle\phi(y)\rangle\langle\phi(x)^2\phi(y)\rangle} \\
& - 3\overline{\langle\phi(x)\rangle^2\langle\phi(y)^2\rangle} + 2\overline{\langle\phi(x)^2\rangle\langle\phi(y)^2\rangle} + 2\overline{\langle\phi(x)\rangle\langle\phi(x)\phi(y)^2\rangle} \\
& + \overline{\langle\phi(x)^2\rangle}\Big(\overline{\langle\phi(y)\rangle^2} - \overline{\langle\phi(y)^2\rangle}\Big) + \overline{\langle\phi(x)\rangle^2}\Big(\overline{\langle\phi(y)^2\rangle} - \overline{\langle\phi(y)\rangle^2}\Big) .
\end{aligned}
\tag{4.49}
$$

### 4.3.4 Symmetries of the replica theory

In contrast to the Cardy theory (4.15), where conformal invariance is obvious, for any $n \neq 0$ the replica action (4.11) is not conformally invariant. More precisely, by looking at the $\mathcal{O}(n)$ and $\mathcal{O}(n^2)$ terms in the action (4.11), it is straightforward to show that the scaling assignments (4.13) do not admit a smooth local deformation away from $n = 0$ which keeps $S_{\text{rep}}$ scale invariant. This is a reflection of the nature of the $n \to 0$ limit pointed out before. In other words, the "categorical" description of the Cardy theory (4.15) for generic $n$ has a manifest conformal symmetry for any $n$, while the finite $n$ replica action (4.11) does not.

$O(n-1)$ **symmetry**    As already discussed, the replica action (4.11) enjoys an $O(n-1)$ symmetry. This symmetry is not manifest in the Cardy basis and it would naively seem that in the limit $n \to 0$ this would amount to an $O(-1)$ symmetry as opposed to the $ISO(-2)$ symmetry discussed in section 4.3.1. Similarly, the $\mathcal{S}_n$ subgroup of $O(n-1)$, which plays an important role being preserved by interactions in the replica formulation, is also not manifest in the Cardy basis so one might wonder if this symmetry can be recovered in terms of $ISO(-2)$ transformations.

Let us first recover the $O(n-1)$ symmetry on Cardy fields by working at finite $n$. Namely, we consider the action (4.11) without dropping $\mathcal{O}(n)$ terms and look for its symmetries. We use the unconstrained fields $\zeta_k$ where symmetry transformations are most easily written. The action reads

$$
\begin{aligned}
S_{\text{rep}} = {} & \frac{1}{2}\varphi\big(nK - n^2 v^2\big)\varphi + \frac{1}{2}\omega\left(\frac{n}{4}K - \frac{(n-2)^2}{4}v^2\right)\omega + \varphi\left(-\frac{n-2}{2}K + \frac{n(n-2)}{2}v^2\right)\omega \\
& + \frac{1}{2}\sum_{i=2}^{n-1}\zeta_k K \zeta_k .
\end{aligned}
\tag{4.50}
$$

This is a quadratic form $\frac{1}{2}\vec{u}^T M \vec{u}$ where $\vec{u} = (\varphi, \zeta_2, \dots, \zeta_{n-1}, \omega)^T$. Linear transformations that keep the action invariant are generated by matrices $\Omega$ ($\delta\vec{u} = \Omega\vec{u}$) such that $\Omega^T M + M\Omega = 0$. The constraint is solved by the generators of $O(n-2)$ acting on the vector $\vec{\zeta} = (\zeta_2, \dots, \zeta_{n-1})$, namely

$$(\Omega_{ij})_{ab} = \delta_{ai}\delta_{bj} - \delta_{aj}\delta_{bi}, \qquad i,j = 2,\dots,n-1, \qquad (4.51)$$

where $a,b = 1,\dots,n$ label the components of $\vec{u}$, and by the following set of $(n-2)$ matrices

$$(\Omega_i)_{ab} = \delta_{a1}\delta_{bi} - \frac{2(n-1)}{n-2}\delta_{ai}\delta_{bn} + \frac{2n}{n-2}\delta_{an}\delta_{bi}, \qquad k = 2,\dots,n-1. \qquad (4.52)$$

To be more explicit, these infinitesimal transformations on the Cardy fields act as

$$\delta_i\varphi = \zeta_i, \qquad \delta_i\zeta_j = -\frac{2(n-1)}{n-2}\delta_{ij}\omega, \qquad \delta_i\omega = \frac{2n}{n-2}\zeta_i. \qquad (4.53)$$

Despite the unusual form of the $\Omega_i$ generators, they can be put together with the $\Omega_{ij}$ generators to reconstruct the manifest $O(n-1)$ symmetry of the replica action (4.4). Indeed, it is easy to compute the following commutators

$$[\Omega_{ij}, \Omega_k] = \delta_{jk}\Omega_i - \delta_{ik}\Omega_j,$$
$$[\Omega_i, \Omega_j] = -\frac{2n(n-1)}{(n-2)^2}\Omega_{ij}, \qquad (4.54)$$

which imply that the generators $L_{ab}$ defined as

$$L_{1i} = -L_{i1} = \sqrt{-\frac{(n-2)^2}{2n(n-1)}}\Omega_i, \qquad L_{ij} = \Omega_{ij}, \qquad (4.55)$$

form an $\mathfrak{so}(n-1)$ algebra.

However, the $O(n-1)$ generators in (4.55) do not have a smooth $n \to 0$ limit. Indeed, as can be seen in (4.54), the algebra of the generators $\{\Omega_i, \Omega_{ij}\}$ acting on the Cardy fields undergoes a form of Inonu-Wigner contraction as $n \to 0$. This is quite non-standard since the parameter $n$ controlling the algebra's contraction also changes its dimension. Nevertheless, this "formal" contraction of the $O(n-1)$ algebra to an $ISO(-2)$ algebra, nicely accounts for the symmetries of the Cardy theory described in section 4.3.1. The $\Omega_{ij}$ generators have a "manifest" limit to $O(-2)$ generators as $n \to 0$, while the transformations (4.53) reduce to the first of the Heisenberg-like transformation given in (4.34).

$\mathcal{S}_n$ **symmetry**    Given the previous discussion, it is clear that we should be able to map permutations in $\mathcal{S}_n$, seen as a subgroup of $O(n-1)$, to limits of $ISO(n-2)$ transformations as $n \to 0$. When $i, j > 1$ the permutation $\phi_i \leftrightarrow \phi_j$ is just mapped to a permutation $\chi_i \leftrightarrow \chi_j$. This is simply a $O(n-2)$ transformation acting on $\chi_i$, which of course commutes with dilations. Taking e.g. $i = 1$ and $j = 2$ the map $\phi_i \leftrightarrow \phi_j$ becomes [13]

$$\varphi \xrightarrow{P_{12}} \varphi + \chi_2 - \omega, \qquad \omega \xrightarrow{P_{12}} \omega, \qquad \chi_2 \xrightarrow{P_{12}} 2\omega - \chi_2, \qquad \chi_{k\geq 3} \xrightarrow{P_{12}} \chi_k - \chi_2 + \omega, \qquad (4.56)$$

where we have dropped $\mathcal{O}(n)$ terms. Note that the map does not commute with dilation since it mixes fields with different scaling dimensions in the Cardy theory.

The transformation (4.56) can be written as a composition of a $O(n-2)$ rotation with a Heisenberg-like transformation, both in the limit $n \to 0$. The Heisenberg-like transformation is (4.41) with parameters $\hat{b}_2 = (n-2) \to -2$ and $\hat{b}_k = -1$ for $k = 3, \ldots, n$. This corresponds to the transformation $T$

$$\varphi \xrightarrow{T} \varphi - \chi_2 - \omega\,, \quad \chi_2 \xrightarrow{T} \chi_2 + 2\omega\,, \quad \chi_k \xrightarrow{T} \chi_k + \omega\,, \quad \omega \xrightarrow{T} \omega\,. \tag{4.57}$$

The discrete $O(n-2)$ rotation will be named $R$ and takes the following form[15]

$$\chi_2 \xrightarrow{R} -\chi_2\,, \qquad \chi_k \xrightarrow{R} \chi_k - \chi_2\,. \tag{4.58}$$

By composing these two transformations it is easy to show that

$$P_{12} = R\,T\,. \tag{4.59}$$

Since $T$ is a Heisenberg-like transformation, $T = e^S = 1 + S + \frac{1}{2}S^2$, we can also write

$$P_{12} = R\left(1 + S + \frac{S^2}{2}\right), \tag{4.60}$$

where $S$ defines the infinitesimal transformations

$$\delta_S \varphi = -\chi_2\,, \quad \delta_S \chi_2 = 2\omega\,, \quad \delta_S \chi_k = \omega\,, \quad \delta_S \omega = 0\,. \tag{4.61}$$

## 5 Random field in generalized free theory

The two CFT descriptions of the RFFT can be easily extended to the case where the action $S_0$ in (1.5) is replaced by a generalized free theory (GFT) defined in terms of a generalized free field (GFF) $\phi$ of dimension $\Delta$. GFTs can be seen as decoupled sectors of a local CFT at leading order in a $1/N$ expansion.[16] They are dual to a free scalar of mass $L^2 m^2 = \Delta(\Delta - d)$ in a rigid $\text{AdS}_{d+1}$ space with radius $L$. GFTs in the presence of a random field were already considered in [22], where the flow induced by $\phi^2$ deformations was studied. We reconsider this setup in light of the CFT descriptions introduced in the previous sections.

Let us first settle some notation. A GFF $\phi$ of dimension $\Delta$ gives rise to a CFT which can be defined through the two-point function

$$\langle \phi(x)\phi(y)\rangle = G(x,y) = \frac{1}{|x-y|^{2\Delta}}\,, \tag{5.1}$$

---

[15]This is a rotation in the sense explained in footnote 11. Namely, it is the $n \to 0$ limit of

$$\chi_2 \xrightarrow{R} -\chi_2\,, \qquad \chi_k \xrightarrow{R} \chi_k + \frac{2}{n-2}\chi_2\,,$$

for which $Q = ARA^T$ satisfies $Q^T Q = \mathbb{1}$.

[16]GFTs also appear as the continuum limit of long-range lattice models. See e.g. [28–31] for studies of long-range lattice models with random field interactions.

and by the prescription of computing higher correlators by the Wick theorem. This is equivalent to using a path integral with a pure action of the form (2.9), where now $K = G^{-1}$ is the non-local kernel

$$K(x,y) = A_d(\Delta) \int \frac{\mathrm{d}^d p}{(2\pi)^d} \, p^{d-2\Delta} e^{ip\cdot(x-y)} \,, \qquad A_d(\Delta) = \frac{4^\Delta \Gamma(\Delta)}{(4\pi)^{d/2} \Gamma(d/2 - \Delta)} \,. \tag{5.2}$$

Demanding cluster decomposition and well-defined correlators requires $\Delta > d/4$. The theory in the presence of a random field, which we denote for brevity as random generalized free theory (RGFT), has correlation functions that are still defined through formulas (1.1), (1.2), (1.3) and $\mathcal{O}_0 = -\phi$.

We will also consider the effect of adding a local $\lambda\phi^2$ deformation. This is of interest because it allows us to study a non-trivial (though exactly solvable) RG flow in a disordered theory. Let us first review the effect of $\phi^2$ terms in the absence of disorder. The action has the usual form of (2.9) where now the kernel is

$$K(x,y) = \int \frac{\mathrm{d}^d p}{(2\pi)^d} \Big( A_d(\Delta) p^{d-2\Delta} + \lambda \Big) e^{ip\cdot(x-y)} \,, \tag{5.3}$$

with $A_d(\Delta)$ as in (5.2). Since correlation functions are given by Wick theorem, it is sufficient to consider the two-point function which is now given by

$$\langle \phi(x)\phi(y) \rangle = \int \frac{\mathrm{d}^d p}{(2\pi)^d} \frac{1}{A_d(\Delta) p^{d-2\Delta} + \lambda} e^{ip\cdot(x-y)} \,. \tag{5.4}$$

We focus on the case $\Delta < d/2$, where the deformation is relevant. In the UV limit, at distances $\tilde{\lambda}|x-y| \ll 1$, with $\tilde{\lambda} = \lambda^{1/(d-2\Delta)}$, the $p^{d-2\Delta}$ term dominates with respect to $\lambda$ in the denominator of (5.4) and we recover the undeformed correlator (5.1). The IR limit $\tilde{\lambda}|x-y| \gg 1$ is instead controlled by the opposite limit ($p \to 0$) where $\lambda$ dominates over $p^{d-2\Delta}$. In this limit, we obtain a GFF of dimension $d - \Delta$:[17]

$$\langle \phi(x)\phi(y) \rangle \simeq \int \frac{\mathrm{d}^d p}{(2\pi)^d} \Big( -\frac{A_d(\Delta)}{\lambda^2} p^{d-2\Delta} \Big) e^{ip\cdot(x-y)}$$
$$= -\frac{A_d(\Delta) A_d(d-\Delta)}{\lambda^2} \frac{1}{|x-y|^{2(d-\Delta)}}, \qquad \tilde{\lambda}|x-y| \gg 1 \,. \tag{5.6}$$

---

[17]For $\Delta = (d-1)/2$ the two-point function can be written in terms of Bessel $J$ and Struve $H$ functions

$$\langle \phi(x)\phi(y) \rangle = \frac{1}{|x-y|^{d-1}} \left[ 1 + \frac{\pi^{3/2}}{\Gamma\left(\frac{d-1}{2}\right)} \xi^{d/2} \left( \frac{H_{1-\frac{d}{2}}(\xi)}{\cos\left(\frac{\pi d}{2}\right)} + \frac{J_{1-\frac{d}{2}}(\xi)}{\sin\left(\frac{\pi d}{2}\right)} + \frac{2 J_{\frac{d}{2}-1}(\xi)}{\sin(\pi d)} \right) \right] \,, \tag{5.5}$$

where $\xi = \lambda|x-y|/A_d((d-1)/2)$. Using the small- and large-argument expansions of the above special functions one reproduces the UV and IR behaviours (5.1) and (5.6). The small momentum expansion of the integrand in (5.4) breaks down when $\Delta = (d-2)/2$, namely when $\phi$ is an ordinary free scalar and $\lambda\phi^2$ is a mass deformation $m = \sqrt{\lambda}$. In that case, as is well known, the two-point function is exponentially suppressed

$$\int \frac{\mathrm{d}^d p}{(2\pi)^d} \frac{1}{p^2 + m^2} e^{ip\cdot x} = \frac{1}{(2\pi)^{d/2}} \left( \frac{m}{|x|} \right)^{\frac{d-2}{2}} G_{\frac{d-2}{2}}(m|x|) \sim \frac{m^{\frac{d-3}{2}}}{2(2\pi)^{\frac{d-1}{2}}} \frac{e^{-m|x|}}{|x|^{\frac{d-1}{2}}} \,.$$

More generally, the small $p$ expansion is ill-defined when $\Delta = d/2 - 2k$, with $k \in \mathbb{N}$, where it would lead to an uncontrolled series of distributions $\int \frac{\mathrm{d}^d p}{(2\pi)^d} (p^2)^n e^{ip\cdot x} \sim \Box^n \delta^{(d)}(x)$.

Thus the $\phi^2$ deformation leads to the RG flow

$$\text{GFT}_\Delta \quad \xrightarrow{\lambda\phi^2} \quad \text{GFT}_{d-\Delta} \,, \tag{5.7}$$

in agreement with the flow induced by a double trace deformation in large $N$ theories [32, 33].

In the next sections, we discuss the RGFT, and the $\phi^2$ deformations, using the descriptions introduced in sections 3 and 4.

## 5.1 $(\alpha, \beta)$ theory

The derivation of the $(\alpha, \beta)$ theory in section 3.1 applies in this more general case since it relies only on the quadratic structure of the action. The ending result is that (3.14) holds, where the $\alpha$ and $\beta$ correlators are computed using the same action (3.5) and the same map (3.4), the only change being in the kernel $K$, which is now given by the non-local expression (5.2).

Two point functions of the $(\alpha, \beta)$ theory are explicitly given by

$$\langle\alpha(x)\alpha(x)\rangle = v^2 \,\frac{A_d\big(2\Delta - \frac{d}{2}\big)}{A_d(\Delta)^2}\,\frac{1}{|x-y|^{2\big(2\Delta-\frac{d}{2}\big)}}\,,$$
$$\langle\beta(x)\beta(x)\rangle = \frac{1}{|x-y|^{2\Delta}}\,. \tag{5.8}$$

Hence, the RGFT admits a description in terms of two decoupled GFFs of dimensions respectively $\Delta_\alpha = 2\Delta - \frac{d}{2}$ and $\Delta_\beta = \Delta$. Interestingly enough, for $\Delta \geq (d-1)/2$ both $\alpha$ and $\beta$ are above the unitarity bound and hence correlators in the $(\alpha, \beta)$ description are reflection positive. The condition $\Delta > d/4$ ensures that the $(\alpha, \beta)$ theory satisfies the cluster property; this replaces the constraint $d > 4$ derived in the RFFT, where $\Delta = (d-2)/2$.

The $(\alpha, \beta)$ theory is not very flexible to generic local deformations of the pure action because it requires computing exactly the $Z[h]$ factor. However, a $\phi^2$ deformation leaves the theory Gaussian and can be treated. In the presence of disorder, the derivation of the $(\alpha, \beta)$ theory in section 3.1 applies again. The two-point functions are now given by

$$\langle\alpha(x)\alpha(x)\rangle = \int \frac{\mathrm{d}^d p}{(2\pi)^d}\,\frac{v^2}{\big(A_d(\Delta)p^{d-2\Delta} + \lambda\big)^2}\,e^{ip\cdot(x-y)}\,,$$
$$\langle\beta(x)\beta(x)\rangle = \int \frac{\mathrm{d}^d p}{(2\pi)^d}\,\frac{1}{A_d(\Delta)p^{d-2\Delta} + \lambda}\,e^{ip\cdot(x-y)}\,. \tag{5.9}$$

For $\Delta < d/2$ the $\beta$ sector has exactly the same RG flow as (5.7). The UV behaviour of the $\alpha$ sector is given by a GFF with $\Delta_\alpha = 2\Delta - \frac{d}{2}$, while the IR one is given by a GFF with $\Delta_\alpha = d - \Delta$, as can be easily derived observing that $\langle\alpha\alpha\rangle = -v^2\frac{\partial}{\partial\lambda}\langle\beta\beta\rangle$.

Thus we conclude that[18]

$$\text{UV}: \text{RGFT}_\Delta + \lambda\phi^2 \longrightarrow \text{IR}: \text{GFT}_{d-\Delta} \times \text{GFT}_{d-\Delta}\,. \qquad (5.10)$$

## 5.2 Cardy theory

The Cardy description allows for any type of deformation, not only Gaussian ones. However since we studied other Gaussian theories in the $(\alpha, \beta)$ theory formulation, it is worth checking that also the Cardy framework would give compatible results.

First, let us define the RGFT in Cardy variables. We can follow the same steps for the derivation of the Cardy theory in section 4.1 but using now the kernel $K$ given in (5.2). The ending result is that (4.19) holds with the action given by

$$S_{\text{Cardy}} = \frac{1}{2v^2}\varphi K^2\varphi + \frac{1}{2}\sum_{i=2}^{n}\chi_i K\chi_i\,, \qquad (5.11)$$

which implies that the operators $\varphi$ and $\chi_i$ have respective dimensions $\Delta_\varphi = 2\Delta - \frac{d}{2}$ and $\Delta_\chi = \Delta$, while the auxiliary field $\omega = \frac{1}{v^2}K\varphi$ has dimension $\Delta_\omega = \frac{d}{2}$. This exactly matches the structure seen in the $(\alpha, \beta)$ theory, where again the only difference is that we now have $-2$ fields $\chi_i$ instead of only one field $\beta$.

One can also treat the $\lambda\phi^2$ deformation. This perturbation is mapped in Cardy variables to $\lambda\big(\varphi\omega + \frac{1}{2}\sum_{i=2}^{n}\chi_i^2\big)$. It is easy to see that this has the effect of changing the propagators of $\varphi$ and $\chi_i$ in the same way as we saw in (5.9). So again we find the same result with the exception that the $-2$ fields $\chi_i$ replace the single $\beta$.

Let us conclude by discussing what happens to RGFT after applying the map (4.47). It is easy to check that the resulting action does not respect Parisi-Sourlas supersymmetry. One might then wonder if there exists a random field version of GFF which when mapped to fermionic variables respects Parisi-Sourlas supersymmetry. To answer this question let us consider the more general model,

$$S[h] = \frac{1}{2}\phi K_1\phi - hK_2\phi\,, \qquad (5.12)$$

where both $K_1$ and $K_2$ are (possibly long range) kinetic terms of the form (5.2) which we respectively parametrize by $\Delta_1$ and $\Delta_2$. This action thus allows for a non-local coupling of the random field variable to the field $\phi$.[19] By applying Cardy construction to this action we find

$$S_{\text{Cardy}} = \varphi K_1\omega - \frac{v^2}{2}\omega(K_2)^2\omega + \frac{1}{2}\sum_{i=2}^{n}\chi_i K_1\chi_i\,, \qquad (5.13)$$

---

[18]The effect of a $\lambda\phi^2$ deformation in a GFF in the presence of disorder has previously been discussed in [22]. Although we agree that in the IR we have a conformal behaviour with $\tilde\Delta = d - \Delta$, our picture is quite different in several aspects. The disorder does not induce an RG flow, the RFFT being the same CFT at all scales for any finite value of $v \neq 0$. When $v = 0$ the $\alpha$-sector decouples both in the UV and in the IR and we have the usual flow $\text{GFT}_\Delta \to \text{GFT}_{d-\Delta}$. The $\lambda\phi^2$ deformation triggers instead a proper RG flow in the $(\alpha, \beta)$ theory, as in (5.10). In particular, we do not find evidence for the "disordered fixed point" depicted in figure 1 of [22].

[19]This is equivalent to the problem of long-range disorder, which has a long history, see e.g. [34–36].

where we notice that $\omega$ now has a kinetic term so it is no longer a simple auxiliary field. The scaling of the operators is $\Delta_\varphi = 2(\Delta_1 - \Delta_2) + \frac{d}{2}$, $\Delta_\chi = \Delta_1$ and $\Delta_\omega = 2\Delta_2 - \frac{d}{2}$. By choosing $\Delta_1 = 1 + \Delta$ and $\Delta_2 = 1 + \frac{d}{4} + \frac{\Delta}{2}$ parameterized by a single constant $\Delta$, we obtain that the three operators have dimensions $\Delta_\varphi = \Delta$, $\Delta_\chi = \Delta + 1$ and $\Delta_\omega = \Delta + 2$. In this case, when we map the action to fermionic variables we find a generalized free model that respects Parisi-Sourlas supersymmetry. This was recently studied in [37].

## 6 Final remarks

We have shown in this paper that the RFFT is invariant under a peculiar conformal symmetry under which the elementary field $\phi$ is not a primary operator. Adding the $h\phi$ interaction in (1.1) does not give rise to an RG flow, but directly to a new (UV) fixed point. Adding further local interactions to the action $S_0$ would instead give rise to an RG flow, but with starting point the new UV fixed point. In contrast to ordinary pure QFTs, the conformal behaviour is manifest only when considering specific linear combinations of (averaged) correlators of local operators. Aside from the emergence of exact global symmetries after quenched average, such as the $\mathbb{Z}_2$ symmetry $\phi \to -\phi$, we have shown that the RFFT also has a large set of new symmetries, such as the $\mathbb{Z}_2'$ in (3.51) or the continuous Heisenberg-like symmetries (3.60) and (3.66), which are not present in the action $S_0$ of a free scalar field. We have also started to explore how the notion of normal ordering of composite operators and OPE can be extended in the presence of quenched disorder. Such properties have been derived using the $(\alpha, \beta)$ description, the direct sum of two simple free CFTs. Despite its "rigidity" of not admitting a simple generalization in the presence of interactions, the $(\alpha, \beta)$ theory is itself an interesting theory with a large set of (higher-spin) currents, including peculiar bosonic nilpotent symmetries like the Heisenberg-like symmetry, worth further investigation.

Local deformations in the RFFT remain local and can be studied perturbatively in the Cardy theory, which is the actual CFT describing the RFFT. We investigated the symmetries of this model. Besides $O(-2)$ and conformal symmetry, it also enjoys a $O(-2)$-vector of Heisenberg-like symmetries and an infinite set of higher-spin extensions. We showed in a few examples how the new symmetries of this formulation (e.g. the $O(-2)$ symmetry) give rise to non-trivial selection rules on the RFFT correlators. Moreover, we discussed the connection between the bosonic Heisenberg-like symmetries with the fermionic Parisi-Sourlas symmetries. As a by-product of this work, we have clarified the role of the $\mathcal{S}_n$ replica symmetry in the Cardy theory, which is written as a composition of a $O(-2)$ rotation and a Heisenberg-like transformation. Replica symmetry is important because is preserved by all local deformations of the RFFT, such as the quartic interaction leading to the random field Ising model and the cubic one leading to the random field $\phi^3$ model. We plan to use the new understanding of this symmetry as a vantage viewpoint to give a fresh look to the RG of this important class of physical models.

## Acknowledgements

We thank G. Delfino for discussions. Work partially supported by INFN Iniziativa Specifica ST&FI. For the purpose of Open Access, a CC-BY public copyright license has been applied by the authors to the present document and will be applied to all subsequent versions up to the Author Accepted Manuscript arising from this submission.

## A  Spectrum of the $(\alpha, \beta)$ theory

The spectrum of primary operators of free scalar field theories can be obtained with standard techniques using conformal characters [38, 39]. Let $\chi(\phi)[q, \vec{x}]$ be the conformal character of the elementary field $\phi = \alpha, \beta$, where $q$ is the fugacity associated to the dilatation operator and $\vec{x} = (x_1, \dots, x_r)$ is the vector of $r$ fugacities labeling the representations of $SO(2r)$ or $SO(2r+1)$. In terms of an orthonormal basis of vectors $e_i \in \mathbb{R}^r$, $e_i^{(j)} = \delta_i^j$, $(i, j, = 1, \dots, r)$, the fugacities $x_i$ are defined as $x_i = \exp(e_i)(\ell) = e^{\ell_i}$, where $\ell = \sum_i \ell_i e_i$ is a weight of $SO(d)$ in the $e_i$ basis. The weights $\ell$ are half-integer valued and are related to the ordinary integer-valued Dynkin labels $\lambda_i$ as $\vec{\ell} = B^t(C^{-1})^t \vec{\lambda}$, where $\alpha_i = B_{ij} e_j$, with $\alpha_i$ the simple roots of the algebra $\mathfrak{so}(d)$ and $C$ its Cartan matrix. For example, for the spinor representation of $SO(2r+1)$ we have $\vec{\lambda} = (0, \dots, 0, 1)$, $\vec{\ell} = (1/2, \dots, 1/2)$. Notably, in both basis spin $\ell$ traceless symmetric representations are given by $(\ell, 0, \dots 0)$, reported simply as $\ell$ in what follows.

We discuss first the individual partition functions $Z_{\alpha, \beta}$ of the two free theories in isolation, given by

$$Z_\phi[q, \vec{x}] = \exp\left[ \sum_{k=1}^{\infty} \frac{1}{k} \chi(\phi)[q^k, \vec{x}^k] \right], \qquad \phi = \alpha, \beta, \tag{A.1}$$

where $\vec{x}^k = (x_1^k, \dots, x_r^k)$. The spectrum is obtained by re-expressing the partition function (A.1) in terms of a sum of conformal characters over the primary operators $\mathcal{O}$:

$$Z_\phi[q, \vec{x}] = \sum_{\mathcal{O}} g_\phi(\mathcal{O}) \chi(\mathcal{O})[q, \vec{x}], \tag{A.2}$$

where $g_\phi(\mathcal{O})$ is the multiplicity of the operator. The $\chi_{\mathcal{O}}$'s are the conformal characters associated with the primary operator $\mathcal{O}$. The elementary fields $\alpha$ and $\beta$, as well as the conserved currents (3.67) and (3.68) discussed in section 3.3, satisfy shortening conditions, and so will the corresponding characters. A general expression of $\chi(\mathcal{O})$ for $d = 2r$ and $d = 2r + 1$ dimensions has been given in [40]. Given an unconstrained primary operator $\mathcal{O}$ with scaling dimension $\Delta$ and spin $\ell = \sum_i \ell_i e_i$, we have

$$
\begin{aligned}
\chi(\mathcal{O})[q, \vec{x}] = \chi(\Delta, \vec{\ell})[q, \vec{x}] &= q^\Delta \left( \prod_{k=1}^r (1 - qx_k)^{-1} (1 - qx_k^{-1})^{-1} \right) \chi_{\vec{\ell}}(\vec{x}), && d = 2r, \\
\chi(\mathcal{O})[q, \vec{x}] = \chi(\Delta, \vec{\ell})[q, \vec{x}] &= \frac{q^\Delta}{1 - q} \left( \prod_{k=1}^r (1 - qx_k)^{-1} (1 - qx_k^{-1})^{-1} \right) \chi_{\vec{\ell}}(\vec{x}), && d = 2r + 1,
\end{aligned}
\tag{A.3}
$$

where $\chi_{\vec{\ell}}(\vec{x})$ are the $SO(d)$ characters of the spin $\ell$ representation. See eqs. (B.4) and (B.8) of [40] for their explicit expression in even and odd dimensions, respectively.

The short characters for the field $\alpha$ and for the conserved currents (3.68) of the $\alpha$-theory are

$$
\begin{aligned}
\chi(\alpha) &= \chi\left(\frac{d-4}{2}, 0\right) - \chi\left(\frac{d+4}{2}, 0\right), \\
\chi\left(\mathcal{J}_\ell^{(0)}\right) &= \chi(d-4+\ell, \ell) - \chi(d-1+\ell, \ell-3), \quad \ell \geq 4 \text{ even}, \\
\chi\left(\mathcal{J}_\ell^{(1)}\right) &= \chi(d-2+\ell, \ell) - \chi(d-1+\ell, \ell-1), \quad \ell \geq 2 \text{ even},
\end{aligned}
\tag{A.4}
$$

where for simplicity of notation we left implicit the $(q, \vec{x})$ dependence of the characters. The short characters for the field $\beta$ and for the conserved currents (3.67) of the $\beta$-theory are

$$
\begin{aligned}
\chi(\beta) &= \chi\left(\frac{d-2}{2}, 0\right) - \chi\left(\frac{d+2}{2}, 0\right), \\
\chi(\mathcal{I}_\ell) &= \chi(d-2+\ell, \ell) - \chi(d-1+\ell, \ell-1), \quad \ell \geq 2 \text{ even}.
\end{aligned}
\tag{A.5}
$$

We report in table 1 the spectrum of the first primary operators in the $\alpha$ and $\beta$ theories for $d = 5$ and $d = 6$. More precisely, we report all operators up to $\Delta \leq 5$ and $\Delta \leq 7$ in the $\alpha$ theory in $d = 5$ and $d = 6$, respectively, and all operators up to $\Delta \leq 21/2$ and $\Delta \leq 11$ in the $\beta$ theory in $d = 5$ and $d = 6$, respectively. We also report the multiplicity $g$ and the number of elementary fields $p$ of the operator. The latter is easily obtained by adding a further fugacity in the partition functions. Note that in the non-unitary $\Box^2$ $\alpha$-theory, it can happen that an operator at the unitary bound does not lead to a short multiplet, despite having an infinite number of descendants with zero norm. Indeed, while in a unitary theory null states are guaranteed to be orthogonal to all the states in the theory by the use of a Cauchy-Schwarz inequality, the latter does not hold in non-unitary theories. Zero-norm states arising from a conservation equation are instead guaranteed to be orthogonal to any other state also in non-unitary theories. Only the last ones are associated with short multiplets.[20] In table 1 blue rows correspond to these conserved operators. The null states which do not decouple recombine in a peculiar way, as pointed out in [21], where this structure was given the name of "extended Verma modules".[21] We observe that eventually the contributions of these states to the partition function can be cast in terms of two standard Verma modules. The operators involved in the recombination process are highlighted in orange and linked by arrows in table 1.

Let us see this in more detail. In panel 1a, the fourth primary operator in the list is a scalar operator ($\alpha^3$) with $\Delta = 3/2$. Despite the operator being at the unitarity bound, it does not satisfy a shortening condition. Its level-two scalar descendant $\Box \alpha^3$ is null and is proportional to the first scalar primary operator with $\Delta = 7/2$, which is also null. The states in these two Verma

---

[20]When writing the character decomposition (A.1), all primary operators are taken unit normalized. In this basis, a null primary gives rise to divergent OPE coefficients in three-point functions with other primary operators.

[21]The divergences in the conformal block expansion of 4-point functions due to the zero norm of such states nicely cancel in the recombination and the final conformal blocks associated with these operators are finite.

modules recombine and eventually give rise to the two full conformal characters with $p = 3$, $\Delta = 3/2$ and $\Delta = 7/2$. A similar phenomenon occurs with spinning operators which are at the unitary bound $\Delta_{\text{UB}} = d - 2 + \ell$, but are not conserved. The primary operators of this kind have $p = 6$ and have a null level-1 descendant, which is proportional to a primary operator with $p = 6$, scaling $\Delta_{\text{UB}} + 1$, and spin $\ell - 1$. The first operator of this kind appears at $\Delta = 6$ for $\ell = 2$, and is the only operator at $\Delta = 6$ which we report in panel 1a, the one in orange at the bottom of that panel. An analogous story applies in $d = 6$, where the scalar operator at the unitarity bound has $p = 2$ and $\Delta = 2$ ($\alpha^2$). It has a level-2 null descendant, and the scalar operator with $p = 2$, $\Delta = 4$, is proportional to it. The non-conserved spinning operators at the unitary bound have now $p = 4$, and are proportional again to a primary operator with $p = 4$, scaling $\Delta_{\text{UB}} + 1$, and spin $\ell - 1$. An operator of this kind appears at $\Delta = 8$ for $\ell = 3$, and is the only operator at $\Delta = 8$ which we report in panel 1b, the one in orange at the bottom of that panel. In $d = 8$ we also can have non-conserved spinning operators with $p = 3$ at the unitary bound, while in $d = 7$ and $d > 8$, there are no, non-conserved, primary operators at the unitarity bound.

For completeness, we report in panels 1c and 1d the first primary operators of the ordinary free theory in $d = 5$ and $d = 6$.

The full spectrum of the $(\alpha, \beta)$ theory is obtained by decomposing in irreducible characters the product $Z_\alpha Z_\beta$. In addition to the operators discussed before, it involves primary operators built with both $\alpha$ and $\beta$ fields. The short characters associated to the conserved primary currents $\mathcal{K}_\ell$ (3.69) are of the form

$$\chi(\mathcal{K}_\ell) = \chi(d - 3 + \ell, \ell) - \chi(d - 1 + \ell, \ell - 2), \qquad \ell \geq 2. \tag{A.6}$$

We report in table 2 the primary operators in the combined $(\alpha, \beta)$-sector up to $\Delta \leq 5$ and $\Delta \leq 7$ in $d = 5$ and $d = 6$, respectively. As before, blue rows correspond to conserved operators (short multiplets) and orange rows to operators with non-decoupled null states which recombine in a way similar to that described before. The operators involved in the recombination process are linked by arrows in table 2. As before, we report in table 2 the two operators (and only these) above $\Delta = 5$ and $\Delta = 7$ which are involved in a recombination process, the ones at the bottom of the panels 2a and 2b. Note that the conserved currents of type $\beta \partial_{\{\mu_1} \cdots \partial_{\mu_{\ell-1}\}} \Box \alpha + \cdots$ do not appear in the spectrum, as expected, being level-1 descendants of $\mathcal{K}_\ell$.

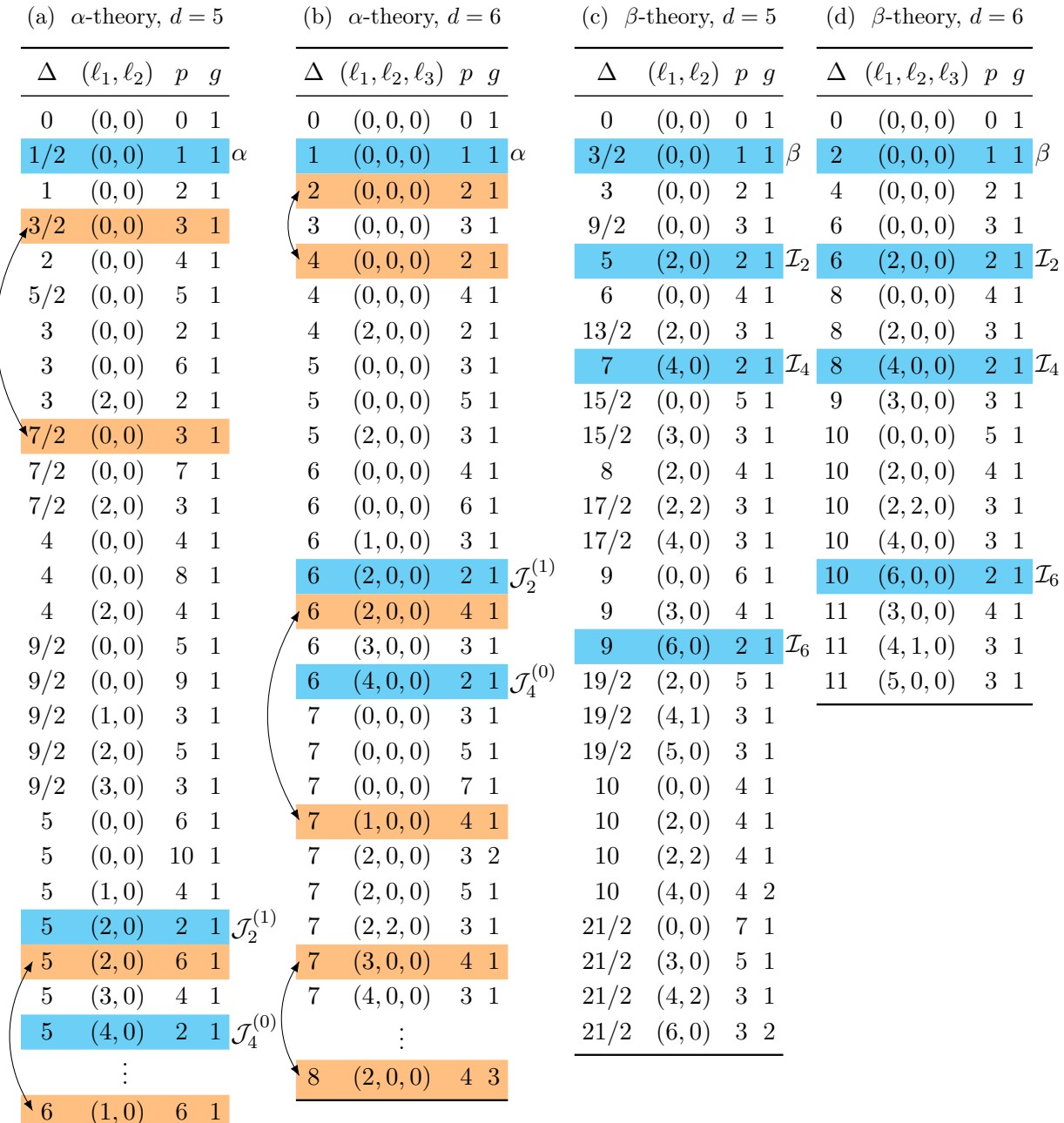

Table 1: Spectrum of the first operators in the free non-unitary $\Box^2$ theory ($\alpha$) and the ordinary free $\Box$ theory ($\beta$) for $d = 5$ and $d = 6$. $\Delta$ is the scaling dimension, $(\ell_1, \dots, \ell_r)$ are the labels of the $\mathfrak{so}(d)$ representations in the $e_i$-basis, $p$ is the number of elementary fields entering the primary operator and $g$ is its degeneracy. Blue rows denote conserved operators/short multiplets. Orange rows denote non-conserved operators at the unitarity bound with null states. The arrows denote the operators subject to a recombination effect, as described in the main text. Not all orange operators are involved in the recombination when their multiplicities differ (the number of arrows denoting the number of operators involved).

**(a)  $d = 5$**

| $\Delta$ | $(\ell_1, \ell_2)$ | $p_\alpha$ | $p_\beta$ | $g$ | |
|---|---|---|---|---|---|
| 2 | $(0,0)$ | 1 | 1 | 1 | |
| 5/2 | $(0,0)$ | 2 | 1 | 1 | |
| 3 | $(0,0)$ | 3 | 1 | 1 | |
| 3 | $(1,0)$ | 1 | 1 | 1 | |
| 7/2 | $(0,0)$ | 1 | 2 | 1 | |
| 7/2 | $(0,0)$ | 4 | 1 | 1 | |
| 7/2 | $(1,0)$ | 2 | 1 | 1 | |
| 4 | $(0,0)$ | 2 | 2 | 1 | |
| 4 | $(0,0)$ | 5 | 1 | 1 | |
| 4 | $(1,0)$ | 3 | 1 | 1 | |
| 4 | $(2,0)$ | 1 | 1 | 1 | $\mathcal{K}_2$ |
| 9/2 | $(0,0)$ | 2 | 1 | 1 | |
| 9/2 | $(0,0)$ | 3 | 2 | 1 | |
| 9/2 | $(0,0)$ | 6 | 1 | 1 | |
| 9/2 | $(1,0)$ | 1 | 2 | 1 | |
| 9/2 | $(1,0)$ | 4 | 1 | 1 | |
| 9/2 | $(2,0)$ | 2 | 1 | 2 | |
| 5 | $(0,0)$ | 1 | 3 | 1 | |
| 5 | $(0,0)$ | 3 | 1 | 1 | |
| 5 | $(0,0)$ | 4 | 2 | 1 | |
| 5 | $(0,0)$ | 7 | 1 | 1 | |
| 5 | $(1,0)$ | 2 | 2 | 1 | |
| 5 | $(1,0)$ | 5 | 1 | 1 | |
| 5 | $(2,0)$ | 3 | 1 | 2 | |
| 5 | $(3,0)$ | 1 | 1 | 1 | $\mathcal{K}_3$ |
| $\vdots$ | | | | | |
| 6 | $(1,0)$ | 3 | 1 | 3 | |

**(b)  $d = 6$**

| $\Delta$ | $(\ell_1, \ell_2, \ell_3)$ | $p_\alpha$ | $p_\beta$ | $g$ | |
|---|---|---|---|---|---|
| 3 | $(0,0,0)$ | 1 | 1 | 1 | |
| 4 | $(0,0,0)$ | 2 | 1 | 1 | |
| 4 | $(1,0,0)$ | 1 | 1 | 1 | |
| 5 | $(0,0,0)$ | 1 | 2 | 1 | |
| 5 | $(0,0,0)$ | 3 | 1 | 1 | |
| 5 | $(1,0,0)$ | 2 | 1 | 1 | |
| 5 | $(2,0,0)$ | 1 | 1 | 1 | $\mathcal{K}_2$ |
| 6 | $(0,0,0)$ | 2 | 1 | 1 | |
| 6 | $(0,0,0)$ | 2 | 2 | 1 | |
| 6 | $(0,0,0)$ | 4 | 1 | 1 | |
| 6 | $(1,0,0)$ | 1 | 2 | 1 | |
| 6 | $(1,0,0)$ | 3 | 1 | 1 | |
| 6 | $(2,0,0)$ | 2 | 1 | 2 | |
| 6 | $(3,0,0)$ | 1 | 1 | 1 | $\mathcal{K}_3$ |
| 7 | $(0,0,0)$ | 1 | 2 | 1 | |
| 7 | $(0,0,0)$ | 1 | 3 | 1 | |
| 7 | $(0,0,0)$ | 3 | 1 | 1 | |
| 7 | $(0,0,0)$ | 3 | 2 | 1 | |
| 7 | $(0,0,0)$ | 5 | 1 | 1 | |
| 7 | $(1,0,0)$ | 2 | 1 | 2 | |
| 7 | $(1,0,0)$ | 2 | 2 | 1 | |
| 7 | $(1,0,0)$ | 4 | 1 | 1 | |
| 7 | $(2,0,0)$ | 1 | 2 | 2 | |
| 7 | $(2,0,0)$ | 3 | 1 | 2 | |
| 7 | $(2,1,0)$ | 2 | 1 | 1 | |
| 7 | $(3,0,0)$ | 2 | 1 | 2 | |
| 7 | $(4,0,0)$ | 1 | 1 | 1 | $\mathcal{K}_4$ |
| $\vdots$ | | | | | |
| 8 | $(2,0,0)$ | 2 | 1 | 3 | |

Table 2: Spectrum of the first operators in the $(\alpha, \beta)$ theory for $d = 5$ and $d = 6$. $\Delta$ is the scaling dimension, $(\ell_1, \ldots, \ell_r)$ are the labels of the $\mathfrak{so}(d)$ representations in the $e_i$-basis, $p_{\alpha,\beta}$ are the number of elementary $\alpha, \beta$ fields entering the primary operator and $g$ is its degeneracy. Blue rows denote conserved operators/short multiplets. Orange rows denote non-conserved operators at the unitarity bound with null states. The arrows denote the operators subject to a recombination effect. Not all orange operators are involved in the recombination when their multiplicities differ (the number of arrows denoting the number of operators involved).

# B  Further details on the correlator mapping in the Cardy description

We report in this section some details regarding the correlator mapping between the RFFT and the Cardy descriptions.

## B.1  Two-point functions of quadratic fields

Let us consider the case of two-point functions of composite operators made out of two fields. In (2.27) and (2.24) we explained how to write normal ordered correlators in the RF model. Here, as a proof of principle, we will not assume (2.27) and (2.24) and we will check that these formulae are compatible with the normal ordering defined in the Cardy basis.

Keeping this in mind we define a list of all possible RF observables with two $\phi$ at position $x_1$ and two $\phi$ at position $x_2$ as follows,

$$
\begin{aligned}
V_{\mathrm{RF}}^{(2)} = \Big\{ & \overline{\langle\phi(x_1)\rangle^2 \, \langle\phi(x_2)\rangle^2}, \; \overline{\langle\phi(x_1)^2\rangle \, \langle\phi(x_2)\rangle^2}, \; \overline{\langle\phi(x_1)\rangle \, \langle\phi(x_2)\rangle \, \langle\phi(x_1)\phi(x_2)\rangle}, \\
& \overline{\langle\phi(x_1)\phi(x_2)\rangle^2}, \; \overline{\langle\phi(x_2)\rangle \, \langle\phi(x_1)^2\phi(x_2)\rangle}, \; \overline{\langle\phi(x_1)\rangle^2 \, \langle\phi(x_2)^2\rangle}, \\
& \overline{\langle\phi(x_1)^2\rangle \, \langle\phi(x_2)^2\rangle}, \; \overline{\langle\phi(x_1)\rangle \, \langle\phi(x_1)\phi(x_2)^2\rangle}, \; \overline{\langle\phi(x_1)^2\phi(x_2)^2\rangle}, \\
& \overline{\langle\phi(x_1)\rangle^2} \; \overline{\langle\phi(x_2)\rangle^2}, \; \overline{\langle\phi(x_1)^2\rangle} \; \overline{\langle\phi(x_2)\rangle^2}, \; \overline{\langle\phi(x_1)\rangle^2} \; \overline{\langle\phi(x_2)^2\rangle}, \; \overline{\langle\phi(x_1)^2\rangle} \; \overline{\langle\phi(x_2)^2\rangle} \Big\}.
\end{aligned}
\tag{B.1}
$$

There are 13 elements in $V_{\mathrm{RF}}^{(2)}$. The last four components of (B.1) are divergent and should be subtracted from the first 9 in some specific tuned way to obtain the finite results which were predicted by (2.27) and (2.24).

We now want to write a set of 13 correlators in the Cardy basis and get an invertible map from Cardy to RF. We want 9 normal ordered Cardy correlators which are finite. Also, we need 4 divergent correlators to match the divergent pieces in the RF theory: indeed divergent correlators of colliding Cardy fields can be mapped to RF variable e.g. $\langle\varphi^2(x)\rangle = \overline{\langle\phi(x)\rangle^2}$. Under these considerations, a possible list of two-point functions of Cardy fields reads

$$
\begin{aligned}
V_{\mathrm{Ca}}^{(2)} = \Big\{ & \langle{:}\varphi^2{:}\,{:}\varphi^2{:}\rangle, \; \langle{:}\varphi\chi_2{:}\,{:}\varphi\chi_3{:}\rangle, \; \langle{:}\chi_2\chi_2{:}\,{:}\chi_3\chi_3{:}\rangle, \; \langle{:}\chi_2\chi_2{:}\,{:}\chi_3\chi_4{:}\rangle, \\
& \langle{:}\chi_2\chi_3{:}\,{:}\chi_4\chi_5{:}\rangle, \; \langle{:}\chi_2\chi_3{:}\,{:}\varphi^2{:}\rangle, \; \langle{:}\varphi^2{:}\,{:}\chi_2\chi_3{:}\rangle, \; \langle{:}\varphi\chi_2{:}\,{:}\chi_3\chi_4{:}\rangle, \; \langle{:}\chi_2\chi_3{:}\,{:}\chi_4\varphi{:}\rangle, \\
& \langle\varphi^2(x_1)\rangle \, \langle\varphi^2(x_2)\rangle, \; \langle\varphi^2(x_1)\rangle \, \langle\chi_2^2(x_2)\rangle, \; \langle\chi_2^2(x_1)\rangle \, \langle\varphi^2(x_2)\rangle, \; \langle\chi_2^2(x_1)\rangle \, \langle\chi_2^2(x_2)\rangle \Big\}.
\end{aligned}
\tag{B.2}
$$

Here the first 9 components are two-point functions of normal ordered operators which give rise to a finite result (we keep the convention that the first operator is inserted at $x_1$ and the second at $x_2$), while the last 4 terms are the sets of divergent contributions built out of four fields.[22]

---

[22]We parametrize the divergent pieces with a product of correlators in order to match the product of averages in (B.1). We could have avoided products of correlators by considering single correlators of non-normal ordered operators. In both cases, we can show that all finite RF observables are functions of the first normal-ordered components.

We find that there is a invertible linear map between $V_{\mathrm{RF}}$ and $V_{\mathrm{Ca}}$ provided by

$$
V_{\mathrm{Ca}}^{(2)} = \left(M_{\mathrm{Ca}}^{(2)}\right)^{-1} V_{\mathrm{RF}}^{(2)}\,, \qquad
M_{\mathrm{Ca}}^{(2)} =
\begin{pmatrix}
1 & 0 & \frac{1}{2} & -\frac{3}{2} & 1 & 0 & 0 & -\frac{1}{4} & -\frac{1}{4} & 1 & 0 & 0 & 0 \\
1 & 0 & \frac{1}{4} & -\frac{3}{4} & \frac{1}{2} & 1 & 0 & -\frac{1}{4} & \frac{1}{4} & 1 & 0 & \frac{1}{2} & 0 \\
1 & 0 & \frac{1}{4} & -\frac{3}{4} & \frac{1}{2} & 0 & 1 & \frac{1}{4} & -\frac{1}{4} & 1 & \frac{1}{2} & 0 & 0 \\
1 & 1 & \frac{1}{4} & -\frac{3}{4} & \frac{1}{2} & 0 & 0 & 0 & 0 & 1 & 0 & 0 & 0 \\
1 & 0 & 1 & -2 & 1 & 1 & 1 & \frac{1}{4} & \frac{1}{4} & 1 & \frac{1}{2} & \frac{1}{2} & \frac{1}{4} \\
1 & 2 & -\frac{3}{4} & \frac{9}{4} & -\frac{3}{2} & 0 & 1 & \frac{5}{4} & \frac{1}{4} & 1 & \frac{1}{2} & 0 & 0 \\
1 & 2 & 1 & -\frac{7}{2} & 3 & 0 & 0 & \frac{1}{4} & \frac{1}{4} & 1 & 0 & 0 & 0 \\
1 & 2 & -\frac{3}{4} & \frac{9}{4} & -\frac{3}{2} & 1 & 0 & \frac{1}{4} & \frac{5}{4} & 1 & 0 & \frac{1}{2} & 0 \\
1 & 4 & -1 & 3 & -1 & 1 & 1 & \frac{9}{4} & \frac{9}{4} & 1 & \frac{1}{2} & \frac{1}{2} & \frac{1}{4} \\
0 & 0 & 0 & 0 & 0 & 0 & 0 & 0 & 0 & 1 & 0 & 0 & 0 \\
0 & 0 & 0 & 0 & 0 & 0 & 0 & 0 & 0 & 1 & 0 & \frac{1}{2} & 0 \\
0 & 0 & 0 & 0 & 0 & 0 & 0 & 0 & 0 & 1 & \frac{1}{2} & 0 & 0 \\
0 & 0 & 0 & 0 & 0 & 0 & 0 & 0 & 0 & 1 & \frac{1}{2} & \frac{1}{2} & \frac{1}{4}
\end{pmatrix}\,.
\tag{B.3}
$$

The last four components of $V_{\mathrm{Ca}}$ are written in terms of the last components in $V_{\mathrm{RF}}$. Instead, the first 9 components of $V_{\mathrm{Ca}}$ exactly provide the correct linear combinations of the 13 components in $V_{\mathrm{RF}}$ which define good CFT two-point functions. In particular, we checked that the 9 finite components are indeed written in terms of the normal ordered RF combinations predicted by (2.27) and (2.24).

Equation (B.3) therefore provides a one-to-one map between Cardy variables and random field ones. Let us stress that the vector (B.2) is only a possible choice of observables, e.g. we have chosen $\langle:\chi_i\chi_j::\varphi^2:\rangle$ with $(i,j) = (2,3)$ and we omitted $(i,j) = (2,2)$ because the latter does not provide a linearly independent combination of RF observables since $\langle:\chi_i\chi_j::\varphi^2:\rangle = G_{ij}\langle:\chi_2\chi_3::\varphi^2:\rangle$.

It is also important to mention that the basis (B.2) gives an overcomplete description of the Cardy theory, indeed many correlators in (B.2) vanish. e.g. in the free Cardy theory the components 6,7,8,9 of (B.2) vanish because $\langle\chi_i\varphi\rangle = 0$. Similarly, there are only two independent two-point functions of operators $\chi_i\chi_j$ because of $O(n-2)$ symmetry. Altogether these facts give a set of five constraints that RFFT correlators must satisfy, namely

$$
\sum_{j=1}^{13}\left(M_{\mathrm{Ca}}^{(2)}\right)^{-1}_{ij} V_{\mathrm{RF},j} = 0\,, \qquad\qquad (i=6,7,8,9)\,, \tag{B.4}
$$

$$
\sum_{j=1}^{13}\left[\left(M_{\mathrm{Ca}}^{(2)}\right)^{-1}_{3j} - 3\left(M_{\mathrm{Ca}}^{(2)}\right)^{-1}_{4j} + 2\left(M_{\mathrm{Ca}}^{(2)}\right)^{-1}_{5j}\right] V_{\mathrm{RF},j} = 0\,. \tag{B.5}
$$

We thus find that of the possible 9 linear combinations of RF correlators, 5 of them vanish because of simple properties of the Cardy theory.

## B.2 Three-point functions

The simplest set of three-point functions in the RFFT is written in terms of one quadratic field and two fundamental fields. Again we will not assume the knowledge of the normal ordered RF correlators provided by (2.27) and (2.24) and we will check that the result is compatible with such formulae. The RF observables are

$$
\begin{aligned}
V_{\text{RF}}^{(3)} = \Big\{ & \overline{\langle\phi(x_1)\rangle^2 \langle\phi(x_2)\rangle \langle\phi(x_3)\rangle}, \; \overline{\langle\phi(x_1)\rangle^2 \langle\phi(x_2)\phi(x_3)\rangle}, \; \overline{\langle\phi(x_1)\rangle \langle\phi(x_2)\rangle \langle\phi(x_1)\phi(x_3)\rangle}, \\
& \overline{\langle\phi(x_1)\rangle \langle\phi(x_1)\phi(x_2)\rangle \langle\phi(x_3)\rangle}, \; \overline{\langle\phi(x_1)\phi(x_2)\rangle \langle\phi(x_1)\phi(x_3)\rangle}, \; \overline{\langle\phi(x_1)\rangle \langle\phi(x_1)\phi(x_2)\phi(x_3)\rangle}, \\
& \overline{\langle\phi(x_1)^2\rangle \langle\phi(x_2)\rangle \langle\phi(x_3)\rangle}, \; \overline{\langle\phi(x_1)^2\rangle \langle\phi(x_2)\phi(x_3)\rangle}, \; \overline{\langle\phi(x_2)\rangle \langle\phi(x_1)^2\phi(x_3)\rangle}, \\
& \overline{\langle\phi(x_1)^2\phi(x_2)\rangle \langle\phi(x_3)\rangle}, \; \overline{\langle\phi(x_1)^2\phi(x_2)\phi(x_3)\rangle}, \; \overline{\langle\phi(x_1)^2\rangle} \; \overline{\langle\phi(x_2)\rangle \langle\phi(x_3)\rangle}, \\
& \overline{\langle\phi(x_1)^2\rangle} \; \overline{\langle\phi(x_2)\phi(x_3)\rangle}, \; \overline{\langle\phi(x_1)\rangle^2 \langle\phi(x_2)\rangle \langle\phi(x_3)\rangle}, \; \overline{\langle\phi(x_1)\rangle^2} \; \overline{\langle\phi(x_2)\phi(x_3)\rangle} \Big\} ,
\end{aligned}
\tag{B.6}
$$

where the last 4 components are just divergent pieces used to normal order the first 11 components.

Similarly, a choice of Cardy observables is

$$
\begin{aligned}
V_{\text{Ca}}^{(3)} = \Big\{ & \langle :\!\varphi^2\!: \varphi\varphi \rangle, \langle :\!\varphi\chi_2\!: \varphi\chi_3 \rangle, \langle :\!\chi_2\chi_2\!: \chi_3\varphi \rangle, \langle :\!\varphi^2\!: \chi_2\chi_3 \rangle, \langle :\!\chi_2\chi_3\!: \varphi\varphi \rangle, \langle :\!\chi_2\chi_3\!: \varphi\chi_4 \rangle, \\
& \langle :\!\chi_2\chi_3\!: \chi_4\varphi \rangle, \langle :\!\varphi\chi_2\!: \chi_3\chi_4 \rangle, \langle :\!\chi_2\chi_3\!: \chi_4\chi_2 \rangle, \langle :\!\chi_2\chi_3\!: \chi_3\chi_2 \rangle, \langle :\!\chi_2\chi_3\!: \chi_4\chi_5 \rangle, \\
& \langle \varphi(x_1)^2 \rangle \langle \varphi(x_2)\varphi(x_3) \rangle, \langle \chi_2(x_1)\chi_3(x_1) \rangle \langle \varphi(x_2)\varphi(x_3) \rangle, \\
& \langle \varphi(x_1)^2 \rangle \langle \chi_2(x_2)\chi_3(x_3) \rangle, \langle \chi_2(x_1)\chi_3(x_1) \rangle \langle \chi_2(x_2)\chi_3(x_3) \rangle \Big\} ,
\end{aligned}
\tag{B.7}
$$

where the first 11 components are finite (and the operators are inserted in order at the points $x_1$, $x_2$ and $x_3$), while the last four are used to reproduce the four last components in $V_{\text{RF}}^{(3)}$ (also in this case footnote 22 applies). Again we find an invertible map between the two sets of correlators

$$
V_{\text{Ca}}^{(3)} = \left( M_{\text{Ca}}^{(3)} \right)^{-1} V_{\text{RF}}^{(3)} , \qquad
M_{\text{Ca}}^{(3)} =
\begin{pmatrix}
1\,0\,0\,0\,0 & -\frac{1}{8} & -\frac{1}{8} & -\frac{1}{4} & -\frac{3}{2} & \frac{1}{2} & 1\,1\,0\,0\,0 \\
1\,0\,0\,1\,0 & -\frac{1}{8} & -\frac{1}{8} & \frac{1}{4} & -\frac{3}{4} & \frac{1}{4} & \frac{1}{2}\,1\,0\,1\,0 \\
1\,1\,0\,0\,0 & \frac{1}{8} & -\frac{1}{8} & 0 & -\frac{3}{4} & \frac{1}{4} & \frac{1}{2}\,1\,0\,0\,0 \\
1\,0\,1\,0\,0 & -\frac{1}{8} & \frac{1}{8} & 0 & -\frac{3}{4} & \frac{1}{4} & \frac{1}{2}\,1\,0\,0\,0 \\
1\,1\,1\,0\,0 & \frac{1}{8} & \frac{1}{8} & \frac{1}{4} & -2 & 1 & 1\,1\,0\,0\,0 \\
1\,1\,1\,1\,0 & \frac{1}{8} & \frac{1}{8} & \frac{5}{4} & \frac{9}{4} & -\frac{3}{4} & -\frac{3}{2}\,1\,0\,1\,0 \\
1\,0\,0\,0\,1 & \frac{1}{8} & \frac{1}{8} & -\frac{1}{4} & -\frac{3}{4} & \frac{1}{4} & \frac{1}{2}\,1\,1\,0\,0 \\
1\,0\,0\,1\,1 & \frac{1}{8} & \frac{1}{8} & \frac{1}{4} & -5 & 1 & 5\,1\,1\,1\,1 \\
1\,2\,0\,0\,1 & \frac{9}{8} & \frac{1}{8} & \frac{1}{4} & \frac{9}{4} & -\frac{3}{4} & -\frac{3}{2}\,1\,1\,0\,0 \\
1\,0\,2\,0\,1 & \frac{1}{8} & \frac{9}{8} & \frac{1}{4} & \frac{9}{4} & -\frac{3}{4} & -\frac{3}{2}\,1\,1\,0\,0 \\
1\,2\,2\,1\,1 & \frac{9}{8} & \frac{9}{8} & \frac{9}{4} & 3 & -1 & -1\,1\,1\,1\,1 \\
0\,0\,0\,0\,0 & 0 & 0 & 0 & 0 & 0 & 0\,1\,1\,0\,0 \\
0\,0\,0\,0\,0 & 0 & 0 & 0 & 0 & 0 & 0\,1\,1\,1\,1 \\
0\,0\,0\,0\,0 & 0 & 0 & 0 & 0 & 0 & 0\,1\,0\,0\,0 \\
0\,0\,0\,0\,0 & 0 & 0 & 0 & 0 & 0 & 0\,1\,0\,1\,0
\end{pmatrix} .
\tag{B.8}
$$

We can again check that the last 4 components of the two vectors map into each other as they should. Moreover, we checked that the first 11 components of $V_{\mathrm{Ca}}$ are written in terms of the normal ordered expressions of (2.27) and (2.24).

The vector $V_{\mathrm{Ca},i}$ contains vanishing correlators for $i = 4, 5, 6, 7, 8$. Moreover there exists a single independent three-point function of $\langle :\chi_i\chi_j: \chi_k\chi_l\rangle$ which takes the form

$$\langle :\chi_i\chi_j(x_1): \chi_k(x_2)\chi_l(x_3)\rangle = (G_{ik}G_{jl} + G_{il}G_{jk})\frac{\kappa^2}{x_{12}^{d-2}x_{13}^{d-2}} \,. \tag{B.9}$$

Using these constraints we can write all normal ordered RFFT correlators in terms of only four normal ordered correlators in $V_{\mathrm{Ca}}$. An example of such formulae is shown in (4.26).

## B.3 Four-point functions

Finally, we consider the case of the four-point functions defined in the main text. The vectors of observables $V_{\mathrm{RF}}$ in (2.17) and $V_{\mathrm{Ca}}$ in (4.23) are linearly related by the invertible map (4.24) where $M_{\mathrm{Ca}}^{(4)}$ reads

$$M_{\mathrm{Ca}}^{(4)} = \begin{pmatrix}
1\ 0\ 0\ 0\ 0\ 0\ 0 & -\frac{1}{12} & -\frac{1}{12} & -\frac{1}{12} & -\frac{1}{12} & -\frac{3}{2} & \frac{1}{2} & 0 & 1 \\
1\ 0\ 1\ 0\ 0\ 0\ 0 & \frac{1}{12} & -\frac{1}{12} & \frac{1}{12} & -\frac{1}{12} & -\frac{3}{4} & \frac{1}{4} & 0 & \frac{1}{2} \\
1\ 0\ 0\ 0\ 1\ 0\ 0 & \frac{1}{12} & -\frac{1}{12} & -\frac{1}{12} & \frac{1}{12} & -\frac{3}{4} & \frac{1}{4} & 0 & \frac{1}{2} \\
1\ 1\ 0\ 0\ 0\ 0\ 0 & \frac{1}{12} & \frac{1}{12} & -\frac{1}{12} & -\frac{1}{12} & -\frac{3}{4} & \frac{1}{4} & 0 & \frac{1}{2} \\
1\ 0\ 0\ 0\ 0\ 1\ 0 & -\frac{1}{12} & \frac{1}{12} & -\frac{1}{12} & \frac{1}{12} & -\frac{3}{4} & \frac{1}{4} & 0 & \frac{1}{2} \\
1\ 0\ 0\ 1\ 0\ 0\ 0 & -\frac{1}{12} & \frac{1}{12} & \frac{1}{12} & -\frac{1}{12} & -\frac{3}{4} & \frac{1}{4} & 0 & \frac{1}{2} \\
1\ 0\ 0\ 0\ 0\ 0\ 1 & -\frac{1}{12} & -\frac{1}{12} & \frac{1}{12} & \frac{1}{12} & -\frac{3}{4} & \frac{1}{4} & 0 & \frac{1}{2} \\
1\ 0\ 1\ 0\ 0\ 1\ 0 & \frac{1}{12} & \frac{1}{12} & \frac{1}{12} & \frac{1}{12} & -2 & \frac{2}{3} & \frac{1}{3} & 1 \\
1\ 0\ 0\ 1\ 1\ 0\ 0 & \frac{1}{12} & \frac{1}{12} & \frac{1}{12} & \frac{1}{12} & -5 & \frac{4}{3} & -\frac{1}{3} & 5 \\
1\ 1\ 0\ 0\ 0\ 0\ 1 & \frac{1}{12} & \frac{1}{12} & \frac{1}{12} & \frac{1}{12} & -2 & 1 & 0 & 1 \\
1\ 1\ 1\ 0\ 1\ 0\ 0 & \frac{3}{4} & \frac{1}{12} & \frac{1}{12} & \frac{1}{12} & \frac{9}{4} & -\frac{3}{4} & 0 & -\frac{3}{2} \\
1\ 1\ 0\ 1\ 0\ 1\ 0 & \frac{1}{12} & \frac{3}{4} & \frac{1}{12} & \frac{1}{12} & \frac{9}{4} & -\frac{3}{4} & 0 & -\frac{3}{2} \\
1\ 0\ 1\ 1\ 0\ 0\ 1 & \frac{1}{12} & \frac{1}{12} & \frac{3}{4} & \frac{1}{12} & \frac{9}{4} & -\frac{3}{4} & 0 & -\frac{3}{2} \\
1\ 0\ 0\ 0\ 1\ 1\ 1 & \frac{1}{12} & \frac{1}{12} & \frac{1}{12} & \frac{3}{4} & \frac{9}{4} & -\frac{3}{4} & 0 & -\frac{3}{2} \\
1\ 1\ 1\ 1\ 1\ 1\ 1 & \frac{3}{4} & \frac{3}{4} & \frac{3}{4} & \frac{3}{4} & 3 & -1 & 0 & -1
\end{pmatrix}. \tag{B.10}$$

It is easy to see that the list $V_{\mathrm{Ca}}$ is overcomplete because some correlators trivially vanish, while others can be related by symmetry. As an example in (4.23) there are four correlators of three fields $\chi_i$ and one $\varphi$, which all vanish in free theory. Similarly, there are four correlators of $\chi_i$ but by symmetry only three of them are independent (there are only three 4-index invariant tensors corresponding to the projectors from the tensor product of two $O(n-2)$ vectors into singlet, traceless symmetric and anti-symmetric representation).

The two examples above give rise to the following selection rules on RF correlators[23]

$$\sum_j \left(M_{\mathrm{Ca}}^{(4)}\right)^{-1}_{ij}(V_{\mathrm{RF}})_j = 0\,, \qquad (i = 8, 9, 10, 11)\,, \tag{B.11}$$

$$\sum_j \left[3\left(M_{\mathrm{Ca}}^{(4)}\right)^{-1}_{12j} - \left(M_{\mathrm{Ca}}^{(4)}\right)^{-1}_{13j} - 2\left(M_{\mathrm{Ca}}^{(4)}\right)^{-1}_{15j}\right](V_{\mathrm{RF}})_j = 0\,. \tag{B.12}$$

The presence of vanishing (linear combinations of) Cardy correlators means that $V_{\mathrm{RF}}$ can be reconstructed by summing over a smaller number of terms. In particular, using the selection rules above, we can obtain all terms in $V_{\mathrm{RF}}$ by summing over 10 Cardy correlators as shown in equation (4.25) in the main text.

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
