# Peer review of "The random free field scalar theory"

_SciPost Physics_

## Round 1 · Referee Report · Anonymous (Referee 1) · 2024-11-16

Strengths

1. a simple exactly solvable model
2. a pedagogical paper.

Weaknesses

1. difficult to reconcile with other methods
2. the model is well defined only in d>4

Report

There has been growing interest in reconsidering random field spin models, particularly the Random Field Ising Model (RFIM), using a representation that dates back to the linear transformations of fields proposed by J. Cardy in 1985. This is a complex problem that has remained unresolved for many years. Significant progress was achieved about a decade ago through the use of functional renormalization group methods, especially in their non-perturbative form. Recent advances in conformal field theory methods have provided an alternative approach to studying this problem; however, to my knowledge, the perspectives offered by the two approaches have yet to be reconciled. This is difficult to realize because, despite the fact that both methods use the renormalization group language, the ways in which they attack this difficult problem are very different.

In the present manuscript the authors apply the conformal field theory (CFT) approach to a much simpler system, the random free field theory (RFFT). This is an exactly solvable model without interactions but with quenched random field. They show that the RFFT after changing variables can be viewed as a simple non-interacting CFT whose primary operators are related to the original elementary fields in a peculiar way.
In contrast to ordinary field theories, conformal behavior is observed only when considering specific linear combinations of correlators of local operators.
The authors provide two different effective conformal field theories with and without replicas and derive the explicit maps of correlation functions between these theories and the original theory with quenched random field.

The random field system with local interactions can then be studied using the RG flow considering the above effective conformal theories as a starting UV fixed point.

The problem under consideration is interesting, and the presentation is clear and pedagogically sound, making the manuscript suitable for publication. However, I am uncertain whether it meets the high standards of novelty and breakthrough required for general acceptance in SciPost Physics.
I encourage the authors to address several points that could enhance the manuscript and add significant value.

As mentioned by the authors, the RFFT model does not satisfy cluster decomposition for $d<4$. A natural question that arises for the reader, therefore, is: what are the consequences of this for the scaling behavior in physically relevant dimensions below $d=4$?

Is the RFFT model in the same universality class as the critical RFIM above $d=6$?

Since the the RFFT model is exactly solvable, could one compute the exact distribution of the order parameter at criticality?

The authors consider an extension of their model to the case where the action $S_0$ is replaced by a generalized free theory. Could the calculations be further extended to the case of correlated random fields, such as power-law correlated random fields?

In the seminal RG framework for the critical behavior of spin systems with random fields, one typically refers to a zero-temperature or infinite-disorder fixed point that governs the criticality of these disordered systems. It would be useful to discuss the connection between this picture and the approach developed by the authors.

The authors identify the $O(-2)$ symmetry in their CFT. Another known example of disordered systems that exhibit this hidden symmetry is Charge Density Waves (CDW) pinned by disorder, which is also described by a zero-temperature fixed point and can be mapped onto a pure system with $O(-2)$ symmetry (see K.J. Wiese and A.A. Fedorenko, Phys. Rev. Lett. 123, 197601 (2019)). Is this a common property of systems controlled by a zero-temperature fixed point?

Recommendation

Ask for major revision

  • validity: good
  • significance: good
  • originality: good
  • clarity: good
  • formatting: good
  • grammar: good

Author:  Marco Serone  on 2024-11-25  [id 4985]

(in reply to Report 1 on 2024-11-16)
Category:
answer to question
reply to objection

We thank the referee for the feedback and suggestions which we have carefully considered. Below we reply to each comment.

1) "As mentioned by the authors, the RFFT model does not satisfy cluster decomposition for $d<4$. A natural question that arises for the reader, therefore, is: what are the consequences of this for the scaling behaviour in physically relevant dimensions below $d=4$?''

The random free field theory (RFFT) is actually well-defined for any dimension (it is, in the end, a Gaussian theory), but it has some subtle features in $d=2,4$ because of the appearance of logarithmic terms in the correlators. In the previous version, we wanted to avoid the discussion of these subtleties and for this reason, we focused on $d>4$, where the dimensions of all operators are positive (and thus the model satisfies cluster decomposition). Note that the model is non-unitary and the dimensions of the operators are allowed to take negative values (this is e.g. the case also for the Yang-Lee minimal model). Therefore focusing on $d>4$ is not a necessary requirement, but it was only a simplifying assumption. We have thus relaxed this assumption and now we consider $d\neq 2,4$. We briefly mention the subtleties related to $d=2,4$ in the revised sentence after eq.(2.7). We also added a new paragraph at the end of section 3.1, after the introduction of the $(\alpha,\beta)$ theory, where we explain better this point. We also changed accordingly a sentence after eq.(5.8).

This also addresses the second weakness of the paper reported by the referee.

2) "Is the RFFT model in the same universality class as the critical RFIM above $d=6$?''

Yes, indeed. We included a paragraph on page 3 in the introduction on this important point.

3) "Since the RFFT model is exactly solvable, could one compute the exact distribution of the order parameter at criticality? ''

Indeed the RFFT is solvable (it is a CFT so it is automatically critical) and we explained how to compute all observables, which can be defined in terms of the conformal dimensions and OPE coefficients of the associated CFT. We also explained how to compute all possible averages of correlation functions of the order parameter. In particular if one just computes the average of the one point function of the order parameter the result vanishes by $\mathbb{Z}_2$ symmetry, as a consequence of e.g. (3.49).

4) "The authors consider an extension of their model to the case where the action $S_0$ is replaced by a generalized free theory. Could the calculations be further extended to the case of correlated random fields, such as power-law correlated random fields? ''

In equation (5.12) and below we did consider the case of both a generalized free theory and a long-range disorder.

5) About the last comment, we thank the referee for letting us know the reference Phys. Rev. Lett. 123, 197601 (2019) about $O(-2)$ symmetry. We added a citation to this work in section 4.1.

6) The RFFT is a conformal field theory, so it does not undergo any RG flow. We thus do not have any difficulty reconciling our results with any other (F)RG method, because there is no RG! For this reason, we also believe that it would not be appropriate to describe any connection of our work with the ``\emph{seminal RG framework [...]}''. Also, we would like to stress that in our work we explicitly compared 3 different formulations of the model. One of these is the formulation in terms of Cardy fields mentioned by the referee. We also introduced a new formulation, the \abtheory. Moreover, these two formulations are always compared with the original RFFT. We proved that the 3 formulations agree and we provided a large number of explicit examples which establish without any doubt that everything is consistent. In particular let us stress again that we compared all our results to the original RFFT, which is the model that we want to study. So there is nothing more that we can compare to.

As a final comment, if any (F)RG setup wants to reproduce the RFFT in the UV, then it should match the standard computations of the RFFT. In this case, our work can be used as a simplifying tool for these types of checks. Indeed both Cardy and \abtheory~formulations make the scaling properties of RFFT more manifest, making computations far less cumbersome than in the original RFFT formulation. So, if the UV fixed point of any (F)RG setup does not agree with both Cardy and \abtheory~formulations, then one would simply conclude that it does not describe the RFFT (since these 3 formulations are proven to be equivalent).

This clarification should address the first weakness and the fifth question raised by the referee.

We hope that the revised version of the manuscript can be accepted for publication.

---

## Round 1 · Referee Report · Anonymous (Referee 2) · 2024-11-25

Strengths

1) Clarity
2) Relevance to the theory of disordered systems as well as CFT descriptions

Weaknesses

1) Lack of consideration of previous literature

Report

In their manuscript Piazza, Serone and Tevisani consider a free scalar field theory coupled to a Gaussian quenched random source and derive two manifestly conformally invariant descriptions of the theory. One involves treating the external random source as a quantum field and the other one relies on Cardy’s parametrization of the replica fields after making use of the replica trick to handle the quenched disorder. The description of the theory greatly simplifies in these two formalisms, which allows the authors to describe operator product expansion, normal ordering of composite operators, and other properties of more standard conformal field theories while providing exact maps between the correlation functions in the original and in the new descriptions. The new description of the free random-source scalar field theory also allows the authors to unveil new, exotic continuous symmetries, in particular nilpotent ones that in Cardy’s parametrization bear some strong resemblance to the supersymmetries of the Parisi-Sourlas construction for the random-field Ising model.

Despite being focused only on a free theory, the paper clarifies a number of properties of random-field systems and casts them in the CFT language. It is interesting and clearly written. I therefore recommend publication in SciPost Physics.

I would nonetheless like the authors to consider the following comments that touch upon comparison with previous literature on random field models:

- In the Introduction, the others state that “basic notions of ordinary QFTs such as how to properly define an RG flow (…) have started to be analyzed for quenched disorder QFTs only very recently”. Here and in several places of the manuscript, the authors overlook the many studies based on the functional RG that has been developed to precisely define RG flows in the presence of quenched disorder. To cite a few: D. Fisher, Phys. Rev. B 31, 7233 (1985), P. Le Doussal et al., Phys. Rev. E 69, 026112 (2004), P. Le Doussal, Ann. Phys. 325, 49 (2010), K. Wiese, J. Phys. A 17, S1889 (2005), M. Baczyk et al., JSTAT P06010 (2014), M. Tissier et al., Phys. Rev. B 85, (2012), etc.

- In a similar vein, accommodating the existence of additional correlators (page 3) and addressing the issue of assigning scaling dimensions in random-field models (pages 4, 6, 8) is precisely what has been done with the notion of a zero-temperature fixed point and the introduction of an additional scaling dimension for a running temperature: see the articles cited above.

Clearly this literature is not framed in the same CFT language and has not addressed several points considered in the present paper but a few words of acknowledgement or, better, of comparison would be welcome.

- Long-range random-field models have been considered, still in a functional RG framework, in
M. Baczyk et al., Phys. Rev. B 88, 014204 (2013) and Balog et al., JSTAT P1017 (2014); the former in particular investigates the Parisi-Sourlas supersymmetry with long-range interactions and disorder correlations and its breakdown.

Recommendation

Ask for minor revision

  • validity: top
  • significance: good
  • originality: good
  • clarity: high
  • formatting: excellent
  • grammar: excellent

Author:  Marco Serone  on 2024-12-01  [id 5012]

(in reply to Report 2 on 2024-11-25)

We thank the referee for the nice feedback and the comments which we thoughtfully considered.

The three comments by the referee were related to missing citations to previous literature.
We would like to stress that the main idea of our paper is to clarify the CFT structure of the random free field theory,
where no RG flow is present. Because of this, the goal of our paper might have implications for RF models (being the starting point of RG flows when interactions are included)
but is not directly related to most of the vast RF literature, where interacting models are studied in a variety of ways
(not only by FRG techniques, but also by other methods like replica trick, Monte-Carlo simulations, etc.).
We think it would be improper and even misleading for the reader to have an excessive list of not so related references.
However, we agree with the referee that some sentences of the introduction were improper and prone to misinterpretation.
We therefore decided to make the introduction more sharp, we better stressed the goal of the paper, and we added a few more references.

More specifically, we removed the sentence ``basic notions of ordinary QFTs [...] have started to be analyzed for quenched disorder QFTs only very recently''.
We agree with the referee that the sentence was ambiguous and, in the way it was written, did not do justice to the previous literature on RF models.

We also rephrased the first two pages of the introduction, see around equation (1.1), after (1.4) and after (1.6).

In order to comply with the referee suggestion of giving more weight to the FRG results, we added 2 extra FRG references to the list of references about the RF Ising model.

Finally, we would like to thank the referee for letting us know about the paper 10.1103/PhysRevB.88.014204 which is very much relevant to our discussion in section 5.2.
We cited it at the end of the section.

We hope that the revised version of the manuscript can be accepted for publication.

---

## Round 2 · Referee Report · Anonymous (Referee 2) · 2025-2-6

Strengths

Same as in my first report

Report

The authors have satisfactorily responded to my comments and in my opinion to the other referee comments. I therefore recommend publication of the revised version.

Recommendation

Publish (meets expectations and criteria for this Journal)

---

## Round 2 · Referee Report · Anonymous (Referee 1) · 2025-2-12

Strengths

The manuscript introduces and studies a simple, exactly solvable model in a relatively pedagogical manner.

Weaknesses

1. It is not completely clear how this toy model can be practically useful for studying the interacting case, which is a really complex problem.

2. As a consequence, it is difficult to reconcile with other methods, particularly FRG and non-perturbative RG, since they can be applied only to the interacting case.

Report

The authors have clarified almost all the points raised in my first report. When I asked about the possibility of computing the critical order parameter distribution, I was referring to the entire distribution, similar to what was done in S. T. Bramwell et al., Phys. Rev. Lett. 84, 3744 (2000), and I. Balog, A. Rançon, and B. Delamotte, Phys. Rev. Lett. 129, 210602 (2022), rather than just the first moment, which trivially vanishes.

The authors have improved the manuscript, and I can recommend the revised version for publication in SciPost.

Recommendation

Publish (meets expectations and criteria for this Journal)

---

## Round 2 · Author Response

We thank the editor for the quick response. Below we reply to the editor's comments.

  • Your paper can be relevant as a pedagogical introduction of how O(2) appears; but this observation is not new. (You can simply put the coupling to zero in the cited and other work.)"

We do not think that our paper can be relevant just as an introduction to O(2) symmetry. This is never said neither in the paper, nor in any of our replies to the referees. It's in fact the other way around. We stress several times in our paper that O(2) was known. For example, we write in the introduction, first paragraph at page 5, While some symmetries, like O(2) and additional Z2 symmetries, are quite obvious and were already considered in [13, 14, 18], we show that new less trivial symmetries are present."

The novelty of our paper is well described in the abstract of the paper, which we invite the editor to carefully read.

  • In order that it is as pedagogical as possible, make the changes below."

The clarity and pedagogical value of our paper was clearly indicated by both referees.

  • Make clear how your work relates to the literature."

In addition to the changes implemented to comply with the referee's requests, we added footnote 3 where we refer the reader for comparisons with previous literature. As we explained in the reply to the second referee, our paper is theoretical in nature and no RG flow is present. We believe that an emphasis on comparisons would be misleading and improper.

  • " (ii) add a section on supersymmetry in the Parisi-Sourlas variables."

The paper actually already contains a section about Parisi-Sourlas supersymmetry written in the SUSY variables, like in the original paper. For reference, the section is called Parisi-Sourlas supersymmetry" and can be found in the index of the paper (sec.4.3.3).

  • Another paper not yet cited is Nucl. Phys. B 946 (2019) 114696."

We added this paper to our reference list.

We hope that the revised version of the manuscript will be accepted for publication.

---

## Editorial Decision

in_refereeing